# Swift Hydra: Self-Reinforcing Generative Framework for Anomaly Detection with Multiple Mamba Models

**Nguyen Do[1], Truc Nguyen[2], Malik Hassanaly[2], Raed Alharbi[3], Jung Taek Seo[4], My T. Thai[1]***
[1]University of Florida, FL, USA     [2]National Renewable Energy Laboratory, CO, USA
[3]Saudi Electronic University, Saudi Arabia     [4]Gachon University, South Korea
nguyen.do@ufl.edu, Truc.Nguyen@nrel.gov, Malik.Hassanaly@nrel.gov,
ri.alharbi@seu.edu.sa, seojt@gachon.ac.kr, mythai@cise.ufl.edu

## Abstract

Despite a plethora of anomaly detection models developed over the years, their ability to generalize to unseen anomalies remains an issue, particularly in critical systems. This paper aims to address this challenge by introducing Swift Hydra, a new framework for training an anomaly detection method based on generative AI and reinforcement learning (RL). Through featuring an RL policy that operates on the latent variables of a generative model, the framework synthesizes novel and diverse anomaly samples that are capable of bypassing a detection model. These generated synthetic samples are, in turn, used to augment the detection model, further improving its ability to handle challenging anomalies. Swift Hydra also incorporates Mamba models structured as a Mixture of Experts (MoE) to enable scalable adaptation of the number of Mamba experts based on data complexity, effectively capturing diverse feature distributions without increasing the model's inference time. Empirical evaluations on ADBench benchmark demonstrate that Swift Hydra outperforms other state-of-the-art anomaly detection models while maintaining a relatively short inference time. From these results, our research highlights a new and auspicious paradigm of integrating RL and generative AI for advancing anomaly detection.

## 1 Introduction

Anomaly detection remains one of the most pressing and challenging tasks in various applications ranging from cybersecurity in critical systems to big data analysis (Liao et al., 2013; Zhang et al., 2021b; Leibig et al., 2017; Yu et al., 2017; Sahu et al., 2024). In simple terms, an anomaly detection method often involves training a machine learning (ML) model that aims to identify unusual patterns in data that deviate from expected behaviors. One real-world challenge in realizing such an approach is the scarcity of available anomalies to train on and the lack of prior knowledge about unseen anomalies. For that reason, supervised methods, including techniques such as one-class metric learning (Görnitz et al., 2013; Pang et al., 2018a; Liu et al., 2019; Ruff et al., 2020) and one-sided anomaly-focused deviation loss (Pang et al., 2021; 2019c; Zhang et al., 2020), tend to overfit to known anomaly patterns and struggle to generalize to unseen anomalies.

Unsupervised methods (Venkataramanan et al., 2020; Zaheer et al., 2020; Zhou et al., 2020; Li et al., 2022; Livernoche et al., 2024), on the other hand, have gained traction for training anomaly detection models with synthetic anomalies, thereby demonstrating an auspicious approach to tackle the data scarcity and generalization issues. Common techniques (Schlegl et al., 2017; Nazari & Branco, 2021) using generative AI models such as Variational Auto Encoders (VAEs) (Kingma & Welling, 2013) and Generative Adversarial Networks (GANs) (Goodfellow et al., 2016) to generate novel synthetic anomalies on which a detection model can be trained. In order to significantly augment the generalization ability of anomaly detection models, the generated samples should be realistic and challenging enough to bypass detection. However, current methods based on these techniques

---

*Corresponding Author

lack a strategy to generate such samples. Moreover, they often struggle to synthesize diverse and high-quality anomalies due to the high complexity of training the generative models (e.g., vanishing gradients and model collapse issues) (Salimans et al., 2016; Arjovsky & Bottou, 2017). Other state-of-the-art models (Zhang et al., 2021a; An & Cho, 2015; Xu et al., 2022) encode the training data distribution and then determine the anomaly score of a newly observed data point using their reconstruction loss. This is based on the assumption that, since normal instances significantly outnumber anomalies, these models should show higher reconstruction losses for anomalies. Nonetheless, neural networks can memorize and reconstruct anomalies well. As a result, the reconstruction losses for both normal and anomalous samples become indistinguishable, undermining the effectiveness of anomaly detection (Child, 2021).

In this work, we take a new approach to foster a more strategic mechanism for generating synthetic anomalies that can tackle the above-mentioned challenges. Specifically, we introduce a reinforcement learning (RL) agent to guide the training of a Conditional VAE (C-VAE) (Sohn et al., 2015) model capable of synthesizing anomalous samples that are both challenging and diverse, which can be used to substantially augment anomaly detection models. The RL agent operates on the latent space of the C-VAE model and its reward function is strategically designed to balance the entropy of the generated samples and their ability to evade detection, presenting a key advantage of our training framework in generating more effective anomalies. Furthermore, with this reward function, we theoretically show that the agent can explore deterministically in the latent space to yield feasible actions, thereby tackling one of the most crucial efficiency problems in RL.

Additionally, the complexity of data generated presents a challenge for training an efficient anomaly detection model. We establish a lower bound on the error rate for any single detection model, showing that even an over-parameterized model cannot fully capture the intricate features of increasingly complex generated data. Moreover, this over-parameterized model could lead to significantly prolonged inference times, which is not ideal for real-time applications. This necessitates a scalable anomaly detection model capable of capturing the increasingly diverse feature distributions. To achieve this, we train Mamba models (Gu & Dao, 2024) structured as a Mixture of Experts (MoE) (Shazeer et al., 2017; Chen et al., 2022; Nguyen et al., 2024) where each expert specializes in different feature regions. Together with a proposed MoE training scheme, this allows for a scalable inference with arbitrarily complex input data without increasing inference times, as only relevant experts are activated for specific input. Our contributions are summarized as follows:

- We introduce a new systematic framework, namely Swift Hydra, for training an anomaly detection model based on synthetic anomalies strategically generated by an RL-guided C-VAE model. The efficiency of the detection model is enhanced via a Mixture of Mamba Experts, thereby enabling high detection accuracy while maintaining short inference time.

- We establish a theorem showing that the RL agent can perform gradient descent on the latent space to yield feasible actions in early training episodes. We also propose a new training scheme for MoE that tackles the "winner-take-all" issue (Fedus et al., 2022).

- Comprehensive experiments are conducted on ADBench, a benchmark including 57 datasets from various domains, to demonstrate the outperforming detection accuracy and the efficiency of inference of our model. The result suggests that RL and generative AI together inspire a new and promising paradigm for advancing anomaly detection.

## 2 PRELIMINARIES AND NOTATION

**Anomaly Detection.** Given observations from a system, represented by $\boldsymbol{x} = \{x_1, x_2, \ldots, x_N\}$ where $\boldsymbol{x} \in \mathbb{R}^{P \times N}$, $P$ is the feature space dimension and the objective is to determine whether each observation $\boldsymbol{x}_i \in \mathbb{R}^P$, for $i \in [N]$, is an anomaly. The approach to anomaly detection can vary depending on the availability of labeled data. In the unsupervised setting, the assumption is that no labeled data is available, and the dataset comprises a mix of unidentified normal and anomalous instances. In the supervised setting, a dataset $\mathcal{D} = \{(\boldsymbol{x}_i, \boldsymbol{y}_i), i = 1, 2, \ldots, N\}$ is used where each $\boldsymbol{x}_i$ is labeled as normal ($\boldsymbol{y}_i = -1$) or anomalous ($\boldsymbol{y}_i = 1$). This dataset is fully labeled with a known proportion of anomalies and normal data, rendering the detection process similar to binary classification with unbalanced classes, where there are typically fewer anomalous than normal instances. The semi-supervised or one-class classification method acts as a hybrid approach, where the training involves only normal data ($\mathcal{D}$ contains only $\boldsymbol{y}_i = -1$), and anomalies, if present, are

identified during inference. This method can also extend to partially labeled datasets, where some anomalies are labeled during training.

**Class-Conditional Data Generation.** In our work, we employ a Conditional Variational Autoencoder (C-VAE) (Sohn et al., 2015), denoted by $\mathcal{F}_\theta = \mathcal{M}_\phi \circ \mathcal{G}_\psi$, conditioned on anomalous data ($y = 1$). The parameters $\psi$ and $\phi$ represent the encoder and decoder, respectively, while $\theta$ encapsulates both sets of C-VAE parameters. The C-VAE operates as follows:

$$\mathcal{F}_\theta = \mathcal{M}_\phi \circ \mathcal{G}_\psi, \quad \hat{\boldsymbol{x}}_i = \mathcal{F}_\theta(\boldsymbol{x}_i, y_i) = \mathcal{M}_\phi(\mathcal{G}_\psi(\boldsymbol{x}_i, y_i)) = \mathcal{M}_\phi(\boldsymbol{z}_i, y_i) \tag{1}$$

where $\hat{\boldsymbol{x}}_i \in \mathbb{R}^P$ represents the reconstructed observation. The generator is trained by optimizing the Evidence Lower Bound (ELBO):

$$\mathcal{L}_{\text{C-VAE}}^{\text{ELBO}} = \mathbb{E}_{q_\psi(\boldsymbol{z}_i|\boldsymbol{x}_i, y_i)} \left[\log p_\phi(\boldsymbol{x}_i \mid \boldsymbol{z}_i, y_i)\right] - \mathbb{E}_{q_\psi(\boldsymbol{z}_i|\boldsymbol{x}_i, y_i)} \left[\log \frac{q_\psi(\boldsymbol{z}_i \mid \boldsymbol{x}_i, y_i)}{p_\phi(\boldsymbol{z}_i \mid y_i)}\right] \tag{2}$$

In the above equation, $\mathbb{E}_{q_\psi(\boldsymbol{z}_i|\boldsymbol{x}_i, y_i)} \left[\log p_\phi(\boldsymbol{x}_i \mid \boldsymbol{z}_i, y_i)\right]$ is called the reconstruction loss term, which aims to measure how well the model can reconstruct the input data from the latent representation. $\mathbb{E}_{q_\psi(\boldsymbol{z}_i|\boldsymbol{x}_i, y_i)} \left[\log \frac{q_\psi(\boldsymbol{z}_i|\boldsymbol{x}_i, y_i)}{p_\phi(\boldsymbol{z}_i|y_i)}\right]$ is called the KL divergence term, which serves to regularize the latent space by making the distribution of the latent variables close to a prior distribution, typically a standard Gaussian. Note that our C-VAE model is a combination of linear functions and 1-Lipschitz activation functions in which all layers are normalized. To generate a new anomalous sample $\tilde{\boldsymbol{x}}_i \in \mathbb{R}^P$, we sample from $\boldsymbol{z}_i \in \mathbb{R}^d \sim \mathcal{N}(\mu, \sigma)$, where $d$ is the latent space dimension, and $\mu$ and $\sigma$ are optimized parameters at the bottleneck. The decoder then transforms $z_i$ into $\tilde{\boldsymbol{x}}_i = \mathcal{M}_\phi(z_i, y_i = 1)$.

## 3 SWIFT HYDRA

This section introduces our Swift Hydra framework, as illustrated in Figure 1, which comprises two main modules: a Self-Reinforcing Generative Module and an Inference Module. First, the Self-Reinforcing Generative Module trains a generative model using RL to synthesize diverse and challenging anomalies. These generated samples are later appended to the original dataset $\mathcal{D}$. Second, from this new dataset, the Inference Module trains an efficient detector using the Mixture of Experts (MoE) technique, which includes a combination of multiple lightweight Mamba models specializing in different data clusters and a gating network directing each data point to the top $k$ experts for collaborative prediction.

### 3.1 SELF-REINFORCING GENERATIVE MODULE

This module includes two main models: a C-VAE generator, $\mathcal{F}_\theta$, that synthesizes new anomalies, and a large Mamba-based detector $\mathcal{W}_\kappa : \mathbb{R}^P \rightarrow [0, 1]$ parameterized by $\kappa$ that maps a sample $\boldsymbol{x}_i$ to a probabilistic score of it being an anomaly. Unlike conventional methods where the generator $\mathcal{F}_\theta$ relies solely on feedback from the detector $\mathcal{W}_\kappa$ to generate new samples which could lead to a model collapse (Salimans et al., 2016; Hassanaly et al., 2022) or vanishing gradient (Arjovsky & Bottou, 2017) problem, we instead leverage an RL agent to guide the training of $\mathcal{F}_\theta$. This RL agent, represented by a policy $\pi_\omega$, explores the latent distribution $p(z)$ of $\mathcal{F}_\theta$ and targets areas that would encourage the generator $\mathcal{F}_\theta$ to synthesize diverse and challenging anomalies that can bypass the detector $\mathcal{W}_\kappa$. These synthetic samples are then used to augment the current training dataset and retrain $\mathcal{F}_\theta$, ultimately improving its ability to generate better anomalies in future episodes.

**Dataset definitions.** The dataset $\mathcal{D}$ is split into a training set, $\mathcal{D}^{train}$, and a testing set, $\mathcal{D}^{test}$. In our approach, the goal is to attain high accuracy on the $\mathcal{D}^{test}$ even with a small training dataset $\mathcal{D}^{train}$. Let $\mathcal{D}^{balanced} = \{(x, y) \in \mathcal{D}^{train} \mid j = \min(|\{y = -1\}|, |\{y = 1\}|), |\{y = -1\}| = j, |\{y = 1\}| = j\}$ be a dataset balanced between normal and anomalous data points, with equal cardinalities determined by the smaller class. Note that, as episodes progress, the generator $\mathcal{F}_\theta$ combined with the RL-agent $\pi_\omega$ adds more anomalous samples to $\mathcal{D}^{train}$, expanding the anomalous data in $\mathcal{D}^{balanced}$. Since $\mathcal{D}^{balanced}$ ensures equal numbers of anomalous and normal data, the increase in anomalous data leads to a corresponding expansion of normal data as well. In the RL context, for each episode, $e$, we denote $\mathcal{D}_e^{train}$ and $\mathcal{D}_e^{balanced}$ as the evolving training and balanced datasets, respectively.

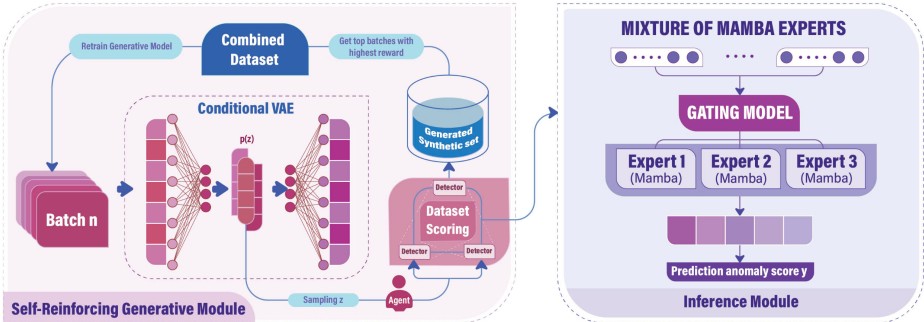

Figure 1: The Swift Hydra Framework consists of two main modules: the Self-Reinforcing Generative Module and the Inference Module. The first module includes a C-VAE, an RL agent, and a large Mamba-based Detector. Initially, the C-VAE is trained on the original dataset (referred to as the Combined Dataset in episode 0). In the early stages, the RL agent generates diverse anomalies by refining latent vectors $z$, then shifts to producing anomalies that more effectively deceive the detector. The top $l$ anomalies are added back to the original dataset, creating a new combined dataset to further improve the Generative Model. The second module employs a Mixture of Mamba Experts (MoME), where lightweight models specialize in different parts of the dataset, providing the same performance as the large detector but with significantly faster inference.

**Training process.** For each episode $e$ (comprising $h$ steps), the C-VAE generator $\mathcal{F}_\theta$ is trained with batches of $\mathcal{D}_e^{train}$ dataset, while the detector $\mathcal{W}_\kappa$ is trained with $\mathcal{D}_e^{balanced}$. Next, an anomalous data point $(x, y = 1)$ is sampled at random from $\mathcal{D}_{e,anomalous}^{train}$, and converted into a latent representation $z = \mathcal{G}_\psi(x, y = 1)$. The RL policy is tasked with generating a modification vector $\delta$ in the latent space, i.e. $\pi_\omega : \boldsymbol{z} \to \delta$. This $\delta$ results in a new sample in the latent space as $\hat{z} = z + \delta$. At the end of the episode, a new dataset is obtained $\hat{\mathcal{X}} = \{\mathcal{M}_\phi(\hat{\boldsymbol{z}}_0, y = 1), \ldots, \mathcal{M}_\phi(\hat{\boldsymbol{z}}_h, y = 1)\}$. From the newly generated set $\hat{\mathcal{X}}$, the top $l$ samples that lead to the highest rewards are selected and denoted as $\hat{\mathcal{X}}^{<l}$. A formal definition of the reward is provided in the next section. At each episode, the selected samples are then merged with $\mathcal{D}_{e-1}^{train}$, forming the evolving dataset $\mathcal{D}_e^{train} = \mathcal{D}_{e-1}^{train} \cup \hat{\mathcal{X}}^{<l}$. Note that we set $\mathcal{D}_0 = \mathcal{D}^{train}$ to ensure that the model $\mathcal{F}_\theta$ does not deviate from the acceptable range of the original data (Shumailov et al., 2024). The dataset $\mathcal{D}_e^{train}$ is used to retrain the generator $\mathcal{F}_\theta$, enhancing its ability to generate high-quality data in future episodes. As $e$ increases, $\mathcal{D}_e^{train}$ is incorporated into $\mathcal{D}^{balanced}$. Thus, $\mathcal{D}_e^{balanced}$ also grows across episodes. Due to page limit, we refer readers to Appendix **A.1** for the pseudocode and further details about the training process.

### 3.1.1 GENERATING SAMPLES AS A MARKOV DECISION PROCESS

The process of policy modeling can be structured as a Markov Decision Process (MDP) (Bellman, 1957), $\mathcal{M} \stackrel{\text{def}}{=} (\mathcal{S}, \mathcal{A}, T, \mathcal{R})$. This includes (i) a finite sets of states $\mathcal{S}$, (ii) a finite set of actions $\mathcal{A}$, (iii) a transition distribution $T(s' \mid s, a)$ where $s, s' \in \mathcal{S}, a \in \mathcal{A}$ and (iv) a reward function $\mathcal{R} : \mathcal{S} \times \mathcal{A} \to \mathbb{R}$. We specify each component as follows:

**States** $(s)$: A state is defined by latent space representations $s_i = (\boldsymbol{z}_i, \mathcal{D}_e^{train})$ for $\boldsymbol{z}_i = \mathcal{G}_\psi(\mathbf{x}_i)$ and $\mathbf{x}_i \in \mathcal{D}_{e,anomalous}^{train}$, where $\boldsymbol{z}_i$ is the latent vector produced by the encoder $\mathcal{G}_\psi$ from the input data $\mathbf{x}_i$.

**Actions** $(a)$: An action $a_i = (\mu_i, \sigma_i)$ is a vector of two components: $\mu_i \in \mathbb{R}^d$ (predicted mean) and $\sigma_i \in \mathbb{R}^d$ (predicted scale). The modification vector $\delta_i = \sigma_i \cdot \epsilon + \mu_i$, where $\epsilon \sim \mathcal{N}(0, I)$, and the latent vector is updated as $\hat{\boldsymbol{z}}_i = \boldsymbol{z}_i + \delta_i$.

**Rewards** $(\mathcal{R})$: The reward function is strategically designed to encourage the generation of a set of samples that are diverse and reduce the detector's confidence. The function is defined as follows:

$$\mathcal{R}(\mathcal{M}_\phi(\hat{z}_i, y_i = 1), e) = \gamma^e \cdot \mathcal{H}(\mathcal{D}_e^{train} \cup \mathcal{M}_\phi(\hat{z}_i, y_i = 1)) - \log \mathcal{W}_\kappa(\mathcal{M}_\phi(\hat{\boldsymbol{z}}_i, y_i = 1)), \quad (3)$$

where $\mathcal{H}(\mathcal{D}_e^{train} \cup \mathcal{M}_\phi(\hat{z}_i, y = 1))$ is the entropy of $\mathcal{D}_e^{train}$ after incorporating $\mathcal{M}_\phi(\hat{z}_i, y_i = 1)$ and is aimed at promoting the generation of diverse samples (additional details on the calculation of the entropy of $\mathcal{D}_e^{train}$ are provided in Appendix **A.3**). The function $\mathcal{W}_\kappa(\mathcal{M}_\phi(\hat{z}_i, y_i = 1))$ assesses the detector's likelihood of classifying the generated sample as anomalous, with the goal of reducing this probability to decrease the detector's confidence.

The hyperparameter $\gamma \in [0, 1]$ dictates the desired rate of entropy reduction. This $\gamma$ implies that the policy $\pi_\omega$ focuses on exploring rare samples to increase the diversity of $\mathcal{D}_{e,anomalous}^{train}$ in the early episodes. Once sufficient data has been explored, the reward function shifts to encourage the agent to exploit this data, generating new samples that are more effective at bypassing the detector.

**Transition Dynamics** ($T$): When an action $a_i = (\mu_i, \sigma_i)$ is taken, a new anomalous data point $\hat{x}_i = \mathcal{M}_\phi(\hat{z}_i, y_i = 1)$ is added to $\hat{\mathcal{X}}$. A new state $s_{i+1} = (z_{i+1}, \mathcal{D}_e^{\text{train}})$, where $z_{i+1} = \mathcal{G}_\psi(\mathbf{x}_{i+1})$, is then formed by selecting the next $x_{i+1} \in \mathcal{D}_{e,\text{anomalous}}^{\text{train}}$.

### 3.1.2 ONE-STEP TO FEASIBLE ACTIONS

The RL agent $\pi_\omega$, which is tasked with generating a new sample $\hat{x}_i$ from $x_i$, can be trained using conventional methods. However, during early training episodes, the agent would often struggle to find suitable actions that maximize the reward function because $\pi_\omega$ has not yet learned effective strategies. In fact, an action $a_i = (\mu_i, \sigma_i)$ may be invalid if the updated latent vector $\hat{z}_i$ derived from $a_i$ falls outside the supported range of the trained model $\mathcal{F}_\theta$. Even with advanced exploration techniques such as those in (Eysenbach & Levine, 2022; Pathak et al., 2017; Burda et al., 2019; Ecoffet et al., 2021), this issue remains challenging for $\pi_\omega$ to overcome due to the high-dimensional and continuous nature of the action space.

A naive strategy to address this is to use the observed data distribution $p(x)$ (i.e., adding Gaussian noise to $x_i$) to generate new samples $\hat{x}_i$. The encoder then provides their latent representation $\hat{z}_i = \mathcal{G}_\psi(\hat{x}_i, y = 1)$, and the modification vector $\hat{\delta}_i = \hat{z}_i - z_i$ is employed to guide exploration at that step. After that, a feasible action $\tilde{a}_i = (\hat{\delta}_i, \sigma_i)$ is derived from $\hat{\delta}_i$ to replace the invalid action $a_i = (\mu_i, \sigma_i)$ of the RL agent in the current step. Once a feasible action is identified, the agent learns it in a supervised manner, facilitating more effective exploration in future steps. However, randomly modifying observations in the input space $p(x)$ can be complex. Instead, we rely on the following theorem to find feasible actions:

**Theorem 1.** *(Reward Estimation Consistency). If the reward function $\mathcal{R}$ is differentiable, $\mathcal{F}_\theta$ is well-converged, and $\hat{z}_i := z_i - \epsilon \cdot \nabla_{z_i}(-\mathcal{R}(\mathcal{M}_\phi(z_i, y_i = 1), e))$ for some small $\epsilon$, then $\mathcal{R}(\hat{x}_i, e) > \mathcal{R}(x_i, e)$, where $\hat{x}_i = \mathcal{M}_\phi(\hat{z}_i, y_i = 1)$. (Proof in Appendix **B.1**)*

In other words, if $\mathcal{F}_\theta$ is well-converged and maintains both continuity (i.e., nearby points in the latent space yield similar content when decoded) and completeness (i.e., points sampled from the latent space produce meaningful content when decoded), the C-VAE described in Equation 2 can explore new states $s$ (i.e., anomalous observations) by utilizing the latent feature space $p(z)$ (which is learned from the original space $p(\mathbf{x})$). This allows us to search for feasible $\hat{x}_i$ in the lower-dimensional and less noisy latent space $p(z)$ as an alternative to creating feasible actions. Specifically, Theorem 1 implies that we can deterministically search for $\hat{z}_i$ in a manner that maximizes the reward function specified in Equation 3 using gradient descent (Ruder, 2017). From that, a feasible action $\tilde{a}_i = (\hat{\delta}_i, \sigma_i)$ is derived where $\hat{\delta}_i = \hat{z}_i - z_i$. With this approach, the policy, value, and reward models are trained simultaneously during these early episodes, allowing the RL agent to generalize effectively and reduce invalid actions in future episodes. Thus, the need for using one-step to feasible actions is eliminated in subsequent stages. Further details on this process can be found in Appendix **A.2** with a preliminary analysis given in Appendix **C.5**.

## 3.2 INFERENCE MODULE

At the conclusion of the first module, the detector $\mathcal{W}_\kappa$ has been augmented by the newly generated dataset and can be used as the final anomaly detection model. Due to the increasingly diverse training data generated by $\mathcal{F}_\theta$, we had to initially overparameterize the detector $\mathcal{W}_\kappa$. For that reason, deploying $\mathcal{W}_\kappa$ directly as the final detection model would not be scalable due to the high inference

cost. Furthermore, the theorem below establishes a lower bound on the detection error, showing that any single detection model is subject to this lower bound regardless of the number of parameters.

**Theorem 2.** *(Inefficiency of single detector in handling evolving balance data). Suppose a feature space $\mathfrak{X} \subset \mathbb{R}^P$ contains $U_n$ normal clusters and $U_a$ anomalous clusters, where each cluster $u$-th $\in [U_n + U_a]$ is modeled as a Gaussian distribution $\mathcal{N}(\boldsymbol{\mu}_u, \sigma^2 \mathbf{I}_P)$. Let $\mathcal{V}_{cluster}$ be the cluster's volume and $\Lambda$ be the total overlapping volume between normal and anomalous clusters, where the number of anomalous data points is equal to the number of normal data points, the training loss $\mathcal{L}_{train}(\mathcal{W}_\kappa)$ is lower bounded by $\frac{1}{4} \cdot \frac{\Lambda}{U_a \cdot V_{cluster} - \frac{\Lambda}{2}}$ in a case of linear $\mathcal{W}_\kappa$. (Proof in Appendix **B.2**)*

This theorem aligns with the findings in (Chen et al., 2022), emphasizing the inefficiency of using a single classifier. To address this issue, in this second module, we use the MoE approach to train an efficient detector on the dataset generated by the first module. Instead of relying on a single large-scale detector, this technique leverages multiple "expert" models, with each one specializing in a subset of the input data. The balanced dataset $\mathcal{D}_e^{balance}$ is first decomposed into clusters $\{\mathcal{C}_1, \mathcal{C}_2, \ldots, \mathcal{C}_U\}$, where the number of clusters $U$ is determined using the elbow method (Yuan & Yang, 2019). We train a set of Mamba models $\{f_1, f_2, \ldots, f_M\}$, each acting as an expert for a specific data cluster following the Sparsely-Gated Mixture-of-Experts approach (Shazeer et al., 2017), and where $f_m(x; \mathbf{W})$ is the output of the $m$-th expert network with input $x$ and parameter $\mathbf{W}$.

**Gating network.** In the mixture-of-experts approach, the experts are complemented by a gating network that directs inputs to the most appropriate expert. Given an input $x \in \mathbb{R}^\mathbb{P}$, the gating network is defined as the following function:

$$\mathbf{h}(x, \aleph_g, \aleph_{\text{noise}}) = x \cdot \aleph_{\text{g}} + \text{StandardNormal}(.) \cdot \text{Softplus}(x \cdot \aleph_{\text{noise}}) \tag{4}$$

where $\aleph_{\text{g}}, \aleph_{\text{noise}} \in \mathbb{R}^{P \times M}$ are weight matrices that determine the linear transformation and noise contribution, respectively. From the output of $\mathbf{h}(x, \aleph_g, \aleph_{\text{noise}})$, a key step is to apply the top $k$ expert selection mechanism, denoted by $\text{TopK}(\mathbf{h}(x, \aleph_g, \aleph_{\text{noise}}), k)$, where it selects the top $k$ largest values from the vector $\mathbf{h}(x, \aleph_g, \aleph_{\text{noise}})$, which represents the performance scores (e.g., accuracy) of different expert networks. The elements in $\mathbf{h}(x, \aleph_g, \aleph_{\text{noise}})$ that are not within the top $k$ are replaced by $-\infty$, effectively excluding them from further consideration. Finally, a softmax function is applied to these top $k$ values to normalize them, i.e., $\lambda(x, \aleph_g, \aleph_{\text{noise}}) = \text{Softmax}(\text{TopK}(\mathbf{h}(x, \aleph_g, \aleph_{\text{noise}}), k))$. This setup forms a Mixture of Mamba Expert, and the output of the MoE layer is then expressed as:

$$\mathfrak{F}(x, \aleph_g, \aleph_{\text{noise}}, \mathbf{W}) = \sum_{m \in \mathfrak{T}_x} \lambda_m(x, \aleph_g, \aleph_{\text{noise}}) f_m(x; \mathbf{W}) \tag{5}$$

where $\mathfrak{T}_{\mathbf{x}} \subseteq [M]$ represents the indices of selected experts ($|\mathfrak{T}_{\mathbf{x}}| = k$).

**Tackling "winner-take-all".** During early training of MoE, experts have arbitrary performance scores, hence the gating network could randomly allocate more samples to a particular expert. With more training data, this expert outperforms others, thus receiving even more samples. This is referred to as the "winner-take-all" phenomenon (Oster & Liu, 2005; Fedus et al., 2022), which reduces the MoE to a single lightweight expert, limiting its ability to generalize. While this expert may excel, it fails to capture the diverse features across $U$ clusters, undermining the model's overall performance.

We tackle this "winner-take-all" issue by temporarily deactivating the gating network during this early training stage and, instead, proposing a probabilistic approach to ensure diversity in cluster assignments across experts, while also considering the complexity of each cluster. For each expert $f_i$, where $i \in [M]$, instead of assigning clusters based on fixed criteria, we dynamically adjust the probability of an expert $f_i$ selecting a cluster $u_i \in [U]$, with the probability inversely proportional to how frequently the cluster has already been assigned to other experts. More importantly, we introduce a scaling factor that adjusts this probability based on the size of the cluster. For larger clusters, which are likely more complex, we reduce the penalty of being selected multiple times, as these clusters require more experts to fully capture their complexity. Specifically, the probability of expert $f_i$ selecting cluster $u$ is given by:

$$\mathcal{P}(u \mid \mathbf{x}, \{n_u\}, \{s_u\}) = \frac{\exp\left(c_0 - \frac{\alpha}{s_u} \cdot n_u\right)}{\sum_{u' \in [U]} \exp\left(c_0 - \frac{\alpha}{s_{u'}} \cdot n_{u'}\right)} \tag{6}$$

where $s_u$ is the size of cluster $u$, $n_u$ is the number of times cluster $u$ has already been assigned, $\alpha$ is the base penalty factor, and $c_0$ is a constant initialization score for cluster selection. Then, the expert $f_i$ will select $u \sim \text{Categorical}\left(\mathcal{P}\left(u \mid \mathbf{x}, \{n_u\}, \{s_u\}\right), u \in [U]\right)$ as its cluster.

Note that, due to the probabilistic nature of the selection algorithm, there could be clusters that are not selected by any experts. Therefore, we overspecify the number of experts $M$. As demonstrated in a theorem from (Nguyen et al., 2024), doing so does not increase prediction time. This is because $\mathfrak{F}(x, \aleph_g, \aleph_{\text{noise}}, \mathbf{W})$ selects only the top $k$ (typically 2 or 3) best experts for making predictions. After training each expert with its selected cluster, we train the gating network to minimize the overall classification loss (e.g., MSE or Cross Entropy Loss). With this setup, we also establish a theorem to demonstrate the effectiveness of our training mechanism as follows:

**Theorem 3.** *(MoME efficiently handles evolving balance data). Let $\mathcal{L}_{test}(\mathfrak{F})$ and $\mathcal{L}_{test}(\mathcal{W}_\kappa)$ represent the expected error on the test set for the Mixture of Mamba Experts (MoME) model and a single detector, respectively. For any value of $\Lambda$, employing MoME with $\{f_1, f_2, \ldots, f_M\}$ guarantees that the minimum expected error on the training set is $\mathcal{L}_{train}(\mathfrak{F}) = 0$ and the expected error on the test set satisfies $\mathcal{L}_{test}(\mathfrak{F}) \leq \mathcal{L}_{test}(\mathcal{W}_\kappa)$. (Proof in Appendix **B.3**)*

The above theorem demonstrates that a Mixture of Mamba Experts model can effectively fit all the data in the training set. Moreover, the expected error on the test set when using the Mixture of Mamba Experts will always be less than or equal to that of a single detector. Once the experts are well-trained, we activate the gating network and use it for routing samples.

# 4 EXPERIMENTAL EVALUATION

**Settings.** We conduct experiments to evaluate the performance of our Swift Hydra framework using the ADBench benchmark (Han et al., 2022), which includes a comprehensive collection of 57 widely used anomaly detection datasets spanning various tasks, from image analysis to natural language processing, as detailed in Appendix **C.6**. We also evaluate a version of Swift Hydra without MoME, i.e., a single large detector is used in the Inference Module. The implementation specifics, such as the training algorithm, model architecture, hyperparameter, model size, and training costs are provided in Appendix **C.1**. We will release the source code once the paper is published.

**Metrics.** In our evaluation, we focus on the performance of the Swift Hydra, particularly in terms of AUC-ROC and TIF (total inference time to predict all data in ADBench). Additionally, we analyze the distribution of generated data at each episode and compare it to the distribution of the test data.

**Baselines.** For the anomaly detection task, we compare Swift Hydra against several state-of-the-art (SOTA) semi-supervised and unsupervised learning methods included in ADBench. These methods are Rejex (Perini & Davis, 2023), ADGym (Jiang et al., 2023) and DTE (Livernoche et al., 2024). We also compare the distribution of our generated data against that of data generated by oversampling techniques such as SMOTE (Chawla et al., 2002), Borderline-SMOTE (Han et al., 2005), ADASYN (He et al., 2008), SVM-SMOTE (Nguyen et al., 2011), CBO (Xu et al., 2021), Oversampling GAN (Nazari & Branco, 2021) and VAE-Geometry (Chadebec et al., 2023). For each method, we use the best-performing hyperparameters as provided in its original paper.

| Methods | DTE | | Rejex | | ADGym | | Swift Hydra (Single) | | Swift Hydra (MoME) | |
|---|---|---|---|---|---|---|---|---|---|---|
| | AUCROC | TIF | AUCROC | TIF | AUCROC | TIF | AUCROC | TIF | AUCROC | TIF |
| Train/Test Ratio (40/60%) | 0.82 | 4.02 | 0.78 | 3.89 | 0.86 | 6.12 | 0.91 | 13.11 | **0.93** | 4.01 |
| Train/Test Ratio (30/70%) | 0.80 | 4.13 | 0.77 | 4.09 | 0.82 | 7.03 | 0.90 | 14.38 | **0.91** | 4.79 |
| Train/Test Ratio (20/80%) | 0.79 | 4.31 | 0.76 | 4.22 | 0.79 | 8.17 | 0.87 | 16.13 | **0.90** | 5.22 |
| Train/Test Ratio (10/90%) | 0.78 | 4.42 | 0.74 | 4.39 | 0.77 | 9.14 | 0.86 | 18.52 | **0.87** | 5.84 |
| | | | | | | | | | TIF = Total Inference Time (Seconds) | |

Table 1: The performance of Swift Hydra and the baselines on the ADBench is evaluated based on two criteria: AUC-ROC and total inference time (TIF). Here, the AUC-ROC is the average calculated across all 57 datasets, while the TIF represents the total time the model takes to predict all data points across all datasets. We vary the train/test ratios to illustrate how the size of the training data impacts the performance. The best AUC-ROC values are highlighted.

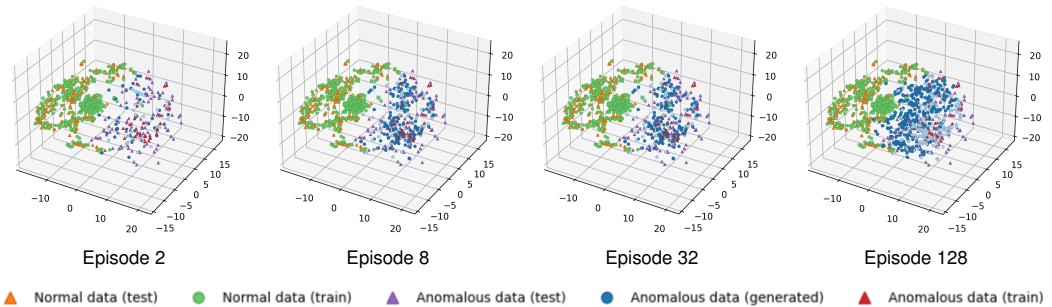

Figure 2: The distribution of the evolving training set $\mathcal{D}_e^{train}$ and test set $\mathcal{D}^{test}$ is visualized using the Cardiotocography dataset, one of the 57 datasets from ADBench, generated by Swift Hydra in each episode. The data points are dimensionally reduced using T-SNE (van der Maaten & Hinton, 2008). Note that the light blue points represent the generated datapoints from previous episodes, providing insight into the trend of generating anomalous data across episodes.

**AUC-ROC Evaluation.** As shown in Table 1, the average AUC-ROC scores show that both versions of Swift Hydra - single large detector and MoME - consistently outperform other state-of-the-art (SOTA) methods with respect to various training sizes (i.e., 40%, 30%, 20%, and 10% of the whole dataset). Notably, with only 10% of the dataset, Swift Hydra outperforms DTE in the semi-supervised setting and Rejex in the unsupervised setting. This demonstrates that our RL algorithm can train a generative model to synthesize effective anomalies that can later be used to train a high-performing detection model. We refer readers to Appendix **C.2** for a comparative analysis with more SOTA detection methods and oversampling techniques, and Appendix **C.3** for a toy example to illustrate the generalization ability of Swift Hydra. Appendix **C.4** presents a series of ablation studies evaluating the impact of the Self-Reinforcing Module, the effectiveness of the probabilistic cluster assignments (as described in Equation 6), and the influence of the KL term and the reconstruction term in Equation 2 on the AUC-ROC of Swift Hydra.

**Inference Time Evaluation.** In terms of total inference time across 57 datasets, Table 1 shows that Rejex has the shortest time, which is expected as it relies on conventional lazy learning methods such as Isolation Forest. DTE, which is based on a diffusion model, requires only a few steps to reconstruct backward and determine whether a sample is anomalous, resulting in relatively short inference times. Although ADGym optimally selects which ML models to use for each dataset, the experiment shows that its overall prediction time is still relatively high compared to that of Swift Hydra (MoME). Swift Hydra (Single) achieves high AUC-ROC scores; nevertheless, its prediction time is significantly longer because a single large model is designed to capture the entire diverse dataset generated by the Self-Reinforcing Generative Module. In contrast, Swift Hydra (MoME) not only attains the best AUC-ROC scores but also has efficient prediction times that are comparable to DTE with respect to the training sizes of 40% and 30%. Overall, Swift Hydra (MoME) offers the best balance between AUC-ROC performance and inference time among the tested methods.

**Generated Data Distribution.** We visualize the distribution of data generated over time by our Self-Reinforcing Generative Module in Figure 2. Initially, the model explores a broad spectrum of widely dispersed anomalous data points. As the episodes progress, a discernible pattern emerges: the generated anamalous points increasingly cluster towards the boundary that separates normal from anomalous data. In fact, this transitional zone at the boundary highlights the anomalies that are not easily distinguishable from normal data points. Hence, this dynamic progression shows that our generative method significantly enriches the diversity of anomalous data points while simultaneously pushing for the most challenging anomalies, thus strengthening the detector's generalization ability.

Figure 3 shows a comparative analysis on the generated data distribution of our method and that of other oversampling methods. As can be seen, methods like SMOTE, Borderline-SMOTE, SVM-SMOTE, ADASYN, and CBO only generate data points within the boundary of the anomalous data in the training set, while our approach allows data points to be generated beyond these boundaries. This enables our method to potentially generate anomalous data points that can cover the distribution of the test set. Although VAE-Geometry and Oversampling GAN also explore beyond the

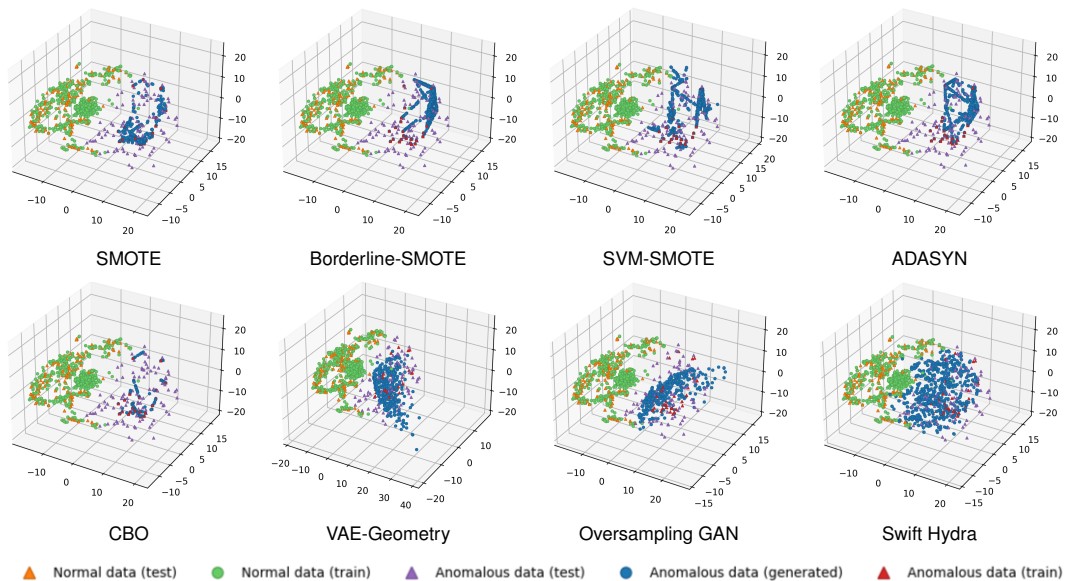

Figure 3: The distribution of the evolving training set $\mathcal{D}_e^{train}$ and test set $\mathcal{D}^{test}$ is visualized using the Cardiotocography dataset, one of the 57 datasets from ADBench, generated by the oversampling methods in our baselines. The data points are dimensionally reduced using T-SNE (van der Maaten & Hinton, 2008).

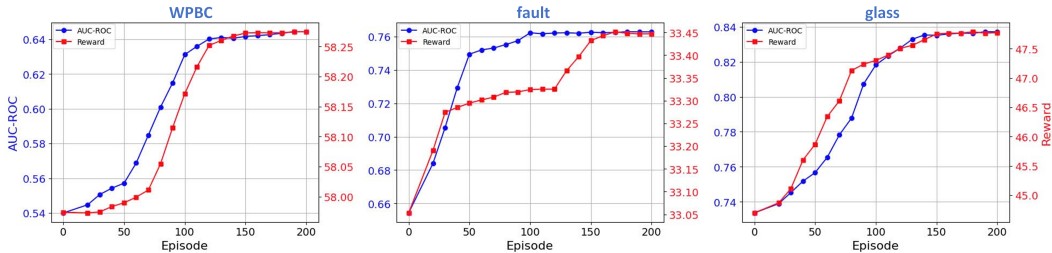

Figure 4: The performance of the RL-Agent is represented by the reward (right y-axis), while the performance of the Mamba-based Detector (single model) is measured by AUC-ROC (left y-axis). Both metrics are plotted against the number of episodes (x-axis) across three challenging datasets from ADBench. Note that both reward and AUC-ROC are averaged over multiple roll-outs.

boundary, they have limitations. Oversampling GAN suffers from model collapse (Salimans et al., 2016; Hassanaly et al., 2022): during the early training steps, if it finds one data point that is very good at fooling the detector, it will only focus on generating samples around that point in subsequent steps. VAE-Geometry performs better as it generates more diverse data points. However, it is highly sensitive to hyperparameters to learn the data manifold correctly, hence, it is less effective compared to our method. Both Figures 2 and 3 demonstrate that data generated by Swift Hydra provides comprehensive coverage over the range of anomalous data in the test set, even though no knowledge about the test data is provided during training.

**Soundness of the Reward Function.** Figure 4 shows that the reward trend (average of multiple roll-outs) closely follows the increase in AUC-ROC. This demonstrates the soundness of our reward function, as the RL agent optimizes the reward function, either by predicting actions itself or using feasible actions as discussed in Section **3.1.2**, leading to the maximization of AUC-ROC in the test set. Interestingly, even though the RL agent receives no feedback on how it performs on the test set (no knowledge about the test set is provided during training), it manages to increase the AUC-ROC over time. This suggests that our reward function helps improve the detector's generalization ability.

## 5 RELATED WORK

**Anomaly Detection.** Due to the high cost and difficulty of data annotation, most recent anomaly detection (AD) research has focused on unsupervised methods with various data distribution assumptions (Aggarwal, 2017; Liu et al., 2008; Zong et al., 2018; Li et al., 2020; 2022; Xu et al., 2022). Common approaches like GAN-based (Donahue et al., 2017; Schlegl et al., 2017), self-supervised (Hojjati et al., 2022; Sehwag et al., 2021; Georgescu et al., 2021; Li et al., 2021), and one-class classification (Shen et al., 2020; Hu et al., 2020) typically rely solely on normal data for training, making it difficult to identify anomalies due to the absence of true anomaly patterns. Reconstruction-based methods (An & Cho, 2015; Xu et al., 2022) use anomaly reconstruction loss to detect outliers but are often unreliable as neural networks can memorize and generalize even with a few samples of anomalies. More recent supervised or weakly-supervised methods (Pang et al., 2018b; 2019a;d; Ruff et al., 2020; Zhou et al., 2021) treat anomalies as negative samples to improve sensitivity, but they risk overfitting and heavily depend on the diversity and quality of the dataset.

Advanced methods like ADGym (Jiang et al., 2023) have improved anomaly detection through optimized data processing, augmentation, network design, and training, but they may fail if settings do not align with the target domain. Learning to Reject (Perini & Davis, 2023) uses uncertainty scores to reject rather than forcibly predict uncertain samples; however, it often rejects data near the normal-anomaly boundary, reducing detection performance. DTE (Livernoche et al., 2024) leverages diffusion models to estimate posterior densities, but the decoder can still memorize and reconstruct anomalies, complicating reliable scoring. AnomalyClip (Zhou et al., 2024) captures general anomalies in images using object-agnostic text prompts but is limited to image-based tasks.

**Oversampling-based techniques.** Traditional oversampling techniques tackle imbalanced data by generating synthetic samples. SMOTE (Chawla et al., 2002) interpolates between minority points to increase diversity but does not focus on challenging samples. Variations like CBO (Xu et al., 2021), Borderline-SMOTE (Han et al., 2005), and SVM-SMOTE (Nguyen et al., 2011) generate samples near boundaries to improve representation but risk introducing noise and overfitting in complex distributions. ADASYN (He et al., 2008) targets harder instances for sample generation, enhancing performance but potentially causing redundancy if not carefully managed.

Recent techniques like Oversampling GAN (Nazari & Branco, 2021) and VAE-Geometry (Chadebec et al., 2023) use deep learning to generate more generalized samples. Oversampling GAN may suffer from issues like vanishing gradients or model collapse, limiting sample diversity. VAE-Geometry employs a Variational Autoencoder that preserves the geometric structure of the data during augmentation, producing synthetic samples that more accurately reflect the true distribution. However, its accuracy depends on correctly learning the data manifold and is highly sensitive to hyperparameters; failure to capture complex structures can result in inaccurate sample generation.

**RL-Guided Generative AI.** Reinforcement Learning (RL) has been used to guide Generative AI (GenAI) in large language models (LLMs), as seen in "Learning from Human Feedback" (Dubois et al., 2023) and ReST (Gulcehre et al., 2023), enhancing generative capabilities through reward models. The direct use of RL to guide the sample generation process of generative models in anomaly detection remains underexplored, with this approach only recently gaining traction through the ReST framework for LLMs.

## 6 CONCLUSION

We propose Swift Hydra, a framework designed to reinforce a generative model's ability to synthesize anomalies in order to augment anomaly detection models. The framework features an RL agent to guide the training of a C-VAE model that generates diverse and challenging anomalies. We further propose a mechanism to help the RL agent choose an action more efficiently during training. Additionally, due to the diverse nature of the generated dataset, we introduce a Mixture of Mamba Experts to train an efficient anomaly detector, where each expert specializes in capturing specific data clusters. As a result, our model demonstrates strong generalization capabilities and fast inference, as evidenced by experiments conducted on the ADBench benchmark against state-of-the-art anomaly detection models. Our research highlights a promising paradigm of integrating RL and generative AI for advancing anomaly detection. It can also be leveraged for generating and synthesizing data in other application contexts where collecting real data is expensive and scarce.

ACKNOWLEDGMENTS

This work was partially supported by the National Science Foundation under the SaCT program, grant number CNS-1935923, and by the Korea Institute of Energy Technology Evaluation and Planning (KETEP) grant funded by the Korea government (MOTIE) (RS-2023-00303559, Study on developing cyber-physical attack response system and security management system to maximize real-time distributed resource availability).

This work was authored in part by the National Renewable Energy Laboratory, operated by Alliance for Sustainable Energy, LLC, for the U.S. Department of Energy (DOE) under Contract No. DE-AC36-08GO28308. Funding provided by the U.S. Department of Energy Office of Cybersecurity, Energy Security, and Emergency Response (CESER), and by the Laboratory Directed Research and Development (LDRD) Program at NREL. The views expressed in the article do not necessarily represent the views of the DOE or the U.S. Government. The U.S. Government retains and the publisher, by accepting the article for publication, acknowledges that the U.S. Government retains a nonexclusive, paid-up, irrevocable, worldwide license to publish or reproduce the published form of this work, or allow others to do so, for U.S. Government purposes.

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

## A  DETAILS OF SWIFT HYDRA

### A.1  SELF-REINFORCING GENERATIVE MODULE

The algorithm **A.1** describes a self-reinforcing generative module for training a Conditional VAE (C-VAE) based generator, a Mamba detector, and a policy network within a reinforcement learning framework. The input includes the number of episodes $E$ and steps per episode $T$, evolving datasets $\mathcal{D}_e^{train}$, $\mathcal{D}_e^{balance}$, and $\mathcal{D}_{e,\mathrm{anomalous}}^{train}$ for each episode $e$, the generator $\mathcal{F}_\theta = \mathcal{G}_\psi \circ \mathcal{M}_\phi$, and the Mamba detector $\mathcal{W}_\kappa$. The process begins by initializing the policy $\pi_\omega$, the generator $\mathcal{F}_\theta$, and the detector $\mathcal{W}_\kappa$. For each episode $e$, the generator is first trained on the current training dataset $\mathcal{D}_e^{train}$, and the detector is trained using the balanced dataset $\mathcal{D}_e^{balance}$. A trajectory buffer $\mathfrak{B}$ is initialized to store states, actions, and rewards.

During each step $t$ in the episode, the generator produces a latent vector $z$ for an anomalous data point, and the policy network samples an action $a = (\mu, \sigma)$ based on the state $s = (z, \mathcal{D}_e^{train})$. A perturbation $\delta = \sigma \cdot \epsilon + \mu$ is applied to obtain a new latent vector $z'$. If $z'$ is within the supported range, a new sample $x' = \mathcal{M}_\phi(z', y = 1)$ is generated; otherwise, the One-Step to Feasible Action algorithm **2** is used to adjust the action, correcting $z'$ and updating the action. The reward $\mathcal{R}(s, a, e)$ is calculated and stored in $\mathfrak{B}$ along with the state and action, and the generated sample $x'$ is added to a temporary set $\hat{\mathcal{X}}$.

After completing all steps within an episode, the policy $\pi_\omega$ is updated using reinforcement learning techniques—such as Proximal Policy Optimization (PPO) or behavior cloning—based on the collected trajectory $\mathfrak{B}$. The top $l$ samples with the highest rewards from $\hat{\mathcal{X}}$ are then added to the training set $\mathcal{D}_e^{train}$. Subsequently, $\mathcal{D}_e^{balance}$ is refined from $\mathcal{D}_e^{train}$ using the helper function Trim(.), ensuring an equal number of elements in both classes.

To further ensure efficient training and avoid unnecessary iterations, a convergence criterion based on the average reward per episode is integrated into the process. At the end of each episode, the algorithm computes the average reward over the collected trajectory. This average reward is compared against the best observed average reward, and if the improvement does not exceed a specified threshold $\Omega$ for $N$ consecutive episodes, the training loop is terminated early. This mechanism ensures that once the models and policy have stabilized—indicating convergence—further training is halted, thereby preventing overfitting and saving computational resources.

Moreover, if the total number of generated anomalous data combined with the training anomalous data exceeds the total number of normal data in the training set, the Trim(.) function could remove some of the generated data. However, across our experiments on all 57 datasets, we observed that Swift Hydra consistently converges before the anomalous data surpasses the normal data in quantity. Note that in most cases, the number of available anomalies in the training data only accounts for 1%-15% of the entire dataset (see Appendix C.6), representing the primary challenge in anomaly detection. If the total anomalous data were to exceed the normal data, one approach would be to start generating (or collecting) more synthetic (or real, respectively) normal data.

### A.2  ONE-STEP TO FEASIBLE ACTIONS

The algorithm implements a one-step optimization process to adjust a latent variable $z$, aiming to increase the diversity in the evolving dataset $\mathcal{D}_e^{train}$ using Kernel Density Estimation (KDE) and entropy maximization. The process begins with initializing key parameters: the gradient step size $\eta$, which controls the size of updates to $z$; the regularization parameter $\gamma$, which determines the importance of diversity in the optimization; and the number of sampled datapoints $\varsigma$, used to estimate the dataset's distribution.

In each iteration, a KDE model (detailed in Appendix **A.3**) is constructed using the dataset $\mathcal{D}_e^{train}$ to capture its distribution. This model helps estimate the density of the data points within the current dataset. After building the KDE, we sample $\varsigma$ data points from it to approximate the dataset's overall distribution. These sampled points are then used to calculate the entropy $\mathcal{H}(z)$ (explained further in Appendix **A.3**), which quantifies the diversity or uncertainty present in the dataset.

Following this, the algorithm calculates the loss $\mathcal{L}_{\mathrm{pred}}$ based on a reward function (as defined in Equation 3). With the loss computed, the latent variable $z$ is updated through gradient descent.

---

**Algorithm 1:** Self-Reinforcing Generative Module

---

**Input:** $E$ episodes, $h$ steps per episode
Evolving datasets $\mathcal{D}_e^{train}, \mathcal{D}_e^{balance}, \mathcal{D}_{e,anomalous}^{train}$ for each episode $e \in [1, E]$
C-VAE-based Generator $\mathcal{F}_\theta = \mathcal{G}_\psi \circ \mathcal{M}_\phi$
Mamba detector $\mathcal{W}_\kappa$
Convergence parameters: threshold $\Omega$ and maximum episodes $N$ with no improvement
**Output:** Trained models $\mathcal{F}_\theta, \mathcal{W}_\kappa$, policy $\pi_\omega$, and datasets $\mathcal{D}_e^{train}, \mathcal{D}_e^{balance}$
    `// Initialize models, policy, and convergence tracking`
1   Initialize Policy $\pi_\omega$, Generator $\mathcal{F}_\theta$, and Detector $\mathcal{W}_\kappa$
2   Set $best\_avg\_reward \leftarrow -\infty, no\_improvement\_count \leftarrow 0, e \leftarrow 1$
3   **while** $e \leq E$ **and** $no\_improvement\_count \leq N$ **do**
       `// Train VAE model on the current training set`
4      `TrainVAE`$(\mathcal{D}_e^{train}, \mathcal{F}_\theta)$
       `// Train detector model on balanced training data`
5      `TrainDetector`$(\mathcal{D}_e^{balance}, \mathcal{W}_\kappa)$
6      Initialize trajectory $\mathfrak{B} \leftarrow \emptyset$ and new samples set $\hat{\mathcal{X}} \leftarrow \emptyset$
       `// Generate new samples to expand the training dataset`
7      **for** $t = 1$ **to** $h$ **do**
          `// For each` $x \in \mathcal{D}_{e,anomalous}^{train}$`, compute latent representation`
8         $z = \mathcal{F}_\theta(x)$
          `// Sample action based on state` $s = (z, \mathcal{D}_e^{train})$
9         Sample action $a = (\mu, \sigma)$ using policy $\pi_\omega$
          `// Compute perturbation and form new latent vector`
10        $\delta = \sigma \cdot \epsilon + \mu$ where $\epsilon \sim \mathcal{N}(0, I)$
11        Form new latent vector $z' = z + \delta$
12        **if** $z'$ *is within the supported range* **then**
13           $x' = \mathcal{M}_\phi(z', y = 1)$
14        **else**
             `// Adjust latent vector if out of feasible range`
15           $z' = $ `OneStepToFeasibleAction`$(z, \mathcal{D}_e^{train})$
16           $x' = \mathcal{M}_\phi(z', y = 1)$
             `// Update action to reflect the feasible adjustment`
17           $a = (\hat{\delta} = z' - z, \sigma)$
          `// Calculate reward for the current state-action pair`
18         Calculate reward $\mathcal{R}(s, a, e)$
19         Append $(s, a, \mathcal{R}(s, a, e))$ to trajectory $\mathfrak{B}$ and sample $(x', y = 1)$ to $\hat{\mathcal{X}}$
       `// Update policy using a gradient descent method (e.g., PPO)`
20      `TrainPolicy`$(\pi_\omega, \mathfrak{B})$
       `// Add top-`$l$ `highest reward samples to the training set`
21      $\mathcal{D}_e^{train} \leftarrow \mathcal{D}_e^{train} \cup \hat{\mathcal{X}}^{<l}$
       `// Balance the dataset by trimming over-represented classes`
22      $\mathcal{D}_e^{balance} \leftarrow \text{Trim}(\mathcal{D}_e^{train})$
       `// Compute the average reward for episode` $e$
23      $avg\_reward_e \leftarrow \frac{1}{|\mathfrak{B}|} \sum_{(s,a,\mathcal{R}) \in \mathfrak{B}} \mathcal{R}(s, a, e)$
       `// Check convergence criteria based on average reward`
24      **if** $avg\_reward_e > best\_avg\_reward + \Omega$ **then**
25        $best\_avg\_reward \leftarrow avg\_reward_e$
26        $no\_improvement\_count \leftarrow 0$
27      **else**
28        $no\_improvement\_count \leftarrow no\_improvement\_count + 1$
29      $e \leftarrow e + 1$
30   **return** $\mathcal{F}_\theta, \mathcal{W}_\kappa, \pi_\omega, \mathcal{D}_e^{train}, \mathcal{D}_e^{balance}$

---

---

**Algorithm 2:** One-Step To Feasible Action

---

**Input:** Latent variable $z$, Current evolving dataset $\mathcal{D}_e^{train}$
**Output:** New Optimized Latent Variable $z'$

1  Initialize gradient step size $\eta$
2  Initialize regularization parameter $\gamma \in [0, 1]$
3  Initialize number of datapoints for KDE sampling $\varsigma$
4  **for** $i = 0$ **to** $\eta$ **do**
     `// Construct Kernel Density Estimation on `$\mathcal{D}_e^{train}$
5       **KDE** $\leftarrow$ `KernelDensityEstimation`$(\mathcal{D}_e^{train})$
     `// Sampling `$\varsigma$` datapoints from KDE`
6       sampled_z $\leftarrow$ `KDE.sample`$(\varsigma)$
     `// Calculate Entropy `$\mathcal{H}(z)$` based on `$\varsigma$` datapoints and KDE`
          `Probability function`
7       $\mathcal{H}(z) \leftarrow$ `Entropy`(sampled_z, KDE)
     `// Compute prediction loss with entropy regularization`
8       $\mathcal{L}_{\text{pred}} \leftarrow \log \mathcal{W}_\kappa \left( \mathcal{M}_\phi(\boldsymbol{z}_i, y_i = 1) \right) - \gamma^e \cdot \mathcal{H}(z_i)$
     `// Update `$z$` by gradient descent`
9       $z_i \leftarrow z_i - \alpha \cdot \nabla \mathcal{L}_{\text{pred}}(\mathbf{x}, z_i)$
10 $z' \leftarrow z_i$
11 **return** $z'$

---

This adjustment directs $z$ towards minimizing the prediction loss, making it more representative of diverse data that can potentially deceive the detector $\mathcal{W}_\kappa$. Once the optimization is completed, the refined latent variable $z'$ is returned, concluding the One-Step to Feasible Action algorithm.

### A.3 ENTROPY ESTIMATION IN DYNAMIC TRAINING DATASETS

To effectively evaluate the diversity of our current evolving dataset $\mathcal{D}_{e,\text{anomalous}}^{train}$, we measure its entropy using Kernel Density Estimation (KDE) followed by sampling-based entropy estimation. KDE helps us estimate the probability density function $p(x)$ from the empirical data $\mathcal{D}_{e,\text{anomalous}}^{train}$. The formula for KDE is:

$$\hat{p}(x) = \frac{1}{nh} \sum_{i=1}^{n} \mathcal{K} \left( \frac{x - x_i}{h} \right)$$

Here, $\hat{p}(x)$ is the estimated probability density at point $x$, $n$ is the total number of points in $\mathcal{D}_{e,\text{anomalous}}^{train}$, $h$ is the bandwidth, and $\mathcal{K}$ is the kernel function. This function, a probability density itself, weights the data points around $x$. For our analysis, we use the Gaussian kernel due to its smooth properties and infinite support:

$$\mathcal{K}(u) = \frac{1}{\sqrt{2\pi}} e^{-\frac{u^2}{2}}$$

The choice of bandwidth $h$ significantly affects the estimator's bias and variance. A smaller $h$ leads to a detailed but potentially noisy estimator (risk of overfitting), whereas a larger $h$ may overly smooth the data (risk of underfitting). We can adopt Silverman's rule of thumb for selecting bandwidth with Gaussian kernels:

$$h = 1.06 \sigma n^{-1/5}$$

where $\sigma$ is the standard deviation of the dataset.

After estimating $\hat{p}(x)$ with KDE, calculating the entropy directly from $\mathcal{D}_{e,\text{anomalous}}^{train}$ would be cumbersome and computationally intensive:

$$\mathcal{H}(\mathcal{D}_{e,\text{anomalous}}^{train}) = -\int \hat{p}(x) \log \hat{p}(x)\, dx$$

Instead, we employ Monte Carlo Sampling to select $\varsigma$ data points $x_j$ from this estimated distribution and approximate the entropy using these samples:

$$\mathcal{H}(\mathcal{D}_{e,\text{anomalous}}^{train}) \approx -\frac{1}{\varsigma} \sum_{j=1}^{\varsigma} \log \hat{p}(x_j)$$

Here, $x_j$ are the samples drawn from $\hat{p}(x)$, and $\log \hat{p}(x_j)$ is the natural logarithm of the estimated density at each sampled point. We calculate the average of these logarithms across all $\varsigma$ sampled points to approximate the entropy. This method provides a practical and computationally efficient approach to estimate the entropy, reflecting the diversity and uncertainty of the dataset $\mathcal{D}_{e,\text{anomalous}}^{train}$.

## B  THEOREMS AND PROOFS

### B.1  REWARD ESTIMATION CONSISTENCY

**Theorem 1** If the reward function $\mathcal{R}$ is differentiable, $\mathcal{F}_\theta$ is well-converged, and $\hat{z}_i := z_i - \epsilon \cdot \nabla_{z_i}(-\mathcal{R}(\mathcal{M}_\phi(z_i, y_i = 1), e))$ for some small $\epsilon$, then $\mathcal{R}(\hat{x}_i, e) > \mathcal{R}(x_i, e)$, where $\hat{x}_i = \mathcal{M}_\phi(\hat{z}_i, y_i = 1)$.

**Proof.** To prove this theorem, we first prove that for a well-converged $\mathcal{F}_\theta$, $\mathcal{M}_\phi$ is Lipschitz-continuous.

Consider the decoder $\mathcal{M}_\phi : \mathbb{R}^d \to \mathbb{R}^P$ composed of $N$ layers. For $j = 1$ to $N - 1$, each layer computes:
$$h_j = q_j(h_{j-1}) = \text{ReLU}(W_j h_{j-1} + b_j),$$
where $h_0 = z_i \in \mathbb{R}^d$, $W_j \in \mathbb{R}^{d_j \times d_{j-1}}$, and $b_j \in \mathbb{R}^{d_j}$. The output layer computes:
$$x_i = \mathcal{M}_\phi(z_i) = q_N(h_{N-1}) = W_N h_{N-1} + b_N,$$
with $W_N \in \mathbb{R}^{P \times d_{N-1}}$ and $b_N \in \mathbb{R}^P$.

To prove that $\mathcal{M}_\phi$ is Lipschitz continuous, consider two inputs $z_i, \hat{z}_i \in \mathbb{R}^d$. We aim to show:
$$\|\mathcal{M}_\phi(z_i) - \mathcal{M}_\phi(\hat{z}_i)\| \leq K\|z_i - \hat{z}_i\|,$$
where $K$ is a finite constant.

Starting from the output layer:
$$\begin{aligned}
\|\mathcal{M}_\phi(z_i) - \mathcal{M}_\phi(\hat{z}_i)\| &= \|q_N(h_{N-1}^{(z_i)}) - q_N(h_{N-1}^{(\hat{z}_i)})\| \\
&= \|W_N h_{N-1}^{(z_i)} + b_N - W_N h_{N-1}^{(\hat{z}_i)} - b_N\| \\
&= \|W_N(h_{N-1}^{(z_i)} - h_{N-1}^{(\hat{z}_i)})\| \\
&\leq \|W_N\|_2 \|h_{N-1}^{(z_i)} - h_{N-1}^{(\hat{z}_i)}\|,
\end{aligned}$$
where $\|W_N\|_2$ denotes the spectral norm of $W_N$.

For each hidden layer $j = N - 1$ down to 1:
$$\begin{aligned}
\|h_j^{(z_i)} - h_j^{(\hat{z}_i)}\| &= \|q_j(h_{j-1}^{(z_i)}) - q_j(h_{j-1}^{(\hat{z}_i)})\| \\
&= \|\text{ReLU}(W_j h_{j-1}^{(z_i)} + b_j) - \text{ReLU}(W_j h_{j-1}^{(\hat{z}_i)} + b_j)\| \\
&\leq \|W_j h_{j-1}^{(z_i)} - W_j h_{j-1}^{(\hat{z}_i)}\| \quad \text{(since ReLU is 1-Lipschitz)} \\
&\leq \|W_j\|_2 \|h_{j-1}^{(z_i)} - h_{j-1}^{(\hat{z}_i)}\|.
\end{aligned}$$

By recursively applying these inequalities, we obtain:

$$\|h_j^{(z_i)} - h_j^{(\hat{z}_i)}\| \leq \left( \prod_{k=1}^{j} \|W_{N-k+1}\|_2 \right) \|h_0^{(z_i)} - h_0^{(\hat{z}_i)}\| = \left( \prod_{k=1}^{j} \|W_{N-k+1}\|_2 \right) \|z_i - \hat{z}_i\|.$$

At the output layer:

$$\|\mathcal{M}_\phi(z_i) - \mathcal{M}_\phi(\hat{z}_i)\| \leq \|W_N\|_2 \|h_{N-1}^{(z_i)} - h_{N-1}^{(\hat{z}_i)}\|.$$

Substituting the recursive bound:

$$\|\mathcal{M}_\phi(z_i) - \mathcal{M}_\phi(\hat{z}_i)\| \leq \left( \prod_{j=1}^{N} \|W_j\|_2 \right) \|z_i - \hat{z}_i\|.$$

Define $K = \prod_{j=1}^{N} \|W_j\|_2$. To ensure $K$ is finite, we enforce bounds (Layer Normalization) on the spectral norms: $\|W_j\|_2 \leq s_j$, where $s_j$ are finite constants. Then:

$$K \leq \prod_{j=1}^{N} s_j.$$

If we choose $s_j = s \leq 1$ for all $j$, then $K \leq s^N \leq 1$, which is finite. Therefore, $\mathcal{M}_\phi$ is Lipschitz continuous with Lipschitz constant $K$, satisfying:

$$\|\mathcal{M}_\phi(z_i) - \mathcal{M}_\phi(\hat{z}_i)\| \leq K \|z_i - \hat{z}_i\|.$$

Thus, we finished proving that $\mathcal{M}_\phi$ is Lipschitz-continuous. Given $\mathcal{F}_\theta$ is well converged, With $\mathcal{M}_\phi$ being Lipschitz continuous and differentiable, a small learning rate $\epsilon$ induces a small change in latent vector $z_i$ which results in a small change in the data point $x_i$ reconstructed by C-VAE. We can use a first-order Taylor expansion for small $\Delta z_i = \hat{z}_i - z_i$:

$$\hat{x}_i = \mathcal{M}_\phi(\hat{z}_i) \approx \mathcal{M}_\phi(z_i) + J_{\mathcal{M}_\phi}(z_i) \cdot \Delta z_i$$

where $J_{\mathcal{M}_\phi}(z_i)$ is the Jacobian matrix of $\mathcal{M}_\phi$ at $z_i$.

From the update rule:

$$\Delta z_i = \hat{z}_i - z_i = \epsilon \cdot \nabla_{z_i} \mathcal{R}(x_i, e)$$

Thus, the change in $x_i$ is:

$$\hat{x}_i - x_i \approx J_{\mathcal{M}_\phi}(z_i) \cdot \Delta z_i = \epsilon \cdot J_{\mathcal{M}_\phi}(z_i) \cdot \nabla_{z_i} \mathcal{R}(x_i, e)$$

Since $x_i = \mathcal{M}_\phi(z_i)$, by the chain rule, we have:

$$\nabla_{z_i} \mathcal{R}(x_i, e) = J_{\mathcal{M}_\phi}^\top(z_i) \cdot \nabla_{x_i} \mathcal{R}(x_i, e)$$

Therefore:

$$\hat{x}_i - x_i \approx \epsilon \cdot J_{\mathcal{M}_\phi}(z_i) \cdot J_{\mathcal{M}_\phi}^\top(z_i) \cdot \nabla_{x_i} \mathcal{R}(x_i, e)$$

Let $F = J_{\mathcal{M}_\phi}(z_i) \cdot J_{\mathcal{M}_\phi}^\top(z_i)$, which is a positive semi-definite matrix. Thus:

$$\hat{x}_i - x_i \approx \epsilon \cdot F \cdot \nabla_{x_i} \mathcal{R}(x_i, e)$$

Using a first-order Taylor expansion of $\mathcal{R}$ around $x_i$ :

$$\Delta\mathcal{R} = \mathcal{R}\left(\hat{x}_i, e\right) - \mathcal{R}\left(x_i, e\right) \approx \nabla_{x_i}\mathcal{R}\left(x_i, e\right)^\top \left(\hat{x}_i - x_i\right)$$

Substituting $\hat{x}_i - x_i$ :

$$\Delta\mathcal{R} \approx \epsilon \cdot \nabla_{x_i}\mathcal{R}\left(x_i, e\right)^\top F \cdot \nabla_{x_i}\mathcal{R}\left(x_i, e\right)$$

Since $F$ is positive semi-definite and $\epsilon > 0$ :

$$\Delta\mathcal{R} \geq 0$$

More specifically, $\Delta\mathcal{R} = 0$ if and only if $\nabla_{x_i}\mathcal{R}\left(x_i, e\right) = 0$. Otherwise, $\Delta\mathcal{R} > 0$. Therefore, under the given conditions and for a sufficiently small $\epsilon$ :

$$\mathcal{R}\left(\hat{x}_i, e\right) > \mathcal{R}\left(x_i, e\right)$$

This completes the proof.

### B.2 ONE DETECTOR INEFFECTIVELY HANDLES EVOLVING BALANCE DATA

**Theorem 2.** Suppose a feature space $\mathfrak{X} \subset \mathbb{R}^P$ contains $U_n$ normal clusters and $U_a$ anomalous clusters, where each cluster $u$-th $\in [U_n + U_a]$ is modeled as a Gaussian distribution $\mathcal{N}(\boldsymbol{\mu}_u, \sigma^2\mathbf{I}_P)$. Let $\mathcal{V}_{\text{cluster}}$ be the cluster's volume and $\Lambda$ be the total overlapping volume between normal and anomalous clusters, where the number of anomalous data points is equal to the number of normal data points, the training loss $\mathcal{L}_{\text{train}}\left(\mathcal{W}_\kappa\right)$ is lower bounded by $\frac{1}{4} \cdot \frac{\Lambda}{U_a \cdot \mathcal{V}_{\text{cluster}} - \frac{\Lambda}{2}}$ in a case of linear $\mathcal{W}_\kappa$.

**Proof.** Consider the feature space $\mathfrak{X} \subset \mathbb{R}^P$ with $U_n$ normal Gaussian clusters and $U_a$ anomalous Gaussian clusters, each modeled as $\mathcal{N}(\boldsymbol{\mu}_u, \sigma^2\mathbf{I}_P)$. The volume of each cluster is:

$$\mathcal{V}_{\text{cluster}} = \frac{\pi^{P/2}(3\sigma)^P}{\Gamma\left(\frac{P}{2}+1\right)},$$

where $3\sigma$ represents the radius covering 99.7% of the data points in a cluster. The total volume occupied by the normal clusters is:

$$\mathcal{V}_{\text{total\_normal}} = U_n \cdot \mathcal{V}_{\text{cluster}},$$

and the total volume occupied by the anomalous clusters is:

$$\mathcal{V}_{\text{total\_anomalous}} = U_a \cdot \mathcal{V}_{\text{cluster}}.$$

The clusters overlap in certain regions, resulting in a total overlapping volume $\Lambda$ between normal and anomalous clusters. Under our assumption, this overlapping region contains 50% of $\Lambda$, i.e., $\frac{\Lambda}{2}$ normal data and 50% of $\Lambda$, i.e., $\frac{\Lambda}{2}$ anomalous data.

Note that since the datapoints are in the overlapping area, we assume that the unique features are negligible while noise features from negative class are dominant. The unique volumes of the normal and anomalous clusters, excluding the overlapping regions, are:

$$\begin{aligned}
\mathcal{V}_{\text{unique\_normal}} &= \mathcal{V}_{\text{total\_normal}} - \frac{\Lambda}{2} \\
&= U_n \cdot \mathcal{V}_{\text{cluster}} - \frac{\Lambda}{2}
\end{aligned}$$

$$\mathcal{V}_{\text{unique\_anomalous}} = \mathcal{V}_{\text{total\_anomalous}} - \frac{\Lambda}{2}$$
$$= U_a \cdot \mathcal{V}_{\text{cluster}} - \frac{\Lambda}{2}$$

For simplicity, we assume that the detector $\mathcal{W}_{\kappa}$ constructs decision boundaries around the normal clusters. Specifically, the detector aims to enclose the normal clusters within its decision regions to classify them as normal, while any data points outside these regions are considered anomalous. As indicated in the work of Chen et al. (2022), a single detector focuses on both unique features and noise features, even though unique features are negligible. This means the detector $\mathcal{W}_{\kappa}$ seeks to minimize False Negatives by primarily capturing the normal data based on noise features, while overlooking unique features. As the model size $\chi$ increases, the decision boundaries of the detector can more precisely conform to the normal clusters, potentially leading to overfitting of the normal data. Let $\mathfrak{q}(\chi)$ denote the proportion of the unique normal volume that the detector's decision boundary covers:

$$V_{\text{covered\_normal}} = \mathfrak{q}(\chi) \cdot \mathcal{V}_{\text{unique\_normal}}.$$

The False Negative Rate (FNR), which represents the proportion of normal data not covered by the detector, is:

$$\text{FNR} = 1 - \mathfrak{q}(\chi).$$

Because the detector's decision boundary encloses the normal clusters, it inevitably includes parts of the overlapping regions $\Lambda$. Therefore, the detector inadvertently covers some anomalous data within the overlapping regions, leading to False Positives. The volume of anomalous data incorrectly classified as normal (False Positives) is:

$$V_{\text{FP}} = \mathfrak{q}(\chi) \cdot \frac{\Lambda}{2}$$

The False Positive Rate (FPR), representing the proportion of anomalous data misclassified as normal, is:

$$\text{FPR} = \frac{V_{\text{FP}}}{\mathcal{V}_{\text{unique\_anomalous}}}$$
$$= \frac{\mathfrak{q}(\chi) \cdot \frac{\Lambda}{2}}{U_a \cdot \mathcal{V}_{\text{cluster}} - \frac{\Lambda}{2}}$$

Assuming equal prior probabilities for normal and anomalous data, the expected error $L(\mathcal{W}_{\kappa})$ is:

$$L(\mathcal{W}_{\kappa}) = \frac{1}{2} \cdot \text{FNR} + \frac{1}{2} \cdot \text{FPR}$$
$$= \frac{1}{2} \left( 1 - \mathfrak{q}(\chi) + \frac{\mathfrak{q}(\chi) \cdot \frac{\Lambda}{2}}{U_a \cdot \mathcal{V}_{\text{cluster}} - \frac{\Lambda}{2}} \right)$$
$$= \frac{1}{2} \left( 1 - \mathfrak{q}(\chi) + \frac{\mathfrak{q}(\chi) \cdot \Lambda}{2 \left( U_a \cdot \mathcal{V}_{\text{cluster}} - \frac{\Lambda}{2} \right)} \right)$$

Our goal is to find the minimum expected error $\mathcal{L}_{train}(\mathcal{W}_{\kappa})$. To achieve this, we consider how $L(\mathcal{W}_{\kappa})$ varies with $\mathfrak{q}(\chi)$. Since the detector aims to maximize coverage of the normal clusters (i.e., maximize $\mathfrak{q}(\chi)$) to minimize False Negatives, we consider the case where $\mathfrak{q}(\chi) = 1$, corresponding to the detector fully covering the unique normal volume.

Substituting $\mathfrak{q}(\chi) = 1$ into $L(\mathcal{W}_{\kappa})$, we get:

$$\mathcal{L}_{\text{train}}\left(\mathcal{W}_\kappa\right) \geq \frac{1}{2}\left(1 - 1 + \frac{1 \cdot \frac{\Lambda}{2}}{U_a \cdot \mathcal{V}_{\text{cluster}} - \frac{\Lambda}{2}}\right)$$

$$= \frac{1}{2} \cdot \frac{\Lambda}{2\left(U_a \cdot \mathcal{V}_{\text{cluster}} - \frac{\Lambda}{2}\right)}$$

$$= \frac{1}{2} \cdot \frac{\Lambda}{2U_a \cdot \mathcal{V}_{\text{cluster}} - \Lambda}$$

$$= \frac{1}{4} \cdot \frac{\Lambda}{U_a \cdot \mathcal{V}_{\text{cluster}} - \frac{\Lambda}{2}}$$

This expression represents the minimum expected error achievable by any detector that constructs decision boundaries around the normal clusters. Due to the overlapping volume $\Lambda$ between the normal and anomalous clusters, there is an inherent lower bound on the expected error $\mathcal{L}_{train}(\mathcal{W}_\kappa)$ that any detector of this type can achieve. The detector cannot reduce the error below this bound because, in maximizing coverage of the normal data to minimize False Negatives, it inevitably includes portions of the overlapping anomalous data, resulting in unavoidable False Positives.

### B.3 MoME EFFECTIVELY HANDLES EVOLVING BALANCE DATA

**Theorem 3.** Let $\mathcal{L}_{test}(\mathfrak{F})$ and $\mathcal{L}_{test}(\mathcal{W}_\kappa)$ represent the expected error on the test set for the Mixture of Mamba Experts (MoME) model and a single detector, respectively. For any value of $\Lambda$, employing MoME with $\{f_1, f_2, \ldots, f_M\}$ guarantees that the minimum expected error on the training set is $\mathcal{L}_{train}(\mathfrak{F}) = 0$ and the expected error on the test set satisfies $\mathcal{L}_{test}(\mathfrak{F}) \leq \mathcal{L}_{test}(\mathcal{W}_\kappa)$.

**Proof.** Similar to the setting in Theorem 2, we also consider the feature space $\mathfrak{X} \subset \mathbb{R}^P$ with $U_n$ normal clusters and $U_a$ anomalous clusters, each modeled as Gaussian distributions $\mathcal{N}(\boldsymbol{\mu}_u, \sigma^2 \mathbf{I}_P)$. Each cluster occupies a volume $\mathcal{V}_{\text{cluster}} = \frac{\pi^{P/2}(3\sigma)^P}{\Gamma\left(\frac{P}{2}+1\right)}$ and the total overlapping volume between normal and anomalous clusters is $\Lambda$. Again, since the data points are in the overlapping area, we assume that unique features are negligible while noise features from the negative class are dominant.

Recall that, we decompose the balanced dataset into clusters $\{\mathcal{C}_1, \mathcal{C}_2, \ldots, \mathcal{C}_U\}$, where $U = U_n + U_a$ is determined using the elbow method. After that, we train a set of experts $\{f_1, f_2, \ldots, f_M\}$, each acting as an expert for specific data clusters, following the top $k$ gated Mixture-of-Experts approach. Let $\mathcal{D}_e^{train}$ be the training dataset. Each data point $x \in \mathcal{D}_e^{train}$ belongs to a cluster $\mathcal{C}_u$ and has a true label $y(x)$:

$$y(x) = \begin{cases} -1, & \text{if } x \in \text{ a normal cluster,} \\ +1, & \text{if } x \in \text{ an anomalous cluster.} \end{cases}$$

In the overlapping regions $\Lambda$, due to probabilistic assignment and expert overspecification, we assume each cluster $\mathcal{C}_u$ has at least one specialized expert $f_m$ predicting $y(x)$ using both unique and noise features, similar to $\mathcal{W}_k$. The gating network $\lambda(x, \aleph_g, \aleph_{\text{noise}})$ minimizes classification loss but primarily focuses on unique features, assigning higher weights to the appropriate experts. These properties of feature capturing have been highlighted by Chen et al. (2022). For each $x \in \mathcal{D}_e^{train}$, the MoE $\mathfrak{F}$'s output is given by:

$$\mathfrak{F}(x, \aleph_g, \aleph_{\text{noise}}, \mathbf{W}) = \sum_{m \in \mathfrak{T}_x} \lambda_m(x, \aleph_g, \aleph_{\text{noise}}) f_m(x; \mathbf{W}).$$

Primarily based on unique features, the gating network learns to assign significant weights to the expert(s) that correctly classify $x$, ensuring that the model output $\mathfrak{F}(x)$ matches the true label $y(x)$. Thus, the expected error on the training set is:

$$\mathcal{L}_{\text{train}}(\mathfrak{F}) = \frac{1}{|\mathcal{D}_e^{train}|} \sum_{x \in \mathcal{D}_e^{train}} \mathbf{1}_{\mathfrak{F}(x) \neq y(x)} = 0,$$

where $\mathbf{1}_{\mathfrak{F}(x) \neq y(x)}$ is an indicator function that equals 1 if $\mathfrak{F}(x) \neq y(x)$ and 0 otherwise. This completes the proof of the expected error of $\mathfrak{F}$ on the training set.

For the test set, we aim to show that: $\mathcal{L}_{test}(\mathfrak{F}) \leq \mathcal{L}_{test}(\mathcal{W}_\kappa)$

Let's consider the test dataset $\mathcal{D}^{\text{test}}$, drawn from the same distribution as the training dataset $\mathcal{D}_e^{\text{train}}$. If the data points in the test set lie completely outside $U$ clusters from the training set, both $\mathfrak{F}$ and $\mathcal{W}$ will fail to make correct predictions. This is because we are assuming that each classifier forms a decision boundary that tightly fits the cluster it captures. Any data point lying outside these decision boundaries is considered negative for the class corresponding to that cluster. Therefore, without loss of generality, we only need to compare the errors of the two models within the region of the $U$ clusters.

Each data point $x \in \mathcal{D}^{\text{test}}$ belongs to one of the clusters $\mathcal{C}_u$ and has a true label $y(x)$ as defined earlier. Suppose input $x$ lying in a non-overlapping region $\mathcal{D}_{\tilde{\Lambda}}$, the expert $f_m$ specialized in cluster $\mathcal{C}_u$ has learned to classify data points from $\mathcal{C}_u$ correctly. The gating network $\lambda\left(x, \aleph_g, \aleph_{\text{noise}}\right)$ effectively routes $x$ to the correct expert $f_m$, resulting in the MoE model predicting $y(x)$ accurately. Therefore, for these non-overlapping regions, the MoE output is:

$$\mathfrak{F}\left(x, \aleph_g, \aleph_{\text{noise}}, \mathbf{W}\right) = y(x), \quad \forall x \in \mathcal{C}_u \text{ and } x \in \mathcal{D}_{\tilde{\Lambda}}$$
$$\implies \mathcal{L}_{\tilde{\Lambda}}(\mathfrak{F}) = 0$$

Similarly, the single detector $\mathcal{W}_\kappa$, having been trained on the entire dataset, can also correctly predict $y(x)$ in these non-overlapping regions since the classes are well-separated. Thus, the expected error in this region is negligible:

$$\mathcal{L}_{\tilde{\Lambda}}(\mathcal{W}_\kappa) = \frac{1}{|\mathcal{D}_{\tilde{\Lambda}}|} \sum_{x \in \mathcal{D}_{\tilde{\Lambda}}} \mathbf{1}_{\mathcal{W}_\kappa(x) \neq y(x)}$$
$$= \mathcal{L}_{\tilde{\Lambda}}(\mathfrak{F})$$
$$= 0$$

where $\mathbf{1}_{\mathcal{W}_\kappa(x) \neq y(x)}$ is an indicator function that equals 1 if $\mathcal{W}_\kappa(x) \neq y(x)$ and 0 otherwise.

Now, consider the overlapping region $\Lambda$. Assume an equal number of anomalous and normal data points within $\Lambda$, with their features being significantly similar. The experts, specialized in their respective clusters, capture cluster-specific patterns even in these overlapping areas. The gating network $\lambda\left(x, \aleph_g, \aleph_{\text{noise}}\right)$, trained to minimize overall classification loss based mainly on unique features, assigns higher weights to the correct experts that are more likely to predict the true label $y(x)$. Consequently, $\mathfrak{F}$ correctly classifies $x$ in $\Lambda$ with high probability, quantified as $1 - \varepsilon_\Lambda$, where $\varepsilon_\Lambda$ is the MoE's error rate in the overlapping region.

In contrast, the single detector $\mathcal{W}_\kappa$ encounters inherent ambiguity in $\Lambda$ due to the negligible presence of unique features and the dominance of noise features from the negative class within the cluster it aims to capture. Furthermore, with an equal number of normal and anomalous data points assumed in this region, the misclassification probability becomes:

$$\mathcal{L}_{\text{overlap}}\left(\mathcal{W}_\kappa\right) = \frac{1}{2}$$

Let $p_{\tilde{\Lambda}}$ be the probability that a test point lies in a non-overlapping region $\tilde{\Lambda}$, and $p_\Lambda$ be the probability that it lies in overlapping region $\Lambda$. The expected error of the single detector on the test set is:

$$\mathcal{L}_{\text{test}}\left(\mathcal{W}_\kappa\right) = p_{\tilde{\Lambda}} \times 0 + p_\Lambda \times \frac{1}{2} = \frac{p_\Lambda}{2}$$

For the MoE model, the expected error on the test set is:

$$\mathcal{L}_{\text{test}}\left(\mathfrak{F}\right) = p_{\tilde{\Lambda}} \times 0 + p_\Lambda \times \varepsilon_\Lambda = p_\Lambda \varepsilon_\Lambda$$

In the overlapping region $\Lambda$, the worst-case scenario for $\mathfrak{F}$ occurs when it fails to route the input to the correct expert, resulting in a maximum error of $\frac{1}{2}$. However, if some test points are identical or very similar to the training points, the routing network is more likely to direct these inputs to the correct expert, as it primarily focuses on unique features. On the other hand, the single detector $\mathcal{W}_\kappa$ considers both unique and noise features, with noise features dominating. Therefore, we have $\varepsilon_\Lambda \leq \frac{1}{2}$, leading to the conclusion:

$$
\begin{aligned}
\mathcal{L}_{\text{test}}\left(\mathfrak{F}\right) &= p_\Lambda \varepsilon_\Lambda \\
&\leq \frac{p_\Lambda}{2} \\
&\leq \mathcal{L}_{\text{test}}\left(\mathcal{W}_\kappa\right).
\end{aligned}
$$

This completes the proof.

## C  MORE EXPERIMENTS

### C.1  SWIFT HYDRA SETTINGS

#### C.1.1  HYPERPARAMETERS

| Hyperparameter | Value |
|---|---|
| Learning rate for C-VAE Model | 0.003 |
| Learning rate for Mamba Model | 0.001 |
| Learning rate for Generator | 0.0001 |
| Total epoch for Detector Model | 600 |
| Total epoch for Generator Model | 500 |
| Optimizer | Adam (Kingma & Ba, 2015) |
| Number of steps per episode | 500 |
| Number of episodes | 200 |
| Minibatch size | 256 |
| Discount factor $\gamma$ | 0.95 |
| Activation function for Mamba Model | LeakyReLU |
| Layer Depth for Mamba Model | 2 |
| Activation function for C-VAE Model | ReLU |
| Bandwidth | 0.5 |
| Weight KL | 0.55 |
| Number of experts | 20 |
| Top $k$ experts | 2 |
| Detection threshold | 0.2 |
| Sampling from a KDE | 300 |
| Policy Training | Proximal Policy Optimization |

Table 2: Hyperparameters for Swift Hydra

In this section, we present the hyperparameters selected for Swift Hydra, as shown in Table 2, and explain the rationale behind each choice. These hyperparameters are carefully designed to balance model performance, training stability, and computational efficiency.

The learning rates for different models are chosen based on their complexity and training dynamics. The C-VAE model uses a relatively high learning rate of 0.003 to promote faster convergence during training. In contrast, the single Mamba-based detector is over-specified in terms of the number of parameters to effectively capture the data generated by the C-VAE, stored in $\mathcal{D}_e^{balance}$. To ensure stability during its optimization, a lower learning rate of 0.001 is set, considering its sensitivity to parameter updates. The Generator model, which is part of a more delicate generative process, has an even smaller learning rate of 0.0001 to prevent large updates that could destabilize training.

The number of steps per episode (500) and the total number of episodes (200) are chosen to allow the model to generate a total of 100,000 datapoints (200 * 500) across all episodes. This quantity

is sufficient to augment any imbalanced dataset within ADBench. The minibatch size of 256 is selected to strike a balance between training stability and computational efficiency, ensuring enough data is processed per update without causing excessive memory usage.

The discount factor $\gamma = 0.95$ is set to ensure that, in the initial phase, the RL agent not only focuses on generating datapoints to deceive $\mathcal{W}_\kappa$ but also actively explores its surrounding environment. As training progresses, the entropy term in the reward function gradually diminishes due to $\gamma$, the model increasingly concentrates on generating points specifically aimed at deceiving the detector $\mathcal{W}_\kappa$.

For activation functions, LeakyReLU is used in the Mamba Model to address the "dying ReLU" problem, allowing the model to handle negative inputs more effectively, while ReLU is used in the VAE model to facilitate faster training.

Recall the fact that, we use Kernel Density Estimation (KDE) to learn the feature distribution of the dataset $\mathcal{D}_e^{train}$ in our framework. An important hyperparameter in this process is the bandwidth $h$, which controls the smoothness of the probability density function, thereby balancing bias and variance. While $h$ can be determined using Silverman's rule of thumb, as described in Appendix **A.3**, for our experiments, we opted to use the well-known bandwidth parameter of 0.5 for KDE. Despite this simple choice, we still achieved the desired results.

The weight of the KL divergence in the VAE is set to 0.55. This is carefully selected to balance reconstruction accuracy and latent space regularization, preventing overfitting while maintaining meaningful latent representations. For a more detailed explanation of how the trade-off between reconstruction loss and KL loss affects the performance of Swift Hydra, we encourage readers to refer to Appendix **C.4**.

To ensure that the model can capture all potential clusters generated, we overspecified the number of experts to 20. This allows the model to avoid missing any clusters. Indeed, the number of clusters found using KMeans+Elbow on datasets in ADBench is usually no more than 10. For selecting the top $k$ in the Mixture of Experts to ensure only $k$ experts are used during inference (thus saving inference time), we set $k = 2$.

Our policy training algorithm is Proximal Policy Optimization (PPO) (Do et al., 2024; Schulman et al., 2017; Do et al., 2021; Ngo et al., 2024; Khoi et al., 2021). In addition, the detection threshold of 0.2 is chosen so that if the model's confidence exceeds this threshold, the predicted datapoint is considered anomalous. Finally, sampling 300 times from KDE provides enough data to accurately model the underlying distributions without incurring excessive computational costs. Overall, these hyperparameters are carefully tuned to enhance model performance, ensure training stability, and optimize computational resources.

### C.1.2 MODEL SIZE AND TRAINING COST

| AI Module Type | | Number of Parameters | Training time per batch | Total training time |
|---|---|---|---|---|
| C-VAE Generator | | 458,907 | 0.0011 | 3.265 |
| Mamba-based Detector (Single) | | 274,542 | 0.0083 | 8.723 |
| Mixture of Mamba Experts Detector (MoME) | Gating Network | 6,164 | 0.0001 | 0.286 |
| | Expert Network | 33,021 | 0.0026 | 1.287 |
| | Mixture of Mamba Experts (20 Experts, topK=2) | 666,584 | 0.0032 | 489.003 |

Table 3: Number of parameters for each component in Swift Hydra, training time per batch, and total training time. The total training time for the C-VAE and the Mamba-based Detector (Single) is measured per episode. For MoME, the two components, Gating Network and Expert Network, are measured over the entire training data and generated data after completing the Self-Reinforcing Module, calculated across all batches in a single epoch. Additionally, we report the total training time of the Mixture of Mamba Experts (20 Experts, topK=2) across all epochs, as shown in the table.

Table 3 presents the number of parameters for each model, including the C-VAE Generator, Mamba-based Detector (Single model), and Mixture of Mamba Experts Detector (MoME). On average, a single expert in MoME has approximately 33,021 parameters. However, with 20 experts and a gating network, the total parameter count for MoME reaches 666,584. In contrast, the Single Mamba-based

Detector contains around 274,542 parameters, while the C-VAE Generator comprises approximately 458,907 parameters.

In addition to reporting the training time per batch alongside the number of parameters, we have also measured the total training time for each stage of our algorithm. These times were recorded on a workstation equipped with two Nvidia RTX 4090 GPUs. Since this is a reinforcement learning framework, the number of episodes required for training heavily depends on the specific problem being addressed. Therefore, we only measure the total training time of the C-VAE Generator and Single Mamba Detector per episode. On the other hand, the total training time for the Mixture of Mamba Experts Detector is measured after completing Phase 1, using all the data generated during that phase.

Our training methodology supports two approaches depending on the hardware configuration: sequential training or parallel training of the experts. In the sequential approach, each expert is trained one at a time, selecting and training on clusters sequentially. This approach is more memory-efficient and requires less GPU VRAM. In contrast, the parallel approach involves all experts selecting their clusters simultaneously and training concurrently. While faster, this method demands significantly more GPU VRAM. The total training time for the Mixture of Mamba Experts (20 Experts, topK=2) across all epochs, as shown in the table, was measured using the parallel training method.

## C.2 SWIFT HYDRA: MORE PERFORMANCE EVALUATION

### C.2.1 BENCHMARKING AGAINST ADDITIONAL DETECTION METHODS ON ADBENCH

We conducted additional experiments to compare our proposed method, Swift Hydra, with several state-of-the-art tabular anomaly detection methods, including the supervised FTTransformer (Gorishniy et al., 2023), the unsupervised ECOD (Li et al., 2022), and the semi-supervised DevNet (Pang et al., 2019b), PReNet (Pang et al., 2019a), DeepSAD (Ruff et al., 2020), and FEAWAD (Zhou et al., 2021). The results is shown in Table 4.

| Method | ECOD | | DevNet | | PReNet | | DeepSAD | | FEAWAD | | FTTransformer | | SwifHydra (MoME) | |
|---|---|---|---|---|---|---|---|---|---|---|---|---|---|---|
| | AUCROC | TIF | AUCROC | TIF | AUCROC | TIF | AUCROC | TIF | AUCROC | TIF | AUCROC | TIF | AUCROC | TIF |
| Train/Test Ratio (40/60%) | 0.80 | 26.00 | 0.82 | 24.08 | 0.85 | 128.71 | 0.84 | 19.02 | 0.86 | 168.62 | 0.89 | 50.66 | **0.93** | **4.01** |
| Train/Test Ratio (30/70%) | 0.78 | 28.32 | 0.81 | 25.54 | 0.84 | 147.74 | 0.83 | 20.53 | 0.82 | 215.26 | 0.84 | 56.23 | **0.91** | **4.79** |
| Train/Test Ratio (20/80%) | 0.77 | 30.04 | 0.80 | 26.00 | 0.83 | 184.29 | 0.82 | 22.75 | 0.80 | 225.00 | 0.80 | 60.98 | **0.90** | **5.22** |
| Train/Test Ratio (10/90%) | 0.75 | 33.36 | 0.79 | 27.90 | 0.82 | 190.39 | 0.81 | 27.58 | 0.79 | 233.13 | 0.77 | 66.23 | **0.87** | **5.84** |
| | | | | | | | | | | | | | **TIF = Total Inference Time (Seconds)** | |

Table 4: The performance of Swift Hydra and other baseline models, including ECOD, DevNet, PReNet, DeepSAD, FEAWAD, and FTTransformer, is evaluated on ADBench using two key criteria: AUC-ROC and Total Inference Time (TIF). The AUC-ROC is computed as the average across all 57 datasets, while the TIF measures the total time required by each model to predict all data points across these datasets. To analyze the impact of training data size on performance, we vary the train/test ratios. The highest AUC-ROC values are highlighted for clarity.

Semi-supervised methods typically require only a small amount of labeled data to achieve AUC-ROC scores comparable to modern unsupervised methods such as DTE, Rejex (as shown in Table 1), and ECOD. However, these methods often come with high computational costs, particularly when applied to large datasets. While PReNet and FEAWAD generally deliver fast prediction times, their performance can significantly slow down on certain datasets within ADBench (e.g., backdoor, celeba, census, donor, fraud), leading to higher overall inference times. This slowdown occurs because some components in these models rely on per-datapoint computations that do not support parallelization, resulting in substantial delays for datasets with a large number of records. The supervised FTTransformer, built on robust backbone architectures like ResNet and Transformer, achieves AUC-ROC scores that are among the highest across all SOTA methods in baselines, nearly matching those of Swift Hydra. However, its performance declines significantly when applied to sparse datasets.

On the other hand, Swift Hydra offers a distinct advantage by generating unseen data to enhance test set coverage. Through its integration with the Mixture of Mamba Experts (MoME), Swift Hydra consistently outperforms other methods in both AUC-ROC and inference time, establishing itself as the most effective and efficient solution for tabular anomaly detection.

## C.2.2 BENCHMARKING AGAINST ADDITIONAL OVERSAMPLING METHODS ON ADBENCH

Figure 5: The average performance of various oversampling methods on the ADBench dataset. The evaluation metric is AUC-ROC

In addition to the previously discussed experiments, we also compare the performance of Swift Hydra against other oversampling methods using the AUC-ROC metric. As shown in Figure 3, our method is capable of generating the most diverse set of anomalous data points. The goal of this experiment is to assess how accurately Swift Hydra (MoME) performs given these diverse generated data points. Figure 5 presents a comparison of our method's performance against various oversampling techniques included in the ADBench baselines. The results clearly demonstrate that Swift Hydra outperforms all other methods, achieving the highest AUC-ROC on the test set, with a 40/60% train/test ratio. This performance is primarily due to Swift Hydra's ability to generate data points that extend beyond the boundaries of the training set.

Specifically, in the dataset illustrated in Figure 3, it is evident that the anomalous data points in the test set lie outside the boundaries of those in the training set. Traditional oversampling methods fail to generate data beyond these boundaries, which naturally results in a lower AUC-ROC on the test set. Deep learning-based oversampling methods, such as VAE-Geometry and Oversampling GAN, can produce more diverse data than traditional techniques. However, Oversampling GAN often encounters a model collapse issue, where it focuses on generating data points that deceive the detector very effectively, leading to an over-concentration of samples. On the other hand, VAE-Geometry relies on the geometric structure of the data for generation, but if the structure is too complex, it struggles to produce sufficiently diverse data points. As a result, both methods ultimately fall short of Swift Hydra in terms of AUC-ROC performance.

## C.3 TOY EXAMPLE: GENERALIZATION ABILITY AND DECISION BOUNDARY ON 2D DATA

We conduct a toy example with 2D data to illustrate the generalization ability and decision boundary of the MoME module. First, we generate a dataset where the normal data follows a sine curve, while the anomalies are randomly distributed around the sine curve. Then, we use the Self-Reinforcing Module to generate additional anomalous data and visualize the boundary of the MoME in Figure 6.

From the visualization, we can observe that the red regions (indicating areas predicted by MoME as containing normal data) fully encapsulate the dark blue points (normal data). The blue regions (indicating areas predicted by MoME as containing anomalous data) completely cover the generated anomalous points (green) and the test set anomalies (orange). An interesting observation is that the green points (generated anomalies) cause the decision boundary to tightly enclose the dark blue points (normal data). This enhances MoME's ability to accurately identify anomalous data in the test set, as these anomalies are positioned farther from the decision boundary. As expected, the model generalized well to the anomalies in the test set.

## C.4 ABLATION STUDY

## C.4.1 IMPACT OF SELF-REINFORCING MODULE ON PERFORMANCE ACROSS DEEP MODELS

We also compare the performance of Swift Hydra with various deep models (as backbone models) using two key metrics: AUC-ROC and Total Inference Time (for predicting the entire ADBench dataset) under two settings: with and without the use of an oversampling technique (i.e., VAE-

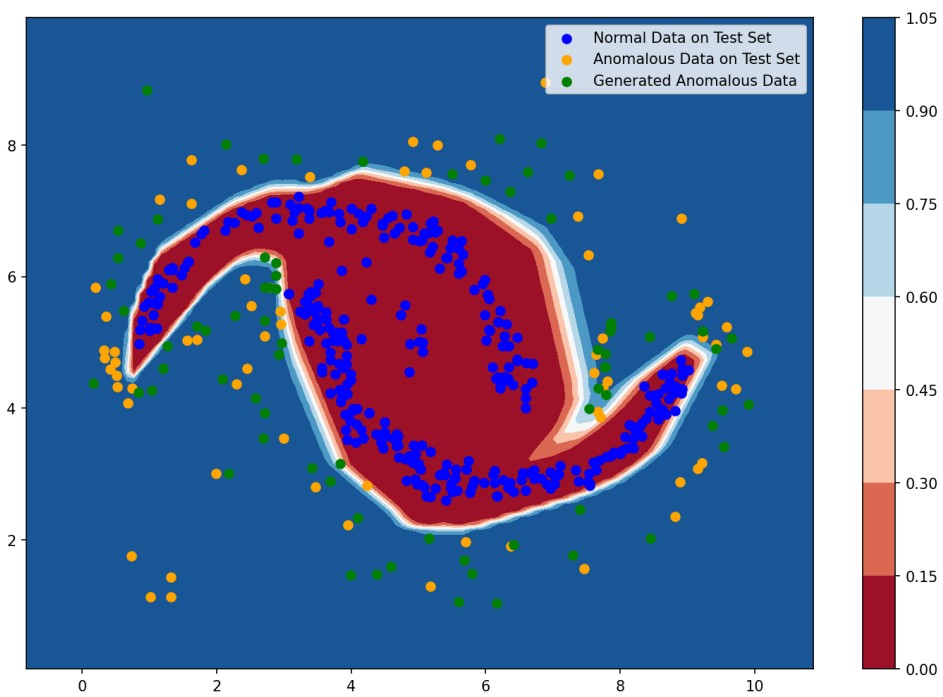

Figure 6: Visualization of the decision boundaries of the Detector Mixture of Mamba Experts (MoME) after training on generated anomalous data, original anomalous data, and normal data from the training set. The visualization highlights the resulting boundaries distinguishing anomalous and normal data in the test set.

| Methods | Transformer | | Mamba | | Swift Hydra(Transformer) | | Swift Hydra(Mamba) | |
|---|---|---|---|---|---|---|---|---|
| | AUCROC | TIF | AUCROC | TIF | AUCROC | TIF | AUCROC | TIF |
| Train/Test Ratio (40/60%) | 0.78 | 5.18 | 0.80 | 3.54 | 0.90 | 7.96 | **0.93** | 4.01 |
| Train/Test Ratio (30/70%) | 0.75 | 6.07 | 0.77 | 3.62 | 0.88 | 8.79 | **0.91** | 4.79 |
| Train/Test Ratio (20/80%) | 0.72 | 7.10 | 0.73 | 4.11 | 0.86 | 9.91 | **0.90** | 5.22 |
| Train/Test Ratio (10/90%) | 0.69 | 8.22 | 0.71 | 4.63 | 0.82 | 10.36 | **0.87** | 5.84 |
| | | | | | | | **TIF = Total Inference Time (Seconds)** | |

Table 5: Comparison of Vanilla Transformer and Vanilla Mamba with Transformer and Mamba enhanced by the Self-Reinforcing Module in Swift Hydra, evaluated based on AUC-ROC and Total Inference Time (TIF).

| Methods | Transformer (VAE-Geometry) | | Mamba (VAE-Geometry) | | Swift Hydra(Transformer) | | Swift Hydra(Mamba) | |
|---|---|---|---|---|---|---|---|---|
| | AUCROC | TIF | AUCROC | TIF | AUCROC | TIF | AUCROC | TIF |
| Train/Test Ratio (40/60%) | 0.83 | 5.23 | 0.85 | 3.40 | 0.90 | 7.96 | **0.93** | 4.01 |
| Train/Test Ratio (30/70%) | 0.80 | 6.11 | 0.83 | 3.58 | 0.88 | 8.79 | **0.91** | 4.79 |
| Train/Test Ratio (20/80%) | 0.78 | 7.02 | 0.81 | 4.02 | 0.86 | 9.91 | **0.90** | 5.22 |
| Train/Test Ratio (10/90%) | 0.75 | 8.38 | 0.78 | 4.65 | 0.82 | 8.36 | **0.87** | 5.84 |
| | | | | | | | **TIF = Total Inference Time (Seconds)** | |

Table 6: AUC-ROC performance of various backbone models when applying oversampling techniques

Geometry). It is important to note that VAE-Geometry is chosen as the oversampling method for other backbone models due to its best performance among the oversampling baselines. As shown in Table 5 and Table 6, Swift Hydra outperforms other methods in terms of AUC-ROC while maintaining competitive total inference time.

**AUC-ROC Evaluation.**    Regarding the accuracy in Table 5, standalone Mamba or Transformer models on ADBench exhibit significantly lower AUC-ROC compared to Swift Hydra (Mamba) and Swift Hydra (Transformer), which leverage the Self-Reinforcing Module. This is primarily because these models do not utilize any oversampling techniques (RL-guiled GenAI), which limits their data generalization capabilities. However, even when employing a powerful oversampling technique like VAE-Geometry (Table 6), the AUC-ROC of these models still falls short of that achieved by Swift Hydra. This is due to the fact that the datapoints generated by VAE-Geometry are not as diverse or of as high quality as those generated by Swift Hydra, as discussed in the previous experiment.

**Total Inference Time Evaluation.** In terms of total inference time, it is important to note that the Mamba model has a prediction complexity of O(1), while the Transformer has a prediction complexity of O(N). This makes Mamba substantially faster than both Transformer and Fully Connected networks. When Mamba is integrated into Swift Hydra with a Top $k$ Mixture of Experts ($k = 2$), one might expect the prediction time of Swift Hydra to be nearly double that of the regular Mamba model. However, the interesting outcome here is that we use only a two-layer depth for Mamba and overspecify the number of experts to 20. This allows us to capture the data complexity effectively, and since the model has just two layers, the additional time difference with $k = 2$ is minimal compared to a larger Mamba model.

### C.4.2   IMPACT OF PROBABILISTIC CLUSTER ASSIGNMENT ON MOME PERFORMANCE

| Method | Swift Hydra (MoME-Traditional Training Approach) | | Swift Hydra (MoME-Our Training Approach) | |
|---|---|---|---|---|
| | AUCROC | TIF | AUCROC | TIF |
| Train/Test Ratio (40/60%) | 0.892 | 4.015 | 0.934 | 4.012 |
| Train/Test Ratio (30/70%) | 0.865 | 4.771 | 0.913 | 4.793 |
| Train/Test Ratio (20/80%) | 0.853 | 5.234 | 0.902 | 5.221 |
| Train/Test Ratio (10/90%) | 0.828 | 5.922 | 0.874 | 5.843 |

Table 7:  Comparison of AUC-ROC and Total Inference Time (TIF) between Swift Hydra using the traditional MoME training approach and Swift Hydra with the proposed probabilistic training approach across different train/test ratios.

The results in Table 7 highlight a clear advantage of Swift Hydra with the proposed probabilistic training approach for MoME over the traditional MoME training approach. The primary reason for this improvement lies in addressing the "winner-take-all" problem that commonly affects the traditional training method.

In the traditional MoME approach, the gating network immediately starts assigning samples to experts based on their performance scores. During the early training steps, when the experts have not yet converged, their performance scores are arbitrary. This randomness can lead to one expert receiving disproportionately more samples than others, simply due to chance. As a result, this expert improves faster and continues to dominate sample assignments, creating a feedback loop where it becomes the sole contributor to predictions. This phenomenon, known as "winner-take-all," reduces the diversity of the expert ensemble and significantly hinders the generalization ability of the model.

In contrast, Swift Hydra with the probabilistic training approach temporarily deactivates the gating network during the early training phase. Instead, it uses a probabilistic assignment mechanism (as described in Equation 6) to ensure that all experts receive equal opportunities to learn. Once the experts have sufficiently converged, the gating network is reactivated to assign samples based on their performance. This method prevents any single expert from monopolizing the training process early on and ensures a more balanced and effective ensemble.

The results in the table demonstrate that this improved training strategy consistently achieves higher AUC-ROC scores across all train/test ratios compared to the traditional approach, without incurring additional Total Inference Time (TIF). This validates the effectiveness of addressing the "winner-take-all" problem to enhance both the generalization and overall performance of the model.

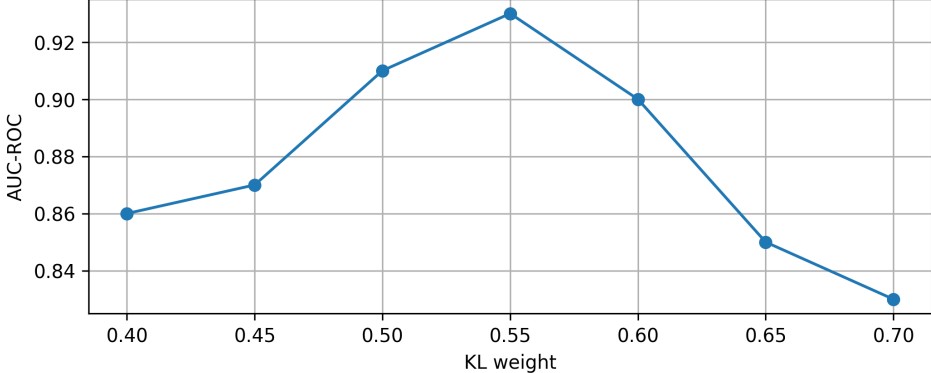

Figure 7: The average performance of various oversampling methods on the ADBench dataset. The evaluation metric is AUC-ROC

### C.4.3 KL DIVERGENCE AND RECONSTRUCTION LOSS TRADE-OFF IN CONDITIONAL VARIATIONAL AUTO ENCODER

In this section, we provide insights into how the weight impacts the C-VAE during data generation. Specifically, when optimizing the ELBO loss of C-VAE, we need to consider two components: the Reconstruction Loss and the KL Loss. If more weight is placed on the Reconstruction Loss, the C-VAE will generate data more cautiously, closely adhering to the target class. However, this caution results in less diverse samples. Conversely, if the weight is placed more on the KL Loss, the C-VAE generates more diverse data, but it may also produce samples that overlap with other classes.

To identify the optimal weight for the KL Loss, we experimented with various values of $p$ ranging from 0.4 to 0.7. If the weight for the KL Loss is $p$, the weight for the Reconstruction Loss will be $1 - p$. As shown in Figure 7, the optimal weight for the KL Loss is found to be 0.55. In the range of 0.4 to 0.5, the C-VAE generates overly cautious samples, resulting in a lack of diversity. Consequently, the 200 episodes of RL training are insufficient to cover the entire set of anomalous data in the test set. On the other hand, in the range of 0.6 to 0.7, the model focuses too much on optimizing the KL Loss, leading to the generation of highly diverse anomalous samples. However, these samples tend to overlap significantly with normal samples, causing Swift Hydra's performance to decline after 200 episodes.

### C.5 RL-AGENT ASSISTANCE FREQUENCY EXPERIMENT

In this section, we illustrate the average percentage of invalid actions taken by the agent in each episode (averaged over 57 datasets from ADBench). From Figure 8, it is evident that in the initial stages, the RL-Agent generates a high number of invalid actions, frequently requiring assistance from the One-Step to Feasible Action algorithm described in Section 3.1.2. However, since every time the RL-Agent takes an invalid action, it learns in a supervised manner using the feasible action provided by the One-Step to Feasible Action algorithm, a significant reduction in invalid actions is observed after about 25 to 50 episodes.

As a result, the training time becomes considerably faster since fewer Gradient Descent steps are needed (as Gradient Descent is primarily performed to optimize the reward function whenever an invalid action is taken). From episode 150 onward, the RL-Agent rarely makes invalid actions, and its actions become highly effective (as reflected in the AUC-ROC in previous experiment), indicating that the RL-Agent has successfully generalized.

### C.6 DATASETS IN ADBENCH

ADBench features a comprehensive collection of 57 datasets designed for anomaly detection research, as detailed in the table. Among these, 47 datasets are well-established and widely used across various real-world domains, such as healthcare (e.g., disease diagnosis), audio and language

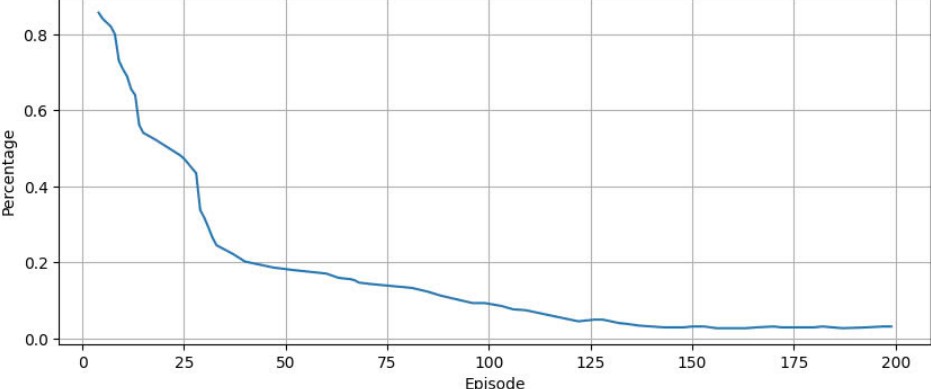

Figure 8: Percentage of invalid actions provided by the RL-Agent in each episode, with each episode consisting of 500 timesteps. The measurement is conducted over a total of 200 episodes. The X-axis represents the episodes, and the Y-axis shows the percentage of invalid actions provided by the RL-Agent.

processing (e.g., speech recognition), image analysis (e.g., object identification), and finance (e.g., fraud detection). Additionally, ADBench introduces 10 more complex datasets from computer vision (CV) and natural language processing (NLP) domains, enriched with larger sample sizes and higher-dimensional features. These datasets utilize pretrained models to extract embeddings, enabling the representation of more complex patterns. All datasets are provided in a user-friendly format as compressed NumPy array files ('.npz'), with detailed instructions and processing codes available for seamless application

| Number | Data | # Samples | # Features | # Anomaly | % Anomaly | Category |
|--------|------|-----------|-----------|-----------|-----------|----------|
| 1 | ALOI | 49534 | 27 | 1508 | 3.04 | Image |
| 2 | annthyroid | 7200 | 6 | 534 | 7.42 | Healthcare |
| 3 | backdoor | 95329 | 196 | 2329 | 2.44 | Network |
| 4 | breastw | 683 | 9 | 239 | 34.99 | Healthcare |
| 5 | campaign | 41188 | 62 | 4640 | 11.27 | Finance |
| 6 | cardio | 1831 | 21 | 176 | 9.61 | Healthcare |
| 7 | Cardiotocography | 2114 | 21 | 466 | 22.04 | Healthcare |
| 8 | celeba | 202599 | 39 | 4547 | 2.24 | Image |
| 9 | census | 299285 | 500 | 18568 | 6.20 | Sociology |
| 10 | cover | 286048 | 10 | 2747 | 0.96 | Botany |
| 11 | donors | 619326 | 10 | 36710 | 5.93 | Sociology |
| 12 | fault | 1941 | 27 | 673 | 34.67 | Physical |
| 13 | fraud | 284807 | 29 | 492 | 0.17 | Finance |
| 14 | glass | 214 | 7 | 9 | 4.21 | Forensic |
| 15 | Hepatitis | 80 | 19 | 13 | 16.25 | Healthcare |
| 16 | http | 567498 | 3 | 2211 | 0.39 | Web |
| 17 | InternetAds | 1966 | 1555 | 368 | 18.72 | Image |
| 18 | Ionosphere | 351 | 32 | 126 | 35.90 | Oryctognosy |
| 19 | landsat | 6435 | 36 | 1333 | 20.71 | Astronautics |
| 20 | letter | 1600 | 32 | 100 | 6.25 | Image |
| 21 | Lymphography | 148 | 18 | 6 | 4.05 | Healthcare |
| 22 | magic.gamma | 19020 | 10 | 6688 | 35.16 | Physical |
| 23 | mammography | 11183 | 6 | 260 | 2.32 | Healthcare |
| 24 | mnist | 7603 | 100 | 700 | 9.21 | Image |
| 25 | musk | 3062 | 166 | 97 | 3.17 | Chemistry |
| 26 | optdigits | 5216 | 64 | 150 | 2.88 | Image |
| 27 | PageBlocks | 5393 | 10 | 510 | 9.46 | Document |
| 28 | pendigits | 6870 | 16 | 156 | 2.27 | Image |
| 29 | Pima | 768 | 8 | 268 | 34.90 | Healthcare |

| 30 | satellite | 6435 | 36 | 2036 | 31.64 | Astronautics |
| 31 | satimage-2 | 5803 | 36 | 71 | 1.22 | Astronautics |
| 32 | shuttle | 49097 | 9 | 3511 | 7.15 | Astronautics |
| 33 | skin | 245057 | 3 | 50859 | 20.75 | Image |
| 34 | smtp | 95156 | 3 | 30 | 0.03 | Web |
| 35 | SpamBase | 4207 | 57 | 1679 | 39.91 | Document |
| 36 | speech | 3686 | 400 | 61 | 1.65 | Linguistics |
| 37 | Stamps | 340 | 9 | 31 | 9.12 | Document |
| 38 | thyroid | 3772 | 6 | 93 | 2.47 | Healthcare |
| 39 | vertebral | 240 | 6 | 30 | 12.50 | Biology |
| 40 | vowels | 1456 | 12 | 50 | 3.43 | Linguistics |
| 41 | Waveform | 3443 | 21 | 100 | 2.90 | Physics |
| 42 | WBC | 223 | 9 | 10 | 4.48 | Healthcare |
| 43 | WDBC | 367 | 30 | 10 | 2.72 | Healthcare |
| 44 | Wilt | 4819 | 5 | 257 | 5.33 | Botany |
| 45 | wine | 129 | 13 | 10 | 7.75 | Chemistry |
| 46 | WPBC | 198 | 33 | 47 | 23.74 | Healthcare |
| 47 | yeast | 1484 | 8 | 507 | 34.16 | Biology |
| 48 | CIFAR10 | 5263 | 512 | 263 | 5.00 | Image |
| 49 | FashionMNIST | 6315 | 512 | 315 | 5.00 | Image |
| 50 | MNIST-C | 10000 | 512 | 500 | 5.00 | Image |
| 51 | MVTec-AD | 5354 | 512 | 1258 | 23.50 | Image |
| 52 | SVHN | 5208 | 512 | 260 | 5.00 | Image |
| 53 | Agnews | 10000 | 768 | 500 | 5.00 | NLP |
| 54 | Amazon | 10000 | 768 | 500 | 5.00 | NLP |
| 55 | Imdb | 10000 | 768 | 500 | 5.00 | NLP |
| 56 | Yelp | 10000 | 768 | 500 | 5.00 | NLP |
| 57 | 20newsgroups | 11905 | 768 | 591 | 4.96 | NLP |

Table 8: Summary of Datasets used in ADBench Benchmark. The table outlines key characteristics of 57 anomaly detection datasets, including the number of samples, features, anomalies, and their respective categories, spanning diverse domains such as image analysis, healthcare, finance, and natural language processing.

## C.7 SWIFT HYDRA: CLASS-WISE PRECISION, RECALL, AND F1-SCORE ON ADBENCH

This final section provides a comprehensive comparison of various state-of-the-art anomaly detection methods, including DTE, Rejex, and ADGym, against Swift Hydra across 57 datasets from ADBench. The evaluation criteria include Precision, Recall, F1 score, support for each class, and AUC-ROC. It is important to note that we did not set a unique threshold for each individual dataset; instead, we applied a common threshold across all 57 datasets in ADBench. As a result, the Precision might not be very high. However, the focus should also be on the Recall and AUC-ROC, as these metrics more accurately reflect the model's ability to detect anomalous data if the correct threshold is chosen. This approach gives a clearer picture of how well Swift Hydra can identify anomalies in various datasets.

| | | DTE | | | Rejex | | | ADGym | | | Swift Hydra | | | |
| --- | --- | --- | --- | --- | --- | --- | --- | --- | --- | --- | --- | --- | --- | --- |
| **ALOI** | | **P** | **R** | **F1** | **P** | **R** | **F1** | **P** | **R** | **F1** | **P** | **R** | **F1** | **support** |
| -1 | | 0.97 | 0.97 | 0.97 | 0.97 | 0.95 | 0.96 | 0.92 | 0.38 | 0.54 | 0.99 | 0.44 | 0.61 | 28821 |
| 1 | | 0.03 | 0.03 | 0.03 | 0.04 | 0.07 | 0.05 | 0.00 | 0.75 | 0.05 | 0.04 | 0.81 | 0.08 | 900 |
| accuracy | | 0.94 | | | 0.92 | | | 0.39 | | | 0.45 | | | 29721 |
| macro avg | | 0.50 | 0.50 | 0.50 | 0.50 | 0.51 | 0.50 | 0.46 | 0.57 | 0.30 | 0.52 | 0.63 | 0.35 | 29721 |
| weighted avg | | 0.94 | 0.94 | 0.94 | 0.94 | 0.92 | 0.93 | 0.89 | 0.39 | 0.53 | 0.96 | 0.45 | 0.60 | 29721 |
| AUC-ROC | | 0.535 | | | 0.53 | | | 0.66 | | | 0.70 | | | |

**Table 9 continued from previous page**

| | | DTE | | | Rejex | | | ADGym | | | Swift Hydra | | | |
|---|---|---|---|---|---|---|---|---|---|---|---|---|---|---|
| **annthyroid** | P | R | F1 | P | R | F1 | P | R | F1 | P | R | F1 | support |
| -1 | 0.94 | 0.95 | 0.95 | 0.98 | 0.94 | 0.96 | 0.95 | 0.88 | 0.92 | 1.00 | 0.95 | 0.97 | 4001 |
| 1 | 0.32 | 0.29 | 0.31 | 0.52 | 0.77 | 0.62 | 0.56 | 0.92 | 0.71 | 0.60 | 0.98 | 0.75 | 319 |
| accuracy | | 0.90 | | | 0.93 | | | 0.89 | | | 0.95 | | 4320 |
| macro avg | 0.63 | 0.62 | 0.63 | 0.75 | 0.86 | 0.79 | 0.75 | 0.90 | 0.82 | 0.80 | 0.97 | 0.86 | 4320 |
| weighted avg | 0.90 | 0.90 | 0.90 | 0.95 | 0.93 | 0.94 | 0.92 | 0.89 | 0.91 | 0.97 | 0.95 | 0.96 | 4320 |
| AUC-ROC | | 0.81 | | | 0.96 | | | 0.93 | | | 0.98 | | |
| **backdoor** | P | R | F1 | P | R | F1 | P | R | F1 | P | R | F1 | support |
| -1 | 0.98 | 0.98 | 0.98 | 0.99 | 0.99 | 0.99 | 0.94 | 0.95 | 0.96 | 1.00 | 0.99 | 0.99 | 55790 |
| 1 | 0.01 | 0.01 | 0.01 | 0.68 | 0.47 | 0.56 | 0.55 | 0.89 | 0.67 | 0.61 | 0.92 | 0.74 | 1408 |
| accuracy | | 0.95 | | | 0.98 | | | 0.95 | | | 0.98 | | 57198 |
| macro avg | 0.49 | 0.49 | 0.49 | 0.83 | 0.73 | 0.77 | 0.75 | 0.92 | 0.82 | 0.81 | 0.95 | 0.86 | 57198 |
| weighted avg | 0.95 | 0.95 | 0.95 | 0.98 | 0.98 | 0.98 | 0.93 | 0.95 | 0.95 | 0.99 | 0.98 | 0.99 | 57198 |
| AUC-ROC | | 0.76 | | | 0.90 | | | 0.94 | | | 0.98 | | |
| **breastw** | P | R | F1 | P | R | F1 | P | R | F1 | P | R | F1 | support |
| -1 | 0.99 | 0.94 | 0.97 | 1.00 | 0.69 | 0.82 | 0.95 | 0.93 | 0.94 | 0.99 | 0.97 | 0.98 | 264 |
| 1 | 0.91 | 0.99 | 0.94 | 0.64 | 1.00 | 0.78 | 0.88 | 0.92 | 0.91 | 0.94 | 0.98 | 0.96 | 146 |
| accuracy | | 0.96 | | | 0.80 | | | 0.93 | | | 0.97 | | 410 |
| macro avg | 0.95 | 0.97 | 0.96 | 0.82 | 0.85 | 0.80 | 0.92 | 0.92 | 0.92 | 0.97 | 0.97 | 0.97 | 410 |
| weighted avg | 0.96 | 0.96 | 0.96 | 0.87 | 0.80 | 0.80 | 0.93 | 0.93 | 0.93 | 0.97 | 0.97 | 0.97 | 410 |
| AUC-ROC | | 0.99 | | | 0.93 | | | 0.94 | | | 0.99 | | |
| **campaign** | P | R | F1 | P | R | F1 | P | R | F1 | P | R | F1 | support |
| -1 | 0.91 | 0.91 | 0.91 | 0.89 | 0.99 | 0.94 | 0.92 | 0.79 | 0.85 | 0.98 | 0.82 | 0.89 | 21920 |
| 1 | 0.32 | 0.32 | 0.32 | 0.27 | 0.04 | 0.08 | 0.33 | 0.79 | 0.47 | 0.38 | 0.83 | 0.52 | 2793 |
| accuracy | | 0.84 | | | 0.88 | | | 0.79 | | | 0.83 | | 24713 |
| macro avg | 0.62 | 0.62 | 0.62 | 0.58 | 0.51 | 0.51 | 0.63 | 0.79 | 0.66 | 0.68 | 0.83 | 0.71 | 24713 |
| weighted avg | 0.85 | 0.84 | 0.85 | 0.82 | 0.88 | 0.84 | 0.86 | 0.79 | 0.81 | 0.91 | 0.83 | 0.85 | 24713 |
| AUC-ROC | | 0.73 | | | 0.77 | | | 0.84 | | | 0.90 | | |
| **cardio** | P | R | F1 | P | R | F1 | P | R | F1 | P | R | F1 | support |
| -1 | 0.94 | 0.92 | 0.93 | 0.94 | 0.82 | 0.87 | 0.95 | 0.88 | 0.93 | 0.99 | 0.94 | 0.97 | 990 |
| 1 | 0.42 | 0.50 | 0.45 | 0.23 | 0.51 | 0.32 | 0.57 | 0.89 | 0.72 | 0.64 | 0.95 | 0.76 | 109 |
| accuracy | | 0.88 | | | 0.79 | | | 0.88 | | | 0.94 | | 1099 |
| macro avg | 0.68 | 0.71 | 0.69 | 0.59 | 0.66 | 0.60 | 0.76 | 0.88 | 0.83 | 0.82 | 0.94 | 0.86 | 1099 |
| weighted avg | 0.89 | 0.88 | 0.89 | 0.87 | 0.79 | 0.82 | 0.91 | 0.88 | 0.91 | 0.96 | 0.94 | 0.95 | 1099 |
| AUC-ROC | | 0.92 | | | 0.74 | | | 0.95 | | | 0.98 | | |
| **Cardiotocography** | P | R | F1 | P | R | F1 | P | R | F1 | P | R | F1 | support |
| -1 | 0.82 | 0.83 | 0.82 | 0.80 | 0.75 | 0.78 | 0.92 | 0.83 | 0.90 | 0.98 | 0.90 | 0.93 | 991 |
| 1 | 0.37 | 0.36 | 0.36 | 0.27 | 0.33 | 0.30 | 0.68 | 0.87 | 0.74 | 0.71 | 0.92 | 0.80 | 278 |
| accuracy | | 0.72 | | | 0.66 | | | 0.84 | | | 0.90 | | 1269 |
| macro avg | 0.59 | 0.59 | 0.59 | 0.54 | 0.54 | 0.54 | 0.80 | 0.85 | 0.82 | 0.84 | 0.91 | 0.87 | 1269 |
| weighted avg | 0.72 | 0.72 | 0.72 | 0.68 | 0.66 | 0.67 | 0.87 | 0.84 | 0.87 | 0.92 | 0.90 | 0.91 | 1269 |

**Table 9 continued from previous page**

| | DTE | | | Rejex | | | ADGym | | | Swift Hydra | | | |
|---|---|---|---|---|---|---|---|---|---|---|---|---|---|
| AUC-ROC | 0.73 | | | 0.53 | | | 0.91 | | | 0.95 | | | |
| **celeba** | **P** | **R** | **F1** | **P** | **R** | **F1** | **P** | **R** | **F1** | **P** | **R** | **F1** | **support** |
| -1 | 0.98 | 0.98 | 0.98 | 0.98 | 1.00 | 0.99 | 0.96 | 0.83 | 0.88 | 1.00 | 0.89 | 0.94 | 118850 |
| 1 | 0.09 | 0.09 | 0.09 | 0.03 | 0.00 | 0.00 | 0.11 | 0.85 | 0.21 | 0.15 | 0.91 | 0.26 | 2710 |
| accuracy | 0.96 | | | 0.98 | | | 0.83 | | | 0.89 | | | 121560 |
| macro avg | 0.53 | 0.53 | 0.53 | 0.50 | 0.50 | 0.50 | 0.53 | 0.84 | 0.55 | 0.58 | 0.90 | 0.60 | 121560 |
| weighted avg | 0.96 | 0.96 | 0.96 | 0.96 | 0.98 | 0.97 | 0.94 | 0.83 | 0.86 | 0.98 | 0.89 | 0.92 | 121560 |
| AUC-ROC | 0.72 | | | 0.79 | | | 0.88 | | | 0.95 | | | |
| **census** | **P** | **R** | **F1** | **P** | **R** | **F1** | **P** | **R** | **F1** | **P** | **R** | **F1** | **support** |
| -1 | 0.94 | 0.94 | 0.94 | 0.94 | 0.88 | 0.91 | 0.92 | 0.72 | 0.84 | 0.99 | 0.79 | 0.88 | 168432 |
| 1 | 0.04 | 0.04 | 0.04 | 0.06 | 0.12 | 0.08 | 0.15 | 0.77 | 0.31 | 0.21 | 0.84 | 0.34 | 11139 |
| accuracy | 0.88 | | | 0.84 | | | 0.73 | | | 0.79 | | | 179571 |
| macro avg | 0.49 | 0.49 | 0.49 | 0.50 | 0.50 | 0.50 | 0.54 | 0.75 | 0.58 | 0.60 | 0.82 | 0.61 | 179571 |
| weighted avg | 0.88 | 0.88 | 0.88 | 0.88 | 0.84 | 0.86 | 0.87 | 0.73 | 0.81 | 0.94 | 0.79 | 0.84 | 179571 |
| AUC-ROC | 0.59 | | | 0.65 | | | 0.79 | | | 0.85 | | | |
| **cover** | **P** | **R** | **F1** | **P** | **R** | **F1** | **P** | **R** | **F1** | **P** | **R** | **F1** | **support** |
| -1 | 0.99 | 0.99 | 0.99 | 0.99 | 0.99 | 0.99 | 0.95 | 0.96 | 0.94 | 1.00 | 0.99 | 1.00 | 169976 |
| 1 | 0.09 | 0.09 | 0.09 | 0.06 | 0.05 | 0.06 | 0.50 | 0.95 | 0.65 | 0.54 | 1.00 | 0.70 | 1653 |
| accuracy | 0.98 | | | 0.98 | | | 0.96 | | | 0.99 | | | 171629 |
| macro avg | 0.54 | 0.54 | 0.54 | 0.53 | 0.52 | 0.52 | 0.73 | 0.96 | 0.80 | 0.77 | 0.99 | 0.85 | 171629 |
| weighted avg | 0.98 | 0.98 | 0.98 | 0.98 | 0.98 | 0.98 | 0.95 | 0.96 | 0.94 | 1.00 | 0.99 | 0.99 | 171629 |
| AUC-ROC | 0.90 | | | 0.74 | | | 0.97 | | | 1.00 | | | |
| **donors** | **P** | **R** | **F1** | **P** | **R** | **F1** | **P** | **R** | **F1** | **P** | **R** | **F1** | **support** |
| -1 | 0.94 | 0.94 | 0.94 | 0.94 | 0.99 | 0.96 | 0.93 | 0.95 | 0.94 | 1.00 | 1.00 | 1.00 | 349483 |
| 1 | 0.10 | 0.10 | 0.10 | 0.14 | 0.03 | 0.05 | 0.95 | 0.95 | 0.95 | 1.00 | 1.00 | 1.00 | 22113 |
| accuracy | 0.89 | | | 0.93 | | | 0.95 | | | 1.00 | | | 371596 |
| macro avg | 0.52 | 0.52 | 0.52 | 0.54 | 0.51 | 0.51 | 0.94 | 0.95 | 0.94 | 1.00 | 1.00 | 1.00 | 371596 |
| weighted avg | 0.89 | 0.89 | 0.89 | 0.89 | 0.93 | 0.91 | 0.93 | 0.95 | 0.94 | 1.00 | 1.00 | 1.00 | 371596 |
| AUC-ROC | 0.76 | | | 0.74 | | | 0.95 | | | 1.00 | | | |
| **fault** | **P** | **R** | **F1** | **P** | **R** | **F1** | **P** | **R** | **F1** | **P** | **R** | **F1** | **support** |
| -1 | 0.69 | 0.66 | 0.68 | 0.68 | 0.89 | 0.77 | 0.78 | 0.62 | 0.69 | 0.81 | 0.65 | 0.72 | 751 |
| 1 | 0.43 | 0.45 | 0.44 | 0.53 | 0.23 | 0.32 | 0.49 | 0.68 | 0.57 | 0.54 | 0.73 | 0.62 | 414 |
| accuracy | 0.59 | | | 0.65 | | | 0.64 | | | 0.68 | | | 1165 |
| macro avg | 0.56 | 0.56 | 0.56 | 0.60 | 0.56 | 0.54 | 0.63 | 0.65 | 0.63 | 0.67 | 0.69 | 0.67 | 1165 |
| weighted avg | 0.60 | 0.59 | 0.59 | 0.62 | 0.65 | 0.61 | 0.67 | 0.64 | 0.65 | 0.71 | 0.68 | 0.69 | 1165 |
| AUC-ROC | 0.54 | | | 0.58 | | | 0.68 | | | 0.74 | | | |
| **fraud** | **P** | **R** | **F1** | **P** | **R** | **F1** | **P** | **R** | **F1** | **P** | **R** | **F1** | **support** |
| -1 | 1.00 | 1.00 | 1.00 | 1.00 | 0.99 | 1.00 | 0.96 | 0.86 | 0.91 | 1.00 | 0.92 | 0.96 | 170593 |
| 1 | 0.30 | 0.28 | 0.29 | 0.13 | 0.81 | 0.22 | 0.00 | 0.85 | 0.00 | 0.02 | 0.92 | 0.04 | 292 |
| accuracy | 1.00 | | | 0.99 | | | 0.86 | | | 0.92 | | | 170885 |
| macro avg | 0.65 | 0.64 | 0.65 | 0.56 | 0.90 | 0.61 | 0.48 | 0.85 | 0.45 | 0.51 | 0.92 | 0.50 | 170885 |

**Table 9 continued from previous page**

| | DTE | | | Rejex | | | ADGym | | | Swift Hydra | | | |
|---|---|---|---|---|---|---|---|---|---|---|---|---|---|
| weighted avg | 1.00 | 1.00 | 1.00 | 1.00 | 0.99 | 0.99 | 0.96 | 0.86 | 0.91 | 1.00 | 0.92 | 0.96 | 170885 |
| AUC-ROC | | 0.95 | | | 0.95 | | | 0.92 | | | 0.96 | | |
| **glass** | **P** | **R** | **F1** | **P** | **R** | **F1** | **P** | **R** | **F1** | **P** | **R** | **F1** | **support** |
| -1 | 0.95 | 0.98 | 0.96 | 0.00 | 0.00 | 0.00 | 0.93 | 0.75 | 0.83 | 0.96 | 0.81 | 0.88 | 123 |
| 1 | 0.00 | 0.00 | 0.00 | 0.05 | 1.00 | 0.09 | 0.04 | 0.27 | 0.09 | 0.08 | 0.33 | 0.13 | 6 |
| accuracy | | 0.93 | | | 0.05 | | | 0.73 | | | 0.79 | | 129 |
| macro avg | 0.48 | 0.49 | 0.48 | 0.02 | 0.50 | 0.04 | 0.48 | 0.51 | 0.46 | 0.52 | 0.57 | 0.51 | 129 |
| weighted avg | 0.91 | 0.93 | 0.92 | 0.00 | 0.05 | 0.00 | 0.89 | 0.73 | 0.80 | 0.92 | 0.79 | 0.85 | 129 |
| AUC-ROC | | 0.76 | | | 0.75 | | | 0.78 | | | 0.84 | | |
| **Hepatitis** | **P** | **R** | **F1** | **P** | **R** | **F1** | **P** | **R** | **F1** | **P** | **R** | **F1** | **support** |
| -1 | 0.85 | 0.83 | 0.84 | 0.00 | 0.00 | 0.00 | 0.90 | 0.62 | 0.75 | 0.97 | 0.67 | 0.79 | 42 |
| 1 | 0.00 | 0.00 | 0.00 | 0.13 | 1.00 | 0.22 | 0.22 | 0.79 | 0.33 | 0.26 | 0.83 | 0.40 | 6 |
| accuracy | | 0.73 | | | 0.13 | | | 0.64 | | | 0.69 | | 0.69 |
| macro avg | 0.43 | 0.42 | 0.42 | 0.06 | 0.50 | 0.11 | 0.56 | 0.71 | 0.54 | 0.61 | 0.75 | 0.59 | 48 |
| weighted avg | 0.75 | 0.73 | 0.74 | 0.02 | 0.13 | 0.03 | 0.81 | 0.64 | 0.70 | 0.88 | 0.69 | 0.74 | 48 |
| AUC-ROC | | 0.72 | | | 0.42 | | | 0.74 | | | 0.80 | | |
| **http** | **P** | **R** | **F1** | **P** | **R** | **F1** | **P** | **R** | **F1** | **P** | **R** | **F1** | **support** |
| -1 | 1.00 | 1.00 | 1.00 | 1.00 | 0.94 | 0.97 | 0.95 | 0.96 | 0.94 | 1.00 | 1.00 | 1.00 | 339161 |
| 1 | 0.23 | 0.02 | 0.03 | 0.06 | 1.00 | 0.12 | 0.72 | 0.95 | 0.79 | 0.76 | 1.00 | 0.86 | 1338 |
| accuracy | | 1.00 | | | 0.94 | | | 0.96 | | | 1.00 | | 340499 |
| macro avg | 0.61 | 0.51 | 0.51 | 0.53 | 0.97 | 0.54 | 0.84 | 0.96 | 0.87 | 0.88 | 1.00 | 0.93 | 340499 |
| weighted avg | 0.99 | 1.00 | 0.99 | 1.00 | 0.94 | 0.97 | 0.95 | 0.96 | 0.94 | 1.00 | 1.00 | 1.00 | 340499 |
| AUC-ROC | | 1.00 | | | 1.00 | | | 0.94 | | | 1.00 | | |
| **InternetAds** | **P** | **R** | **F1** | **P** | **R** | **F1** | **P** | **R** | **F1** | **P** | **R** | **F1** | **support** |
| -1 | 0.88 | 0.88 | 0.88 | 0.89 | 0.44 | 0.59 | 0.93 | 0.63 | 0.74 | 0.98 | 0.68 | 0.81 | 965 |
| 1 | 0.45 | 0.46 | 0.46 | 0.23 | 0.75 | 0.35 | 0.37 | 0.88 | 0.53 | 0.40 | 0.94 | 0.56 | 215 |
| accuracy | | 0.80 | | | 0.50 | | | 0.67 | | | 0.73 | | 1180 |
| macro avg | 0.67 | 0.67 | 0.67 | 0.56 | 0.60 | 0.47 | 0.65 | 0.75 | 0.63 | 0.69 | 0.81 | 0.68 | 1180 |
| weighted avg | 0.80 | 0.80 | 0.80 | 0.77 | 0.50 | 0.55 | 0.83 | 0.67 | 0.70 | 0.88 | 0.73 | 0.76 | 1180 |
| AUC-ROC | | 0.68 | | | 0.67 | | | 0.88 | | | 0.92 | | |
| **Ionosphere** | **P** | **R** | **F1** | **P** | **R** | **F1** | **P** | **R** | **F1** | **P** | **R** | **F1** | **support** |
| -1 | 0.82 | 0.76 | 0.79 | 0.00 | 0.00 | 0.00 | 0.86 | 0.85 | 0.84 | 0.90 | 0.90 | 0.90 | 135 |
| 1 | 0.62 | 0.70 | 0.65 | 0.36 | 1.00 | 0.53 | 0.76 | 0.78 | 0.76 | 0.83 | 0.82 | 0.82 | 76 |
| accuracy | | 0.74 | | | 0.36 | | | 0.82 | | | 0.87 | | 211 |
| macro avg | 0.72 | 0.73 | 0.72 | 0.18 | 0.50 | 0.27 | 0.81 | 0.81 | 0.80 | 0.86 | 0.86 | 0.86 | 211 |
| weighted avg | 0.74 | 0.74 | 0.74 | 0.13 | 0.36 | 0.19 | 0.82 | 0.82 | 0.81 | 0.87 | 0.87 | 0.87 | 211 |
| AUC-ROC | | 0.86 | | | 0.96 | | | 0.85 | | | 0.91 | | |
| **landsat** | **P** | **R** | **F1** | **P** | **R** | **F1** | **P** | **R** | **F1** | **P** | **R** | **F1** | **support** |
| -1 | 0.81 | 0.79 | 0.80 | 0.80 | 0.94 | 0.86 | 0.92 | 0.80 | 0.86 | 0.97 | 0.86 | 0.91 | 3068 |
| 1 | 0.24 | 0.26 | 0.25 | 0.23 | 0.06 | 0.10 | 0.56 | 0.83 | 0.68 | 0.62 | 0.89 | 0.73 | 793 |
| accuracy | | 0.68 | | | 0.76 | | | 0.80 | | | 0.87 | | 3861 |

**Table 9 continued from previous page**

| | DTE | | | Rejex | | | ADGym | | | Swift Hydra | | | |
|---|---|---|---|---|---|---|---|---|---|---|---|---|---|
| macro avg | 0.52 | 0.52 | 0.52 | 0.51 | 0.50 | 0.48 | 0.74 | 0.81 | 0.77 | 0.79 | 0.87 | 0.82 | 3861 |
| weighted avg | 0.69 | 0.68 | 0.68 | 0.68 | 0.76 | 0.71 | 0.85 | 0.80 | 0.82 | 0.90 | 0.87 | 0.87 | 3861 |
| AUC-ROC | 0.50 | | | 0.53 | | | 0.91 | | | 0.94 | | | |
| **letter** | **P** | **R** | **F1** | **P** | **R** | **F1** | **P** | **R** | **F1** | **P** | **R** | **F1** | **support** |
| -1 | 0.94 | 0.94 | 0.94 | 0.98 | 0.65 | 0.78 | 0.94 | 0.76 | 0.83 | 0.98 | 0.82 | 0.89 | 900 |
| 1 | 0.09 | 0.10 | 0.10 | 0.14 | 0.83 | 0.24 | 0.16 | 0.63 | 0.27 | 0.20 | 0.68 | 0.31 | 60 |
| accuracy | 0.88 | | | 0.66 | | | 0.75 | | | 0.81 | | | 960 |
| macro avg | 0.52 | 0.52 | 0.52 | 0.56 | 0.74 | 0.51 | 0.55 | 0.69 | 0.55 | 0.59 | 0.75 | 0.60 | 960 |
| weighted avg | 0.89 | 0.88 | 0.89 | 0.93 | 0.66 | 0.75 | 0.89 | 0.75 | 0.80 | 0.93 | 0.81 | 0.85 | 960 |
| AUC-ROC | 0.57 | | | 0.85 | | | 0.77 | | | 0.80 | | | |
| **Lymphography** | **P** | **R** | **F1** | **P** | **R** | **F1** | **P** | **R** | **F1** | **P** | **R** | **F1** | **support** |
| -1 | 1.00 | 0.95 | 0.98 | 0.00 | 0.00 | 0.00 | 0.90 | 0.94 | 0.92 | 0.95 | 0.99 | 0.97 | 84 |
| 1 | 0.56 | 1.00 | 0.71 | 0.06 | 1.00 | 0.11 | 0.47 | 0.17 | 0.24 | 0.50 | 0.20 | 0.29 | 5 |
| accuracy | 0.96 | | | 0.06 | | | 0.90 | | | 0.94 | | | 89 |
| macro avg | 0.78 | 0.98 | 0.85 | 0.03 | 0.50 | 0.05 | 0.68 | 0.56 | 0.58 | 0.73 | 0.59 | 0.63 | 89 |
| weighted avg | 0.98 | 0.96 | 0.96 | 0.00 | 0.06 | 0.01 | 0.88 | 0.90 | 0.89 | 0.93 | 0.94 | 0.93 | 89 |
| AUC-ROC | 0.99 | | | 0.99 | | | 0.64 | | | 0.69 | | | |
| **magic.gamma** | **P** | **R** | **F1** | **P** | **R** | **F1** | **P** | **R** | **F1** | **P** | **R** | **F1** | **support** |
| -1 | 0.75 | 0.75 | 0.75 | 0.72 | 0.95 | 0.82 | 0.87 | 0.69 | 0.76 | 0.93 | 0.74 | 0.82 | 7422 |
| 1 | 0.54 | 0.55 | 0.54 | 0.75 | 0.31 | 0.44 | 0.60 | 0.86 | 0.71 | 0.65 | 0.90 | 0.75 | 3990 |
| accuracy | 0.68 | | | 0.72 | | | 0.75 | | | 0.79 | | | 11412 |
| macro avg | 0.65 | 0.65 | 0.65 | 0.74 | 0.63 | 0.63 | 0.73 | 0.77 | 0.74 | 0.79 | 0.82 | 0.79 | 11412 |
| weighted avg | 0.68 | 0.68 | 0.68 | 0.73 | 0.72 | 0.69 | 0.77 | 0.75 | 0.75 | 0.83 | 0.79 | 0.80 | 11412 |
| AUC-ROC | 0.73 | | | 0.76 | | | 0.88 | | | 0.92 | | | |
| **mammography** | **P** | **R** | **F1** | **P** | **R** | **F1** | **P** | **R** | **F1** | **P** | **R** | **F1** | **support** |
| -1 | 0.98 | 0.98 | 0.98 | 0.99 | 0.83 | 0.90 | 0.93 | 0.80 | 0.90 | 1.00 | 0.87 | 0.93 | 6558 |
| 1 | 0.25 | 0.26 | 0.25 | 0.08 | 0.62 | 0.14 | 0.06 | 0.83 | 0.17 | 0.13 | 0.88 | 0.23 | 152 |
| accuracy | 0.97 | | | 0.82 | | | 0.80 | | | 0.87 | | | 6710 |
| macro avg | 0.62 | 0.62 | 0.62 | 0.53 | 0.72 | 0.52 | 0.50 | 0.82 | 0.53 | 0.56 | 0.87 | 0.58 | 6710 |
| weighted avg | 0.97 | 0.97 | 0.97 | 0.97 | 0.82 | 0.88 | 0.91 | 0.80 | 0.88 | 0.98 | 0.87 | 0.91 | 6710 |
| AUC-ROC | 0.86 | | | 0.84 | | | 0.86 | | | 0.91 | | | |
| **mnist** | **P** | **R** | **F1** | **P** | **R** | **F1** | **P** | **R** | **F1** | **P** | **R** | **F1** | **support** |
| -1 | 0.93 | 0.93 | 0.93 | 0.94 | 0.97 | 0.95 | 0.94 | 0.89 | 0.91 | 1.00 | 0.94 | 0.97 | 4172 |
| 1 | 0.29 | 0.30 | 0.29 | 0.46 | 0.28 | 0.35 | 0.57 | 0.91 | 0.70 | 0.60 | 0.97 | 0.74 | 390 |
| accuracy | 0.88 | | | 0.91 | | | 0.90 | | | 0.94 | | | 4562 |
| macro avg | 0.61 | 0.61 | 0.61 | 0.70 | 0.62 | 0.65 | 0.76 | 0.90 | 0.80 | 0.80 | 0.95 | 0.85 | 4562 |
| weighted avg | 0.88 | 0.88 | 0.88 | 0.89 | 0.91 | 0.90 | 0.91 | 0.90 | 0.89 | 0.96 | 0.94 | 0.95 | 4562 |
| AUC-ROC | 0.77 | | | 0.79 | | | 0.92 | | | 0.98 | | | |
| **musk** | **P** | **R** | **F1** | **P** | **R** | **F1** | **P** | **R** | **F1** | **P** | **R** | **F1** | **support** |
| -1 | 1.00 | 1.00 | 1.00 | 0.98 | 1.00 | 0.99 | 0.96 | 0.93 | 0.95 | 1.00 | 1.00 | 1.00 | 1777 |
| 1 | 0.95 | 1.00 | 0.98 | 0.72 | 0.34 | 0.47 | 0.95 | 0.94 | 0.96 | 1.00 | 1.00 | 1.00 | 61 |

**Table 9 continued from previous page**

| | DTE | | | Rejex | | | ADGym | | | Swift Hydra | | | |
|---|---|---|---|---|---|---|---|---|---|---|---|---|---|
| accuracy | 1.00 | | | 0.97 | | | 0.93 | | | 1.00 | | | 1838 |
| macro avg | 0.98 | 1.00 | 0.99 | 0.85 | 0.67 | 0.73 | 0.95 | 0.94 | 0.95 | 1.00 | 1.00 | 1.00 | 1838 |
| weighted avg | 1.00 | 1.00 | 1.00 | 0.97 | 0.97 | 0.97 | 0.96 | 0.93 | 0.95 | 1.00 | 1.00 | 1.00 | 1838 |
| AUC-ROC | 1.00 | | | 0.96 | | | 0.94 | | | 1.00 | | | |
| **optdigits** | **P** | **R** | **F1** | **P** | **R** | **F1** | **P** | **R** | **F1** | **P** | **R** | **F1** | **support** |
| -1 | 0.97 | 0.96 | 0.97 | 0.97 | 0.88 | 0.92 | 0.94 | 0.91 | 0.92 | 1.00 | 0.97 | 0.98 | 3044 |
| 1 | 0.01 | 0.01 | 0.01 | 0.00 | 0.01 | 0.00 | 0.38 | 0.97 | 0.59 | 0.45 | 1.00 | 0.62 | 86 |
| accuracy | 0.94 | | | 0.86 | | | 0.91 | | | 0.97 | | | 3130 |
| macro avg | 0.49 | 0.49 | 0.49 | 0.49 | 0.45 | 0.46 | 0.66 | 0.94 | 0.75 | 0.72 | 0.98 | 0.80 | 3130 |
| weighted avg | 0.95 | 0.94 | 0.94 | 0.94 | 0.86 | 0.90 | 0.93 | 0.91 | 0.91 | 0.99 | 0.97 | 0.97 | 3130 |
| AUC-ROC | 0.71 | | | 0.36 | | | 0.97 | | | 1.00 | | | |
| **PageBlocks** | **P** | **R** | **F1** | **P** | **R** | **F1** | **P** | **R** | **F1** | **P** | **R** | **F1** | **support** |
| -1 | 0.94 | 0.94 | 0.94 | 0.97 | 0.95 | 0.96 | 0.94 | 0.86 | 0.89 | 1.00 | 0.92 | 0.95 | 2916 |
| 1 | 0.43 | 0.45 | 0.44 | 0.59 | 0.69 | 0.64 | 0.49 | 0.90 | 0.66 | 0.56 | 0.96 | 0.71 | 320 |
| accuracy | 0.89 | | | 0.92 | | | 0.86 | | | 0.92 | | | 3236 |
| macro avg | 0.69 | 0.69 | 0.69 | 0.78 | 0.82 | 0.80 | 0.72 | 0.88 | 0.78 | 0.78 | 0.94 | 0.83 | 3236 |
| weighted avg | 0.89 | 0.89 | 0.89 | 0.93 | 0.92 | 0.93 | 0.90 | 0.86 | 0.87 | 0.95 | 0.92 | 0.93 | 3236 |
| AUC-ROC | 0.92 | | | 0.95 | | | 0.92 | | | 0.98 | | | |
| **pendigits** | **P** | **R** | **F1** | **P** | **R** | **F1** | **P** | **R** | **F1** | **P** | **R** | **F1** | **support** |
| -1 | 0.98 | 0.98 | 0.98 | 0.99 | 0.82 | 0.89 | 0.95 | 0.93 | 0.94 | 1.00 | 0.98 | 0.99 | 4032 |
| 1 | 0.26 | 0.26 | 0.26 | 0.07 | 0.59 | 0.12 | 0.50 | 0.94 | 0.68 | 0.56 | 0.99 | 0.72 | 90 |
| accuracy | 0.97 | | | 0.81 | | | 0.93 | | | 0.98 | | | 4122 |
| macro avg | 0.62 | 0.62 | 0.62 | 0.53 | 0.70 | 0.51 | 0.72 | 0.93 | 0.81 | 0.78 | 0.99 | 0.85 | 4122 |
| weighted avg | 0.97 | 0.97 | 0.97 | 0.97 | 0.81 | 0.88 | 0.94 | 0.93 | 0.94 | 0.99 | 0.98 | 0.99 | 4122 |
| AUC-ROC | 0.93 | | | 0.83 | | | 0.95 | | | 1.00 | | | |
| **Pima** | **P** | **R** | **F1** | **P** | **R** | **F1** | **P** | **R** | **F1** | **P** | **R** | **F1** | **support** |
| -1 | 0.73 | 0.66 | 0.69 | 0.00 | 0.00 | 0.00 | 0.74 | 0.59 | 0.66 | 0.78 | 0.64 | 0.70 | 294 |
| 1 | 0.48 | 0.56 | 0.52 | 0.36 | 1.00 | 0.53 | 0.48 | 0.60 | 0.54 | 0.52 | 0.67 | 0.58 | 167 |
| accuracy | 0.62 | | | 0.36 | | | 0.60 | | | 0.65 | | | 461 |
| macro avg | 0.60 | 0.61 | 0.60 | 0.18 | 0.50 | 0.27 | 0.61 | 0.60 | 0.60 | 0.65 | 0.66 | 0.64 | 461 |
| weighted avg | 0.64 | 0.62 | 0.63 | 0.13 | 0.36 | 0.19 | 0.65 | 0.60 | 0.62 | 0.68 | 0.65 | 0.66 | 461 |
| AUC-ROC | 0.66 | | | 0.57 | | | 0.66 | | | 0.70 | | | |
| **satellite** | **P** | **R** | **F1** | **P** | **R** | **F1** | **P** | **R** | **F1** | **P** | **R** | **F1** | **support** |
| -1 | 0.81 | 0.79 | 0.80 | 0.70 | 0.98 | 0.82 | 0.91 | 0.84 | 0.86 | 0.96 | 0.88 | 0.92 | 2626 |
| 1 | 0.58 | 0.62 | 0.60 | 0.75 | 0.11 | 0.20 | 0.74 | 0.89 | 0.80 | 0.79 | 0.92 | 0.85 | 1235 |
| accuracy | 0.74 | | | 0.70 | | | 0.86 | | | 0.89 | | | 3861 |
| macro avg | 0.70 | 0.70 | 0.70 | 0.72 | 0.55 | 0.51 | 0.82 | 0.87 | 0.83 | 0.87 | 0.90 | 0.88 | 3861 |
| weighted avg | 0.74 | 0.74 | 0.74 | 0.72 | 0.70 | 0.62 | 0.85 | 0.86 | 0.84 | 0.90 | 0.89 | 0.90 | 3861 |
| AUC-ROC | 0.70 | | | 0.73 | | | 0.89 | | | 0.96 | | | |
| **satimage-2** | **P** | **R** | **F1** | **P** | **R** | **F1** | **P** | **R** | **F1** | **P** | **R** | **F1** | **support** |
| -1 | 1.00 | 1.00 | 1.00 | 1.00 | 0.93 | 0.96 | 0.97 | 0.93 | 0.96 | 1.00 | 0.98 | 0.99 | 3440 |

**Table 9 continued from previous page**

| | DTE | | | Rejex | | | ADGym | | | Swift Hydra | | | |
|---|---|---|---|---|---|---|---|---|---|---|---|---|---|
| 1 | 0.80 | 0.83 | 0.81 | 0.13 | 0.86 | 0.22 | 0.30 | 0.86 | 0.46 | 0.36 | 0.91 | 0.52 | 42 |
| accuracy | | 1.00 | | | 0.93 | | | 0.93 | | | 0.98 | | 3482 |
| macro avg | 0.90 | 0.92 | 0.91 | 0.56 | 0.89 | 0.59 | 0.63 | 0.89 | 0.71 | 0.68 | 0.94 | 0.75 | 3482 |
| weighted avg | 1.00 | 1.00 | 1.00 | 0.99 | 0.93 | 0.95 | 0.96 | 0.93 | 0.95 | 0.99 | 0.98 | 0.98 | 3482 |
| AUC-ROC | | 0.99 | | | 0.97 | | | 0.94 | | | 0.99 | | |
| **shuttle** | **P** | **R** | **F1** | **P** | **R** | **F1** | **P** | **R** | **F1** | **P** | **R** | **F1** | **support** |
| -1 | 0.99 | 1.00 | 1.00 | 0.94 | 1.00 | 0.96 | 0.97 | 0.96 | 0.94 | 1.00 | 1.00 | 1.00 | 27388 |
| 1 | 0.95 | 0.92 | 0.93 | 0.55 | 0.09 | 0.15 | 0.91 | 0.93 | 0.92 | 0.95 | 1.00 | 0.98 | 2071 |
| accuracy | | 0.99 | | | 0.93 | | | 0.96 | | | 1.00 | | 29459 |
| macro avg | 0.97 | 0.96 | 0.96 | 0.74 | 0.54 | 0.56 | 0.94 | 0.94 | 0.93 | 0.98 | 1.00 | 0.99 | 29459 |
| weighted avg | 0.99 | 0.99 | 0.99 | 0.91 | 0.93 | 0.91 | 0.96 | 0.96 | 0.94 | 1.00 | 1.00 | 1.00 | 29459 |
| AUC-ROC | | 1.00 | | | 0.99 | | | 0.95 | | | 1.00 | | |
| **skin** | **P** | **R** | **F1** | **P** | **R** | **F1** | **P** | **R** | **F1** | **P** | **R** | **F1** | **support** |
| -1 | 0.76 | 0.76 | 0.76 | 0.79 | 0.79 | 0.79 | 0.94 | 0.94 | 0.94 | 1.00 | 1.00 | 1.00 | 116448 |
| 1 | 0.09 | 0.09 | 0.09 | 0.19 | 0.18 | 0.18 | 0.95 | 0.94 | 0.96 | 1.00 | 1.00 | 1.00 | 30587 |
| accuracy | | 0.62 | | | 0.66 | | | 0.94 | | | 1.00 | | 147035 |
| macro avg | 0.42 | 0.43 | 0.42 | 0.49 | 0.49 | 0.49 | 0.95 | 0.94 | 0.95 | 1.00 | 1.00 | 1.00 | 147035 |
| weighted avg | 0.62 | 0.62 | 0.62 | 0.66 | 0.66 | 0.66 | 0.94 | 0.94 | 0.94 | 1.00 | 1.00 | 1.00 | 147035 |
| AUC-ROC | | 0.68 | | | 0.74 | | | 0.94 | | | 1.00 | | |
| **smtp** | **P** | **R** | **F1** | **P** | **R** | **F1** | **P** | **R** | **F1** | **P** | **R** | **F1** | **support** |
| -1 | 1.00 | 1.00 | 1.00 | 1.00 | 0.68 | 0.81 | 0.95 | 0.85 | 0.91 | 1.00 | 0.90 | 0.95 | 57075 |
| 1 | 0.00 | 0.00 | 0.00 | 0.00 | 1.00 | 0.00 | 0.00 | 0.81 | 0.00 | 0.00 | 0.84 | 0.01 | 19 |
| accuracy | | 1.00 | | | 0.68 | | | 0.85 | | | 0.90 | | 57094 |
| macro avg | 0.50 | 0.50 | 0.50 | 0.50 | 0.84 | 0.41 | 0.48 | 0.83 | 0.45 | 0.50 | 0.87 | 0.48 | 57094 |
| weighted avg | 1.00 | 1.00 | 1.00 | 1.00 | 0.68 | 0.81 | 0.95 | 0.85 | 0.91 | 1.00 | 0.90 | 0.95 | 57094 |
| AUC-ROC | | 0.92 | | | 0.95 | | | 0.81 | | | 0.85 | | |
| **SpamBase** | **P** | **R** | **F1** | **P** | **R** | **F1** | **P** | **R** | **F1** | **P** | **R** | **F1** | **support** |
| -1 | 0.65 | 0.67 | 0.66 | 0.58 | 0.64 | 0.61 | 0.92 | 0.81 | 0.86 | 0.96 | 0.86 | 0.91 | 1513 |
| 1 | 0.49 | 0.47 | 0.48 | 0.37 | 0.32 | 0.34 | 0.77 | 0.89 | 0.84 | 0.82 | 0.95 | 0.88 | 1012 |
| accuracy | | 0.59 | | | 0.51 | | | 0.84 | | | 0.89 | | 2525 |
| macro avg | 0.57 | 0.57 | 0.57 | 0.48 | 0.48 | 0.48 | 0.84 | 0.85 | 0.85 | 0.89 | 0.90 | 0.89 | 2525 |
| weighted avg | 0.59 | 0.59 | 0.59 | 0.50 | 0.51 | 0.50 | 0.86 | 0.84 | 0.85 | 0.90 | 0.89 | 0.89 | 2525 |
| AUC-ROC | | 0.61 | | | 0.51 | | | 0.92 | | | 0.95 | | |
| **speech** | **P** | **R** | **F1** | **P** | **R** | **F1** | **P** | **R** | **F1** | **P** | **R** | **F1** | **support** |
| -1 | 0.98 | 0.99 | 0.99 | 0.99 | 0.13 | 0.23 | 0.93 | 0.49 | 0.67 | 1.00 | 0.53 | 0.70 | 2177 |
| 1 | 0.00 | 0.00 | 0.00 | 0.02 | 0.91 | 0.03 | 0.00 | 0.83 | 0.01 | 0.03 | 0.89 | 0.06 | 35 |
| accuracy | | 0.97 | | | 0.15 | | | 0.49 | | | 0.54 | | 2212 |
| macro avg | 0.49 | 0.49 | 0.49 | 0.50 | 0.52 | 0.13 | 0.47 | 0.66 | 0.34 | 0.51 | 0.71 | 0.38 | 2212 |
| weighted avg | 0.97 | 0.97 | 0.97 | 0.97 | 0.15 | 0.23 | 0.92 | 0.49 | 0.65 | 0.98 | 0.54 | 0.69 | 2212 |
| AUC-ROC | | 0.47 | | | 0.47 | | | 0.67 | | | 0.73 | | |
| **Stamps** | **P** | **R** | **F1** | **P** | **R** | **F1** | **P** | **R** | **F1** | **P** | **R** | **F1** | **support** |

**Table 9 continued from previous page**

| | DTE | | | Rejex | | | ADGym | | | Swift Hydra | | | support |
|---|---|---|---|---|---|---|---|---|---|---|---|---|---|
| -1 | 0.97 | 0.90 | 0.93 | 0.00 | 0.00 | 0.00 | 0.96 | 0.79 | 0.87 | 1.00 | 0.86 | 0.92 | 185 |
| 1 | 0.41 | 0.68 | 0.51 | 0.09 | 1.00 | 0.17 | 0.39 | 0.94 | 0.54 | 0.42 | 1.00 | 0.59 | 19 |
| accuracy | | 0.88 | | | 0.09 | | | 0.81 | | | 0.87 | | 204 |
| macro avg | 0.69 | 0.79 | 0.72 | 0.05 | 0.50 | 0.09 | 0.67 | 0.87 | 0.70 | 0.71 | 0.93 | 0.76 | 204 |
| weighted avg | 0.91 | 0.88 | 0.89 | 0.01 | 0.09 | 0.02 | 0.91 | 0.81 | 0.84 | 0.95 | 0.87 | 0.89 | 204 |
| AUC-ROC | | 0.89 | | | 0.74 | | | 0.93 | | | 0.96 | | |
| **thyroid** | **P** | **R** | **F1** | **P** | **R** | **F1** | **P** | **R** | **F1** | **P** | **R** | **F1** | **support** |
| -1 | 0.99 | 0.99 | 0.99 | 1.00 | 0.88 | 0.94 | 0.96 | 0.93 | 0.92 | 1.00 | 0.97 | 0.99 | 2207 |
| 1 | 0.63 | 0.65 | 0.64 | 0.18 | 0.98 | 0.30 | 0.43 | 0.97 | 0.60 | 0.49 | 1.00 | 0.66 | 57 |
| accuracy | | 0.98 | | | 0.88 | | | 0.93 | | | 0.97 | | 2264 |
| macro avg | 0.81 | 0.82 | 0.81 | 0.59 | 0.93 | 0.62 | 0.69 | 0.95 | 0.76 | 0.75 | 0.99 | 0.82 | 2264 |
| weighted avg | 0.98 | 0.98 | 0.98 | 0.98 | 0.88 | 0.92 | 0.94 | 0.93 | 0.91 | 0.99 | 0.97 | 0.98 | 2264 |
| AUC-ROC | | 0.98 | | | 0.99 | | | 0.94 | | | 1.00 | | |
| **vertebral** | **P** | **R** | **F1** | **P** | **R** | **F1** | **P** | **R** | **F1** | **P** | **R** | **F1** | **support** |
| -1 | 0.85 | 0.86 | 0.86 | 0.00 | 0.00 | 0.00 | 0.90 | 0.73 | 0.81 | 0.93 | 0.77 | 0.84 | 123 |
| 1 | 0.11 | 0.10 | 0.10 | 0.15 | 1.00 | 0.26 | 0.29 | 0.63 | 0.40 | 0.33 | 0.67 | 0.44 | 21 |
| accuracy | | 0.75 | | | 0.15 | | | 0.72 | | | 0.76 | | 144 |
| macro avg | 0.48 | 0.48 | 0.48 | 0.07 | 0.50 | 0.13 | 0.59 | 0.68 | 0.60 | 0.63 | 0.72 | 0.64 | 144 |
| weighted avg | 0.74 | 0.75 | 0.75 | 0.02 | 0.15 | 0.04 | 0.81 | 0.72 | 0.75 | 0.84 | 0.76 | 0.79 | 144 |
| AUC-ROC | | 0.42 | | | 0.37 | | | 0.72 | | | 0.79 | | |
| **vowels** | **P** | **R** | **F1** | **P** | **R** | **F1** | **P** | **R** | **F1** | **P** | **R** | **F1** | **support** |
| -1 | 0.97 | 0.98 | 0.98 | 1.00 | 0.69 | 0.82 | 0.95 | 0.80 | 0.89 | 1.00 | 0.86 | 0.92 | 844 |
| 1 | 0.26 | 0.17 | 0.20 | 0.10 | 0.93 | 0.18 | 0.15 | 0.89 | 0.27 | 0.19 | 0.93 | 0.32 | 30 |
| accuracy | | 0.96 | | | 0.70 | | | 0.80 | | | 0.86 | | 874 |
| macro avg | 0.62 | 0.58 | 0.59 | 0.55 | 0.81 | 0.50 | 0.55 | 0.84 | 0.58 | 0.59 | 0.90 | 0.62 | 874 |
| weighted avg | 0.95 | 0.96 | 0.95 | 0.97 | 0.70 | 0.79 | 0.92 | 0.80 | 0.87 | 0.97 | 0.86 | 0.90 | 874 |
| AUC-ROC | | 0.76 | | | 0.91 | | | 0.92 | | | 0.97 | | |
| **Waveform** | **P** | **R** | **F1** | **P** | **R** | **F1** | **P** | **R** | **F1** | **P** | **R** | **F1** | **support** |
| -1 | 0.97 | 0.97 | 0.97 | 0.97 | 0.53 | 0.69 | 0.96 | 0.75 | 0.83 | 1.00 | 0.82 | 0.90 | 2008 |
| 1 | 0.04 | 0.03 | 0.04 | 0.03 | 0.48 | 0.05 | 0.09 | 0.81 | 0.16 | 0.12 | 0.88 | 0.21 | 58 |
| accuracy | | 0.95 | | | 0.53 | | | 0.75 | | | 0.82 | | 2066 |
| macro avg | 0.50 | 0.50 | 0.50 | 0.50 | 0.51 | 0.37 | 0.52 | 0.78 | 0.49 | 0.56 | 0.85 | 0.55 | 2066 |
| weighted avg | 0.95 | 0.95 | 0.95 | 0.95 | 0.53 | 0.67 | 0.93 | 0.75 | 0.81 | 0.97 | 0.82 | 0.88 | 2066 |
| AUC-ROC | | 0.78 | | | 0.54 | | | 0.83 | | | 0.90 | | |
| **WBC** | **P** | **R** | **F1** | **P** | **R** | **F1** | **P** | **R** | **F1** | **P** | **R** | **F1** | **support** |
| -1 | 0.99 | 0.99 | 0.99 | 0.00 | 0.00 | 0.00 | 0.94 | 0.93 | 0.94 | 1.00 | 0.98 | 0.99 | 128 |
| 1 | 0.83 | 0.83 | 0.83 | 0.05 | 1.00 | 0.09 | 0.70 | 0.95 | 0.79 | 0.75 | 1.00 | 0.86 | 6 |
| accuracy | | 0.99 | | | 0.05 | | | 0.93 | | | 0.99 | | 134 |
| macro avg | 0.91 | 0.91 | 0.91 | 0.02 | 0.50 | 0.04 | 0.82 | 0.94 | 0.87 | 0.88 | 0.99 | 0.93 | 134 |
| weighted avg | 0.99 | 0.99 | 0.99 | 0.00 | 0.05 | 0.00 | 0.92 | 0.93 | 0.93 | 0.99 | 0.99 | 0.99 | 134 |
| AUC-ROC | | 0.99 | | | 0.90 | | | 0.94 | | | 1.00 | | |

**Table 9 continued from previous page**

| | | DTE | | | Rejex | | | ADGym | | | Swift Hydra | | | |
|---|---|---|---|---|---|---|---|---|---|---|---|---|---|---|
| **WDBC** | P | R | F1 | P | R | F1 | P | R | F1 | P | R | F1 | support |
| -1 | 0.99 | 0.99 | 0.99 | 0.98 | 0.19 | 0.32 | 0.96 | 0.89 | 0.91 | 1.00 | 0.93 | 0.96 | 217 |
| 1 | 0.25 | 0.25 | 0.25 | 0.02 | 0.75 | 0.03 | 0.18 | 0.95 | 0.29 | 0.21 | 1.00 | 0.35 | 4 |
| accuracy | | 0.97 | | | 0.20 | | | 0.90 | | | 0.93 | | 221 |
| macro avg | 0.62 | 0.62 | 0.62 | 0.50 | 0.47 | 0.18 | 0.57 | 0.92 | 0.60 | 0.61 | 0.97 | 0.66 | 221 |
| weighted avg | 0.97 | 0.97 | 0.97 | 0.96 | 0.20 | 0.32 | 0.95 | 0.90 | 0.90 | 0.99 | 0.93 | 0.95 | 221 |
| AUC-ROC | | 0.98 | | | 0.51 | | | 0.93 | | | 1.00 | | |
| **Wilt** | P | R | F1 | P | R | F1 | P | R | F1 | P | R | F1 | support |
| -1 | 0.95 | 0.95 | 0.95 | 0.98 | 0.77 | 0.86 | 0.93 | 0.87 | 0.91 | 1.00 | 0.92 | 0.96 | 2746 |
| 1 | 0.01 | 0.01 | 0.01 | 0.14 | 0.68 | 0.23 | 0.35 | 0.93 | 0.50 | 0.38 | 0.96 | 0.55 | 146 |
| accuracy | | 0.90 | | | 0.77 | | | 0.88 | | | 0.92 | | 2892 |
| macro avg | 0.48 | 0.48 | 0.48 | 0.56 | 0.72 | 0.54 | 0.64 | 0.90 | 0.70 | 0.69 | 0.94 | 0.75 | 2892 |
| weighted avg | 0.90 | 0.90 | 0.90 | 0.94 | 0.77 | 0.83 | 0.90 | 0.88 | 0.89 | 0.97 | 0.92 | 0.94 | 2892 |
| AUC-ROC | | 0.42 | | | 0.79 | | | 0.94 | | | 0.98 | | |
| **wine** | P | R | F1 | P | R | F1 | P | R | F1 | P | R | F1 | support |
| -1 | 0.96 | 0.95 | 0.95 | 0.00 | 0.00 | 0.00 | 0.95 | 0.91 | 0.92 | 1.00 | 0.95 | 0.97 | 73 |
| 1 | 0.33 | 0.40 | 0.36 | 0.06 | 1.00 | 0.12 | 0.52 | 0.97 | 0.67 | 0.56 | 1.00 | 0.71 | 5 |
| accuracy | | 0.91 | | | 0.06 | | | 0.91 | | | 0.95 | | 78 |
| macro avg | 0.65 | 0.67 | 0.66 | 0.03 | 0.50 | 0.06 | 0.74 | 0.94 | 0.79 | 0.78 | 0.97 | 0.84 | 78 |
| weighted avg | 0.92 | 0.91 | 0.91 | 0.00 | 0.06 | 0.01 | 0.92 | 0.91 | 0.90 | 0.97 | 0.95 | 0.96 | 78 |
| AUC-ROC | | 0.84 | | | 0.35 | | | 0.95 | | | 0.99 | | |
| **WPBC** | P | R | F1 | P | R | F1 | P | R | F1 | P | R | F1 | support |
| -1 | 0.78 | 0.86 | 0.82 | 0.00 | 0.00 | 0.00 | 0.76 | 0.43 | 0.55 | 0.83 | 0.47 | 0.60 | 93 |
| 1 | 0.24 | 0.15 | 0.19 | 0.22 | 1.00 | 0.36 | 0.20 | 0.62 | 0.33 | 0.26 | 0.65 | 0.37 | 26 |
| accuracy | | 0.71 | | | 0.22 | | | 0.47 | | | 0.51 | | 119 |
| macro avg | 0.51 | 0.51 | 0.50 | 0.11 | 0.50 | 0.18 | 0.48 | 0.52 | 0.44 | 0.54 | 0.56 | 0.49 | 119 |
| weighted avg | 0.66 | 0.71 | 0.68 | 0.05 | 0.22 | 0.08 | 0.64 | 0.47 | 0.50 | 0.71 | 0.51 | 0.55 | 119 |
| AUC-ROC | | 0.54 | | | 0.53 | | | 0.54 | | | 0.59 | | |
| **yeast** | P | R | F1 | P | R | F1 | P | R | F1 | P | R | F1 | support |
| -1 | 0.60 | 0.61 | 0.60 | 0.64 | 0.87 | 0.74 | 0.77 | 0.49 | 0.62 | 0.80 | 0.54 | 0.65 | 587 |
| 1 | 0.22 | 0.21 | 0.22 | 0.19 | 0.06 | 0.09 | 0.40 | 0.68 | 0.53 | 0.46 | 0.74 | 0.57 | 304 |
| accuracy | | 0.47 | | | 0.60 | | | 0.56 | | | 0.61 | | 891 |
| macro avg | 0.41 | 0.41 | 0.41 | 0.42 | 0.47 | 0.42 | 0.59 | 0.59 | 0.57 | 0.63 | 0.64 | 0.61 | 891 |
| weighted avg | 0.47 | 0.47 | 0.47 | 0.49 | 0.60 | 0.52 | 0.64 | 0.56 | 0.59 | 0.69 | 0.61 | 0.62 | 891 |
| AUC-ROC | | 0.37 | | | 0.41 | | | 0.65 | | | 0.70 | | |
| **CIFAR10** | P | R | F1 | P | R | F1 | P | R | F1 | P | R | F1 | support |
| -1 | 0.97 | 0.84 | 0.90 | 0.98 | 1.00 | 0.99 | 0.98 | 0.87 | 0.92 | 0.99 | 0.99 | 0.99 | 2812 |
| 1 | 0.29 | 0.74 | 0.42 | 0.80 | 0.52 | 0.63 | 0.34 | 0.75 | 0.47 | 0.79 | 0.79 | 0.79 | 106 |
| accuracy | | 0.83 | | | 0.98 | | | 0.86 | | | 0.98 | | 2918 |
| macro avg | 0.63 | 0.79 | 0.66 | 0.89 | 0.76 | 0.81 | 0.66 | 0.81 | 0.69 | 0.89 | 0.89 | 0.89 | 2918 |
| weighted avg | 0.92 | 0.83 | 0.86 | 0.98 | 0.98 | 0.98 | 0.92 | 0.86 | 0.88 | 0.98 | 0.98 | 0.98 | 2918 |

**Table 9 continued from previous page**

| | | DTE | | | Rejex | | | ADGym | | | Swift Hydra | | | |
|---|---|---|---|---|---|---|---|---|---|---|---|---|---|---|
| AUC-ROC | | 0.88 | | | 0.76 | | | 0.90 | | | 0.98 | | | |
| **FashionMNIST** | P | R | F1 | P | R | F1 | P | R | F1 | P | R | F1 | support |
| -1 | 0.99 | 0.85 | 0.92 | 0.99 | 1.00 | 0.99 | 0.99 | 0.93 | 0.96 | 1.00 | 1.00 | 1.00 | 3441 |
| 1 | 0.35 | 0.94 | 0.52 | 0.85 | 0.70 | 0.77 | 0.51 | 0.92 | 0.66 | 0.93 | 0.91 | 0.92 | 92 |
| accuracy | | 0.85 | | | 0.99 | | | 0.93 | | | 0.99 | | | |
| macro avg | 0.67 | 0.89 | 0.71 | 0.92 | 0.85 | 0.88 | 0.73 | 0.92 | 0.81 | 0.97 | 0.95 | 0.96 | 3533 |
| weighted avg | 0.94 | 0.86 | 0.88 | 0.99 | 0.99 | 0.99 | 0.95 | 0.92 | 0.93 | 0.99 | 0.99 | 0.99 | 3533 |
| AUC-ROC | | 0.96 | | | 0.85 | | | 0.97 | | | 0.99 | | | |
| **MNIST-C** | P | R | F1 | P | R | F1 | P | R | F1 | P | R | F1 | support |
| -1 | 0.60 | 0.61 | 0.60 | 0.64 | 0.87 | 0.74 | 0.77 | 0.49 | 0.62 | 0.80 | 0.54 | 0.65 | 587 |
| 1 | 0.22 | 0.21 | 0.22 | 0.19 | 0.06 | 0.09 | 0.40 | 0.68 | 0.53 | 0.46 | 0.74 | 0.57 | 304 |
| accuracy | | 0.47 | | | 0.60 | | | 0.56 | | | 0.61 | | | 891 |
| macro avg | 0.41 | 0.41 | 0.41 | 0.42 | 0.47 | 0.42 | 0.59 | 0.59 | 0.57 | 0.63 | 0.64 | 0.61 | 891 |
| weighted avg | 0.47 | 0.47 | 0.47 | 0.49 | 0.60 | 0.52 | 0.64 | 0.56 | 0.59 | 0.69 | 0.61 | 0.62 | 891 |
| AUC-ROC | | 0.37 | | | 0.41 | | | 0.65 | | | 0.70 | | | |
| **MVTec-AD** | P | R | F1 | P | R | F1 | P | R | F1 | P | R | F1 | support |
| -1 | 0.60 | 0.61 | 0.60 | 0.64 | 0.87 | 0.74 | 0.77 | 0.49 | 0.62 | 0.80 | 0.54 | 0.65 | 587 |
| 1 | 0.22 | 0.21 | 0.22 | 0.19 | 0.06 | 0.09 | 0.40 | 0.68 | 0.53 | 0.46 | 0.74 | 0.57 | 304 |
| accuracy | | 0.47 | | | 0.60 | | | 0.56 | | | 0.61 | | | 891 |
| macro avg | 0.41 | 0.41 | 0.41 | 0.42 | 0.47 | 0.42 | 0.59 | 0.59 | 0.57 | 0.63 | 0.64 | 0.61 | 891 |
| weighted avg | 0.47 | 0.47 | 0.47 | 0.49 | 0.60 | 0.52 | 0.64 | 0.56 | 0.59 | 0.69 | 0.61 | 0.62 | 891 |
| AUC-ROC | | 0.37 | | | 0.41 | | | 0.65 | | | 0.70 | | | |
| **SVHN** | P | R | F1 | P | R | F1 | P | R | F1 | P | R | F1 | support |
| -1 | 0.92 | 0.87 | 0.90 | 0.95 | 0.98 | 0.97 | 0.92 | 0.89 | 0.91 | 0.96 | 0.98 | 0.97 | 2562 |
| 1 | 0.10 | 0.16 | 0.12 | 0.08 | 0.03 | 0.04 | 0.12 | 0.16 | 0.13 | 0.26 | 0.16 | 0.02 | 133 |
| accuracy | | 0.82 | | | 0.94 | | | 0.83 | | | 0.93 | | | 2695 |
| macro avg | 0.51 | 0.51 | 0.51 | 0.52 | 0.51 | 0.51 | 0.52 | 0.53 | 0.52 | 0.61 | 0.57 | 0.58 | 2695 |
| weighted avg | 0.86 | 0.82 | 0.83 | 0.91 | 0.94 | 0.92 | 0.86 | 0.83 | 0.84 | 0.92 | 0.93 | 0.93 | 2695 |
| AUC-ROC | | 0.37 | | | 0.51 | | | 0.55 | | | 0.63 | | | |
| **Agnews** | P | R | F1 | P | R | F1 | P | R | F1 | P | R | F1 | support |
| -1 | 0.95 | 0.90 | 0.93 | 0.95 | 0.97 | 0.96 | 0.94 | 0.96 | 0.95 | 0.98 | 0.99 | 0.99 | 5383 |
| 1 | 0.30 | 0.50 | 0.38 | 0.10 | 0.06 | 0.07 | 0.46 | 0.39 | 0.42 | 0.75 | 0.69 | 0.72 | 277 |
| accuracy | | 0.87 | | | 0.93 | | | 0.91 | | | 0.97 | | | 5660 |
| macro avg | 0.63 | 0.70 | 0.65 | 0.52 | 0.51 | 0.52 | 0.70 | 0.67 | 0.69 | 0.87 | 0.84 | 0.85 | 5660 |
| weighted avg | 0.90 | 0.87 | 0.88 | 0.91 | 0.93 | 0.92 | 0.91 | 0.91 | 0.91 | 0.97 | 0.97 | 0.97 | 5660 |
| AUC-ROC | | 0.79 | | | 0.51 | | | 0.80 | | | 0.90 | | | |
| **Amazon** | P | R | F1 | P | R | F1 | P | R | F1 | P | R | F1 | support |
| -1 | 0.92 | 0.94 | 0.93 | 0.95 | 0.97 | 0.96 | 0.92 | 0.94 | 0.93 | 0.97 | 0.98 | 0.98 | 5393 |
| 1 | 0.11 | 0.09 | 0.10 | 0.06 | 0.04 | 0.05 | 0.07 | 0.05 | 0.06 | 0.54 | 0.36 | 0.43 | 288 |
| accuracy | | 0.87 | | | 0.92 | | | 0.86 | | | 0.95 | | | 5681 |
| macro avg | 0.52 | 0.51 | 0.51 | 0.50 | 0.50 | 0.50 | 0.50 | 0.50 | 0.51 | 0.75 | 0.67 | 0.70 | 5681 |

**Table 9 continued from previous page**

| | DTE | | | Rejex | | | ADGym | | | Swift Hydra | | | support |
|---|---|---|---|---|---|---|---|---|---|---|---|---|---|
| weighted avg | 0.86 | 0.87 | 0.86 | 0.90 | 0.92 | 0.91 | 0.95 | 0.86 | 0.86 | 0.95 | 0.95 | 0.95 | 5681 |
| AUC-ROC | 0.58 | | | 0.50 | | | 0.53 | | | 0.93 | | | |
| **Imdb** | P | R | F1 | P | R | F1 | P | R | F1 | P | R | F1 | support |
| -1 | 0.92 | 0.94 | 0.93 | 0.95 | 0.97 | 0.96 | 0.92 | 0.95 | 0.94 | 0.97 | 0.98 | 0.98 | 5389 |
| 1 | 0.09 | 0.07 | 0.08 | 0.06 | 0.04 | 0.05 | 0.09 | 0.05 | 0.06 | 0.54 | 0.36 | 0.43 | 290 |
| accuracy | 0.87 | | | 0.92 | | | 0.88 | | | 0.95 | | | 5679 |
| macro avg | 0.51 | 0.51 | 0.51 | 0.51 | 0.50 | 0.50 | 0.50 | 0.51 | 0.50 | 0.75 | 0.67 | 0.70 | 5679 |
| weighted avg | 0.85 | 0.87 | 0.86 | 0.90 | 0.92 | 0.91 | 0.85 | 0.88 | 0.87 | 0.95 | 0.95 | 0.95 | 5679 |
| AUC-ROC | 0.55 | | | 0.50 | | | 0.55 | | | 0.93 | | | |
| **Yelp** | P | R | F1 | P | R | F1 | P | R | F1 | P | R | F1 | support |
| -1 | 0.92 | 0.94 | 0.93 | 0.95 | 0.96 | 0.96 | 0.89 | 0.96 | 0.93 | 0.98 | 0.98 | 0.98 | 5369 |
| 1 | 0.09 | 0.07 | 0.09 | 0.03 | 0.02 | 0.03 | 0.25 | 0.09 | 0.13 | 0.66 | 0.65 | 0.66 | 283 |
| accuracy | 0.89 | | | 0.92 | | | 0.86 | | | 0.97 | | | 5652 |
| macro avg | 0.53 | 0.52 | 0.52 | 0.49 | 0.49 | 0.49 | 0.57 | 0.53 | 0.53 | 0.82 | 0.82 | 0.82 | 891 |
| weighted avg | 0.86 | 0.89 | 0.87 | 0.90 | 0.92 | 0.91 | 0.81 | 0.86 | 0.83 | 0.97 | 0.97 | 0.97 | 5652 |
| AUC-ROC | 0.58 | | | 0.49 | | | 0.62 | | | 0.94 | | | |
| **20newsgroups** | P | R | F1 | P | R | F1 | P | R | F1 | P | R | F1 | support |
| -1 | 0.97 | 0.40 | 0.57 | 0.95 | 1.00 | 0.97 | 0.95 | 0.68 | 0.79 | 0.98 | 0.99 | 0.98 | 787 |
| 1 | 0.11 | 0.89 | 0.20 | 0.00 | 0.00 | 0.00 | 0.23 | 0.71 | 0.34 | 0.74 | 0.57 | 0.64 | 39 |
| accuracy | 0.44 | | | 0.95 | | | 0.68 | | | 0.61 | | | 826 |
| macro avg | 0.55 | 0.65 | 0.39 | 0.48 | 0.50 | 0.49 | 0.59 | 0.69 | 0.57 | 0.86 | 0.78 | 0.81 | 826 |
| weighted avg | 0.91 | 0.44 | 0.54 | 0.91 | 0.95 | 0.93 | 0.86 | 0.68 | 0.74 | 0.97 | 0.97 | 0.97 | 826 |
| AUC-ROC | 0.76 | | | 0.50 | | | 0.77 | | | 0.94 | | | |
| **P = Precision, R = Recall, F1 = F1-Score** | | | | | | | | | | | | | |

