# OpenReview forum: "Swift Hydra:  Self-Reinforcing Generative Framework for Anomaly Detection with Multiple Mamba Models"
_ICLR.cc/2025/Conference — ICLR 2025 Poster_

### Official Review · Reviewer_uiUH · 2024-10-24

**Soundness:** 3
**Presentation:** 2
**Contribution:** 2
**Rating:** 6
**Confidence:** 3

**Summary:**

This paper proposes Swift Hydra, a novel anomaly detection framework based on the conditional variational autoencoder (CVAE) and reinforcement learning.
The proposed method generates high-quality anomalies using CVAE and reinforcement learning, and performs anomaly detection using a mixture of expert models based on Mamba (MoME).
Experiments on various datasets demonstrated improved anomaly detection performance and reduced inference time compared to existing methods.

**Strengths:**

- This paper introduces several strategies to achieve high-accuracy anomaly detection. For example, the reward function in reinforcement learning incorporates the entropy of the union of the training data and the generated data, which encourages the generation of diverse anomalies. Additionally, in the MMoE model, experts are selected probabilistically to avoid the winner-take-all problem.
- The visualization of the generated anomalies is interesting. It clearly shows a clean separation between normal data and anomalies, and we can also observe the generation of a wide variety of anomalies.

**Weaknesses:**

- Although the proposed method works well, there are a few parts that are unclear to me. I will write more details in the Questions section, so please refer to it.

**Questions:**

1. Why does the proposed method use CVAE? Since VAE generally has lower data generation quality compared to other generative models, I don't think it's very suitable for data augmentation purposes in this case. For example, would the proposed method work well with a diffusion model, which can be considered a special case of VAE?
2. The proposed method performs anomaly detection in the data space, but wouldn’t it be more beneficial to take advantage of CVAE’s strengths by performing anomaly detection in the lower-dimensional latent space? What is the reason for performing anomaly detection in the data space?
3. I understand that MMoE performs binary classification between normal and anomaly data. In binary classification, a decision boundary is set between the normal and anomaly data included in the training set, which makes it difficult to handle unseen anomalies that are far from the known anomalies in the training data. Can the proposed method address this issue?
4. If the proposed method can handle unseen anomalies, I believe it is because CVAE can generate such unseen anomalies. Why is the proposed method able to generate unseen anomalies? Even though the entropy is incorporated into the reward of the reinforcement learning, I think generative models can only generate data similar to the training data.
5. Is it possible to visualize the generated samples and decision boundaries of the proposed method using two-dimensional toy data? For example, if there is normal data shaped like a sine curve, and the generated anomalies are arranged surrounding that curve, I believe the method would be able to handle both seen and unseen anomalies.

---

> ### Author Response · Authors · 2024-11-20
>
> Question 1: Why does the proposed method use CVAE? Since VAE generally has lower data generation quality compared to other generative models, I don’t think it’s very suitable for data augmentation purposes in this case. For example, would the proposed method work well with a diffusion model, which can be considered a special case of VAE?
>
> Answer: Our method is compatible with various generative models, including VAEs, diffusion models, GANs, and more. While we acknowledge that VAEs typically generate outputs of lower quality compared to models like diffusion models or GANs (e.g., VAEs often produce blurred backgrounds in image generation tasks), we chose VAEs due to their significantly simpler training process compared to GANs or diffusion models.

---

> ### Author Response · Authors · 2024-11-20
>
> Question 2: The proposed method performs anomaly detection in the data space, but wouldn’t it
> be more beneficial to take advantage of CVAE’s strengths by performing anomaly detection in the
> lower-dimensional latent space? What is the reason for performing anomaly detection in the data
> space?
>
> Answer: We believe that operating within the data space enables a more intuitive interpretation and
> understanding of anomalies, which is particularly useful in applications where data transparency and
> interpretability are critical. More importantly, predicting performance in the latent space requires
> invoking the Encoder for each individual prediction. This frequent invocation can significantly
> increase inference time, thereby impacting the overall efficiency and real-time capabilities of the
> system. Nonetheless, we agree that leveraging the latent space offers advantages, as it encapsulates
> condensed, distinctive features of normal and anomalous data, which can be utilized for predictions.
> In future work, we will consider this approach as part of a hybrid method, combining data space and
> latent space anomaly detection to achieve a balance between accuracy and efficiency.

---

> ### Author Response · Authors · 2024-11-20
>
> Question 3: I understand that MoME performs binary classification between normal and anomaly
> data. In binary classification, a decision boundary is set between the normal and anomaly data
> included in the training set, which makes it difficult to handle unseen anomalies that are far from the
> known anomalies in the training data. Can the proposed method address this issue?
>
> Answer: We want to clarify that the MoME alone does not fully address the generalization to unseen
> anomalies. Instead, our framework leverages the Self-Reinforcing Generative Module to produce
> diverse and challenging anomalies, potentially even capturing unseen anomaly points (as illustrated
> in Fig. 2, page 8, and further elaborated in Question 4). These novel anomalies allow the MoME to
> learn from a broader range of data and improve its generalization.

---

> ### Author Response · Authors · 2024-11-20
>
> Question 4: If the proposed method can handle unseen anomalies, I believe it is because CVAE can generate such unseen anomalies. Why is the proposed method able to generate unseen anomalies? Even though the entropy is incorporated into the reward of the reinforcement learning, I think generative models can only generate data similar to the training data.
>
> Answer: The proposed method’s ability to generate unseen anomalies beyond those in the training
> set comes from the combined contributions of two key components: C-VAE and RL with entropy
> maximization in the reward function.
>
> For the C-VAE, training on both normal and anomalous data allows it to generate anomalous data
> that not only carries the features of real anomalies in the training set but also blends these with features from normal data. This blending creates more diverse and challenging anomalies, enabling the
> C-VAE to produce unseen anomalies that go beyond the training set and even match many anomalies
> in the test set. Additionally, the degree to which normal features are incorporated into the generated
> anomalies can be controlled by adjusting the weights of the KL divergence and reconstruction loss
> terms in the CVAE’s objective function. Increasing the weight on the reconstruction loss promotes
> diversity by encouraging the generation of data with normal-like features. However, placing too
> much emphasis on reconstruction loss can risk model collapse if these normal-like data are often
> labeled as anomalies. We demonstrate this trade-off in the main manuscript (Appendix C.4).
>
> A key limitation of C-VAE, similar to other generative AI models for creating unseen anomalies, is
> its tendency to focus on and generate anomalies with the most frequently occurring features (i.e.,
> features common among anomalies in the training set). This can result in limited diversity and fail to
> fully represent the range of possible anomalies. In anomaly detection, however, rare features—those
> that combine characteristics of different classes but appear infrequently—are often critical in define ing anomalies. To overcome this limitation, we employ reinforcement learning (RL) with a reward function that incorporates an entropy term. This encourages the generation of a wider variety of anomalies, increasing the chance of producing rare-feature anomalies. These anomalies with rare features are then reintroduced into the dataset, increasing their representation and enabling the generative model to learn and replicate these challenging features, thereby improving the detector’s performance over subsequent iterations.

---

> ### Author Response · Authors · 2024-11-20
>
> Question 5: Is it possible to visualize the generated samples and decision boundaries of the proposed
> method using two-dimensional toy data? For example, if there is normal data shaped like a sine
> curve, and the generated anomalies are arranged surrounding that curve, I believe the method would
> be able to handle both seen and unseen anomalies
>
> Answer: Per your suggestion, we generated data where the normal data follows a sine curve, while
> the anomalies was randomly generated around the sine curve. Note that we generated data for both
> the training set and a similarly structured test set. Additionally, we plotted the generated anomalous
> data from the training set alongside the test set data. Finally, we plotted the decision boundary of
> the MoME (detector) model. As expected, the model was able to generalize well to the anomalies
> in the test set. The visualization is shown in this anonymized link: **https://imgur.com/a/yGJcWdY**
>
> Moreoever, In Figures 2 and 3 of the main manuscript, we visualize the evolution of anomaly generation by Swift Hydra’s self-reinforcing generative model. As training progresses, the generated
> anomalies (blue points) increasingly overlap with the unseen anomalies in the test set (purple points).
> This highlights ability of Self-Reinforcing Generative Model of SwiftHydra to generate unseen
> anomalies. And thus, MoME (the detector of Swift Hydra) can also generalize to both seen and
> unseen anomalies.

---

> ### Comment · Reviewer_uiUH · 2024-11-22
> **Thanks for rebuttal**
>
> Thank you for your rebuttal.
> While I think there is room for improvement in the proposed method,
> the content of the rebuttal is reasonable and the experimental results are good.
> Therefore, I will raise my score to 6.
>
> I have just one comment to make.
> You have uploaded the toy data experiment result on Imgur, but could you include it in the Appendix?
> Since revisions are allowed in ICLR rebuttals, I think that would be a better.

---

> > ### Author Response · Authors · 2024-11-30
> >
> > Dear Reviewer uiUH,
> >
> > We appreciate the time and effort you have contributed to reviewing our paper. Your constructed feedback has been instrumental in improving our work.
> >
> > Per our discussion, we have addressed the weaknesses by clarifying the generalization ability of our method and conducting additional experiments during this rebuttal period, including your suggested toy experiment, to further strengthen the manuscript. Given that a score of 6 is considered "marginally above the acceptance threshold," we believe an adjustment to 8, which corresponds to "accept, good paper" on the ICLR scoring metrics, would more accurately reflect your positive evaluation that our paper introduces new *strategies that achieve high-accuracy*, with *good experimental results* and *interesting visualization insights*.
> >
> > We greatly appreciate your careful evaluation and hope that the improvements made to our paper warrant a higher rating. Thank you once again for your time and constructive dialogue!
> >
> >
> > Sincerely,
> >
> > Authors

---

> ### Author Response · Authors · 2024-11-22
>
> Dear Reviewer uiUH,
>
> Thank you for increasing the rating! We are glad that our response has addressed your concerns.
> We will include the Decision Boundary section in the appendix as you suggested. Thank you for your constructive comments.
>
> Best regards,
>
> Authors

---

### Official Review · Reviewer_8B4d · 2024-10-29

**Soundness:** 3
**Presentation:** 2
**Contribution:** 4
**Rating:** 8
**Confidence:** 2

**Summary:**

This method focus on generalization to unseen anomalies. To do that, this method generates new anomaly samples using reinforcement learning and conditional VAE. The conditional VAE generate samples conditioned on the class anomaly. The reinforcement learning enforces the generated samples to be diverse and difficult to detect by the detection model. For the detection model, in order to keep the complexity low while can capture diverse feature distribution, Mixture of Experts consisting of multiple Mamba experts are utilized.

**Strengths:**

Adding new anomalous synthetic data during training to increase generalization using reinforcement learning seems to be novel.

**Weaknesses:**

See questions.

**Questions:**

1. Which part of the experiments that prove the generalization to unseen anomalies?

2. It is unclear whether this work use both real and fake data or only real data. I can't find the information in Line 94-105. My guess from other part of the paper is using both real and fake, but I am not 100\% sure.

3. Why is this method called reinforcement learning method if the action is (afaik) backpropagatable directly from the reward/loss?

4. How is the balanced dataset updated? I can't find any information in Algorithm 1 in the appendix.

5. Why not also generating normal data?

---

> ### Author Response · Authors · 2024-11-19
>
> Question 1: Which part of the experiments that prove the generalization to unseen anomalies?
>
> Answer: Dear reviewer, Figure 2 shows the trend of generated data points across different episodes,
> while Figure 3 compares the data points generated by Swift Hydra with those produced by other
> oversampling methods. In Figures 2 and 3, we plot the anomalies generated by Swift Hydra (blue
> points) as the generative model learns from the anomalies in the training set (red points). Notably,
> these blue points gradually start to cover the anomalies points in the test set (purple points), even
> though the test set anomalies differ significantly from those in the training set. This demonstrates
> Swift Hydra’s ability to generalize by generating unseen anomalies.

---

> ### Author Response · Authors · 2024-11-19
>
> Question 2: It is unclear whether this work use both real and fake data or only real data. I can’t find
> the information in Line 94-105. My guess from other part of the paper is using both real and fake,
> but I am not 100% sure.
>
> Answer: By fake, did you mean the synthetic/generated data? If so, we confirm that Swift Hydra
> trains the C-VAE on both original (real) anomalies and synthetic/generated (fake) anomalies. This is
> detailed in lines 182-196 of the main manuscript and in the pseudocode of Algorithm 1, specifically
> lines 19 and 21. We utilize a self-reinforcing generative model to produce diverse anomalous data
> that effectively deceive the detector. These generated anomalies are then combined with the original
> anomalous data for training. We avoid training exclusively on synthetic/generated anomalies, as the
> original data more accurately represent the true nature of the problem.

---

> ### Author Response · Authors · 2024-11-19
>
> Question 3: Why is this method called reinforcement learning method if the action is (afaik) back-
> propagatable directly from the reward/loss?
>
> Answer: This is an interesting question. In fact, as demonstrated in Theorem 1, we can indeed find a feasible action by directly optimizing latent space based on the reward function (considering it as a loss function). However, in practice, this optimization is time-consuming. For instance, in each episode, suppose we generate $m$ data points (inferred from actions), and each data point requires $n$ gradient steps; the complexity then becomes $O(m \cdot n)$. As a result, this method is typically reserved for cases where the RL agent makes an error—namely invalid action when the updated latent vector $\hat{z_i}$ derived from predicted action $a_i=\left(\mu_i, \sigma_i\right)$ falls outside the supported range of the trained model $\mathcal{F}_\theta$.
>
>
> In Appendix C.5, we further show that this approach is more critical during the early stages of
> training, when the RL agent’s policy is still underdeveloped and prone to errors. During this phase,
> optimizing the latent space based on the reward function helps identify suitable actions. Over time,
> however, as the RL agent’s policy improves and generalizes, the reliance on this fallback method
> diminishes.

---

> ### Author Response · Authors · 2024-11-19
>
> Question 4: How is the balanced dataset updated? I can’t find any information in Algorithm 1 in
> the appendix.
>
> Answer: Apologies for the lack of clarity in this section. From a coding perspective, we create
> a balanced dataset by trimming the majority class data in the training set to match the size of the
> minority class. This clarification will be added to the pseudocode in Algorithm 1, immediately after
> line 21.
>
> $\mathcal{D}^{balance}_e \gets \text{Trim}(\mathcal{D}^{train}_e)$
>
> *Helper function that randomly trims elements from classes with more data points to equalize the class sizes based on the smallest class.*

---

> > ### Comment · Reviewer_8B4d · 2024-11-20
> >
> > About question 4: Is it possible that the generated data got trimmed out?

---

> ### Author Response · Authors · 2024-11-19
>
> Question 5: Why not also generating normal data?
>
> Answer: The primary challenge in anomaly detection is the limited availability of anomalous data
> for training models. Therefore, we focus on generating additional anomalies to improve detection
> performance. In fact, normal data often vastly outnumber anomalies, and in most systems, real
> normal data can be collected directly from normal operating conditions, thus generating synthetic
> normal data is not a pressing need. This approach may be more relevant in scenarios where normal
> data is expensive, but it is beyond the scope of our current research.

---

> > ### Comment · Reviewer_8B4d · 2024-11-20
> >
> > About question 1 & 2: Apologies since I am not familiar with the dataset. Since original (real) anomalies are used for training, I wonder, is the original anomalous data in training and testing different type of anomalies? If I see Figure 2, it also seems that anomalous data (train) is intermingled with the anomalous data (test), which probably show that both are similar type of anomalies. If there are the similar type of anomalies, is it fair to say that the method is generalized to unseen anomalies?
> >
> > Separate question regarding Fig. 2: what is the light blue circle?

---

> ### Comment · Reviewer_8B4d · 2024-11-20
>
> About Question 3:
> So, the fact that the optimization is time-consuming is what makes it reinforcement learning? And is episode here same as iteration?

---

> ### Author Response · Authors · 2024-11-21
>
> About Question 3: So, the fact that the optimization is time-consuming is what makes it reinforcement learning? And is episode here same as iteration?
>
> Answer: Thanks for your swift response! The short answer is *no* for both questions. Let us clarify the self-reinforcing module: during the **early training episodes**, when the RL agent’s policy is still underdeveloped and is prone to making invalid actions, anomalies are generated via the one-step-to-feasible action method (Algo 2, page 17). Specifically, in each of these early episodes, we solve an optimization problem that finds a latent variable ($\hat{z}$) that maximizes the reward function. This information can be found from line 230 to 236 in the main manuscript.
>
> The one-step-to-feasible action is necessary but *extremely time-consuming* with an $O(mn)$ time complexity *per episode*, where $m$ is the number of generated anomalies and $n$ is the number of gradient *iterations*. Each gradient iteration here involves calculating the reward over the entire anomalous data points $ \mathcal{D}^{train}_{e, \text{anomalous}}$ (Appendix A.3, page 18), implying that as we have more data points in later episodes, the training time per episode will increase significantly.
>
> **After the initial training stage**, the one-step-to-feasible action approach is replaced by reinforcement learning, where an agent selects actions in a single inference step (instead of requiring $n$ gradient iterations). This transition is viable because, during the early stage—when the one-step-to-feasible action method is frequently used to find valid actions—we simultaneously train both the policy model, the value model, and the reward model. This training enables the RL agent to quickly generalize and reduce invalid actions in later stages. As noted in Appendix A.1, we use proximal policy optimization (PPO) to update a neural network that encodes a policy capable of predicting the next actions in a single inference step toward maximizing the reward.
>
> In fact, we note that an analogy can be drawn between our method and AlphaGo [10]. In AlphaGo, the initial training episodes involved training the RL policy network to predict moves based on human expert labels (similar to our one-step-to-feasible action). Once supervised learning was no longer viable (just as it becomes impractical to use the one-step-to-feasible action in later episodes in Swift Hydra), AlphaGo transitioned into a standard reinforcement learning framework to further optimize its moves and surpass human expert levels. Similarly, our method transitions to RL for efficiency in exploring and exploiting state-action pairs in later episodes.
>
> [10] David Silver, Aja Huang, Chris J Maddison, Arthur Guez, Laurent Sifre, George Van Den Driessche,
> Julian Schrittwieser, Ioannis Antonoglou, Veda Panneershelvam, Marc Lanctot, et al. Mastering
> the game of go with deep neural networks and tree search. nature, 529(7587):484–489, 2016.

---

> ### Author Response · Authors · 2024-11-21
>
> About question 4: Is it possible that the generated data got trimmed out?
>
>
>
> Answer: This can only occur when the total number of generated anomalous data combined with the training anomalous data exceeds the total number of normal data in the training set. Across our experiments on all 57 datasets, we observed that Swift Hydra consistently converges before the anomalous data surpasses the normal data in quantity. Note that in most cases, the number of available anomalies in the training data only accounts for 1%-15% of the entire dataset, representing the primary challenge in anomaly detection (i.e., limited availability of anomalous data for training models).
>
> However, if the total anomalous data were to exceed the normal data, one approach would be to start generating (or collecting) more synthetic (or real, resp.) normal data. But doing this is beyond the scope of our research.

---

> ### Author Response · Authors · 2024-11-21
>
> About question 1 & 2: Apologies since I am not familiar with the dataset. Since original (real) anomalies are used for training, I wonder, is the original anomalous data in training and testing different type of anomalies? If I see Figure 2, it also seems that anomalous data (train) is intermingled with the anomalous data (test), which probably show that both are similar type of anomalies. If there are the similar type of anomalies, is it fair to say that the method is generalized to unseen anomalies?
>
> Separate question regarding Fig. 2: what is the light blue circle?
>
> Answer: In Figure 2, the light blue points represent anomalous data points generated from previous episodes, while the heavy blue points indicate the anomalous data points generated in the current episode (we will clarify the legend further in the main manuscript). If you observe Figure 2 closely, particularly in the early episodes, you can discern the distribution of the original anomalous data in the training and test sets.
>
> Specifically, you’ll notice that, aside from some overlap between anomalous data points from the training set (red points) and those from the test set (purple points), there are many anomalous data points in the test set that are not covered by the training set. The red points are clustered together, while the purple points are more scattered across the space.
>
> However, by episode 128, the total generated anomalous data points from all episodes (light and heavy blue points) almost entirely cover the purple points (the anomalous data points in the test set). Therefore, it is reasonable to conclude that the method generalizes well to unseen anomalies.

---

> > ### Comment · Reviewer_8B4d · 2024-11-22
> >
> > Thank you for a nice rebuttal. I am increasing my rating.

---

> ### Author Response · Authors · 2024-11-22
>
> Dear Reviewer 8B4d,
>
> We are delighted to have addressed your questions.
> Thank you for appreciating our work and raising the score from 6 to 8.
>
> Best regards,
>
> Authors

---

### Official Review · Reviewer_9izT · 2024-10-29

**Soundness:** 3
**Presentation:** 3
**Contribution:** 3
**Rating:** 6
**Confidence:** 4

**Summary:**

This paper proposes a novel tabular anomaly detection method by designing a RL-based anomaly data augmentation technique and a Mixture of Experts (MOE) based anomaly detection module. The authors verified the effectiveness of their proposed method with 3 baselines on ADBench and also provided visualized proofs of the effectiveness of their reward function and anomalous data augmentation technique.

**Strengths:**

1. Overall well written. The paper is easy to follow and organized in a clear orchestration.
2. Novel idea. This paper proposes a novel idea that using RL to make anomalous data augmentation. Besides, the authors also prove the effectiveness of their proposed reward function by showing the consistency between AUC-ROC and the reward.
3. It is proved that the RL-based anomaly data generation method can generate anomalies of high diversities, as show in Fig 2.

**Weaknesses:**

1. Weak soundness of experiment. The experiments are not persuasive enough. The baselines do not cover enough SOTA tabular anomaly detection methods. The authors only compares the proposed methods with 3 baselines. Besides, there is no ablation study to verify the effectiveness of RL-based anomaly data generation technique and the MOE anomaly detection module separately.
2. Unrealistic assumption in mathematical proof. In the mathematical proof of theorem 2 and theorem 3, it is actually assumed that single anomaly detector can not distinguish anomalies and normality in overlap region. However, this assumption is only available for linear classification methods (e.g. support vector machine). The deep neural networks are non-linear classification methods and can deal with interlaced anomalies and normality in  overlap region well. In this situation, theorem 2 and 3 can not be deduced.
3. The illustration of the process of "gating network" and "Tackling winner-take-all" is somewhat ambiguous. What confused me is that given both of these two mechanism actually are used to assign input to different expert, how could we use them together?

**Questions:**

1. In theorem 1, do you mean $\hat{x}_i=\mathcal{M}_\phi(\hat{z}_i,y_i=1)$?
2. Could you please add more SOTA tabular anomaly detection methods?
3. Could you please add ablation experiments?
4. In Table 1, why Swift Hydra (MOME) has less inference time than Swift Hydra (Single)?
5. Could you please further explain how to use "gating network" and "Tackling winner-take-all" together?

---

> ### Author Response · Authors · 2024-11-19
>
> Question 1: In theorem 1, do you mean $\hat{x}_i=\mathcal{M}\phi(\hat{z}_i,y_i=1)$?
>
> Answer: Yes, thank you for pointing that out. That was a typo on our part. We will correct it in the main paper.

---

> ### Author Response · Authors · 2024-11-19
>
> Question 2: Could you please add more SOTA tabular anomaly detection methods?
>
> Answer:
> | Method                     |  ECOD  |        | DevNet |        | PReNet |         | DeepSAD |        | FEAWAD |         | FTTransformer |        | SwifHydra (MoME) |       |
> |----------------------------|:------:|--------|:------:|--------|:------:|---------|:-------:|--------|:------:|---------|:-------------:|--------|:----------------:|:-----:|
> |                            | AUCROC |   TIF  | AUCROC |   TIF  | AUCROC |   TIF   |  AUCROC |   TIF  | AUCROC |   TIF   |     AUCROC    |   TIF  |      AUCROC      |  TIF  |
> | Train/Test Ratio (40/60\%) |  0.803 | 26.002 |  0.822 | 24.084 |  0.853 | 128.709 |  0.843  | 19.020 |  0.856 | 168.615 |     0.892     | 50.662 |       0.934      | 4.012 |
> | Train/Test Ratio (30/70\%) |  0.782 | 28.318 |  0.810 | 25.544 |  0.839 | 147.743 |  0.831  | 20.532 |  0.822 | 215.262 |     0.837     | 56.232 |       0.913      | 4.793 |
> | Train/Test Ratio (20/80\%) |  0.770 | 30.042 |  0.801 | 26.006 |  0.828 | 184.292 |  0.829  | 22.750 |  0.795 | 224.999 |     0.789     | 60.976 |       0.902      | 5.221 |
> | Train/Test Ratio (10/90\%) |  0.748 | 33.364 |  0.797 | 27.897 |  0.816 | 190.397 |  0.811  | 27.577 |  0.789 | 233.126 |     0.771     | 66.226 |       0.874      | 5.843 |
>
> We conducted additional experiments to compare our proposed method, SwiftHydra, with several state-of-the-art tabular anomaly detection methods, including the supervised FTTransformer [1], the unsupervised ECOD [2], and the semi-supervised DevNet [4], PreNet [3], DeepSAD [5], and FEAWAD [6]. While semi-supervised methods require only a small amount of labeled data to achieve per- formance comparable to modern unsupervised methods like DTE, Rejex (as shown in Table 1 as shown in our manuscript) and ECOD, they often face high computational costs, particularly when applied to large datasets. Although PReNET and FEAWAD generally demonstrate fast prediction times, their performance can significantly slow down on specific datasets such as backdoor, celeba, census, donor, fraud, etc, leading to higher overall inference times. This occurs because certain components in these models require computations to be performed on each individual datapoint (without parallelization). As a result, for datasets with an extremely large number of datapoints, the processing time becomes significantly longer.
>
> The supervised FTTransformer, built on robust backbone architectures like ResNet and Transformer, achieves the best AUC-ROC results out of all the SOTA methods compared against SwiftHydra. However, its performance rapidly degrade for scare datasets. In contrast, SwiftHydra provides a distinct advantage by generating unseen data that improve test set coverage. Combined with the Mixture of Mamba Experts (MoME), SwiftHydra consistently outperforms other methods in terms of both AUC-ROC and inference time, making it the most effective and efficient solution for tabular anomaly detection.
>
> [1] Yury Gorishniy, Ivan Rubachev, Valentin Khrulkov, and Artem Babenko. Revisiting deep learning models for tabular data, 2023.
>
> [2] Zheng Li, Yue Zhao, Xiyang Hu, Nicola Botta, Cezar Ionescu, and George Chen. Ecod: Unsuper- vised outlier detection using empirical cumulative distribution functions. IEEE Transactions on Knowledge and Data Engineering, 2022.
>
> [3] Guansong Pang, Chunhua Shen, Huidong Jin, and Anton van den Hengel. Deep weakly-supervised anomaly detection. ArXiv, 1910.13601, 2019a. URL https://arxiv.org/abs/1910. 13601.
>
> [4] Guansong Pang, Chunhua Shen, and Anton van den Hengel. Deep anomaly detection with deviation networks, 2019b. URL https://arxiv.org/abs/1911.08623.
>
> [5] Lukas Ruff, Robert A. Vandermeulen, Nico G¨ornitz, Alexander Binder, Emmanuel M¨uller, Klaus- Robert M¨uller, and Marius Kloft. Deep semi-supervised anomaly detection. In ICLR. OpenRe- view.net, 2020.
>
> [6] Yingjie Zhou, Xucheng Song, Yanru Zhang, Fanxing Liu, Ce Zhu, and Lingqiao Liu. Feature encoding with autoencoders for weakly supervised anomaly detection. TNNLS, 2021.

---

> ### Author Response · Authors · 2024-11-19
>
> Question 3: Could you please add ablation experiments?
>
> Answer: The original manuscript (Appendix C3) already contains ablation experiments for the detector backbone (Transformer or Mamba), and the Self-Reinforcing module. The results show that the AUC-ROC of both Transformer and Mamba models improves significantly when our Self- Reinforcing module is added. The results also show that Mamba outperforms the Transformer backbone for both AUC-ROC and TIF.
>
> We also conducted an additional ablation study (which will be included in Appendix C3 of the revised manuscript) comparing MoME using our proposed probabilistic approach for cluster assignment with traditional MoME. The results are shown below and demonstrate that the probabilistic cluster assignment improves the AUC-ROC.
>
> |           Method           | Swift Hydra (MoME-Traditional Training Approach) |       | Swift Hydra (MoME-Our Training Approach) |       |
> |:--------------------------:|:----------------------------:|:-----:|:----------------:|:-----:|
> |                            |            AUCROC            |  TIF  |      AUCROC      |  TIF  |
> | Train/Test Ratio (40/60\%) |             0.892            | 4.015 |       0.934      | 4.012 |
> | Train/Test Ratio (30/70\%) |             0.865            | 4.771 |       0.913      | 4.793 |
> | Train/Test Ratio (20/80\%) |             0.853            | 5.234 |       0.902      | 5.221 |
> | Train/Test Ratio (10/90\%) |             0.828            | 5.922 |       0.874      | 5.843 |
>
> Furthermore, looking at Table 1 as shown in our manuscript, we can see that the MoME detector achieves a higher average AUCROC across 57 datasets compared to the large single model, while also having a much lower prediction time. This highlights the contribution of the MoME module within the overall framework.

---

> ### Author Response · Authors · 2024-11-19
>
> Question 4: In Table 1, why Swift Hydra (MOME) has less inference time than Swift Hydra (Single)?
>
> Answer: In Swift Hydra with MoME, each expert model is lightweight with a two-layer depth (See Appendix C.1), with each expert learning a specific region of the data. During inference, the routing network of MoME directs the input to the two most suitable experts (k=2), so the prediction time relies only on two lightweight experts. By contrast, a single model (without MoME) requires a larger number of parameters making it significantly slower than MoME. This aligns with Nguyen et al. (2024), demonstrating that we can scale the number of MoME experts indefinitely to capture the data’s complexity without increasing inference complexity. The number of parameters for each model is recapped in the table below.
>
> |               AI Module Type              |                                                  | Number of Parameters | Training time per batch | Total training time |
> |:-----------------------------------------:|--------------------------------------------------|:--------------------:|:-----------------------:|:-------------------:|
> |              C-VAE Generator              |                                                  |        458,907       |          0.0011         |        3.265        |
> |       Mamba-based Detector (Single)       |                                                  |        274,542       |          0.0083         |        8.723        |
> | Mixture of Mamba  Experts Detector (MoME) | Gating Network                                   |         6,164        |          0.0001         |        0.286        |
> |                                           | Expert Network                                   |        33,021        |          0.0026         |        1.287        |
> |                                           | Mixture of Mamba Experts  (20 Experts, topK = 2) |        666,584       |          0.0032         |       489.003       |

---

> ### Author Response · Authors · 2024-11-19
>
> Question 5: Could you please further explain how to use ”gating network” and ”Tackling winner- take-all” together?
>
> Answer: We first want to clarify that the gating network is a mechanism for assigning inputs to experts in MoE, while "winner-take-all" is a phenomenon that occurs in the early training steps of MoE. Given an input sample $x$, the gating network decides which experts would best handle the sample. It does so by selecting the top-$k$ experts based on their respective performance scores (represented by the function $h(x, \cdot)$ in Equ. 4, page 6)
>
> Now, during the early training steps, all the experts have not yet converged and have arbitrary performance scores, resulting in a scenario in which one expert could randomly receive more samples than others via the gating network mechanism. By having more samples to train on, this expert performs significantly better than others (i.e., having a higher performance score) and thus continues to receive more samples via the gating network. Over time, this reduces the MoE to a single expert and hinders the generalization ability of the MoE. This phenomenon is referred to as **"winner-take-all"**.
>
> To tackle the "winner-take-all" problem, during the early training steps, we temporarily deactivate the gating network and instead use a probabilistic approach for assigning samples (Equ. 6, page 6). This ensures that all experts learn equally during this early stage. Once the experts are well-trained, we activate the gating network and use it for routing samples.

---

> > ### Comment · Reviewer_9izT · 2024-11-21
> >
> > Thank you for answering my questions. I appreciate the extensive experiments that authors have made in the rebuttal process. My questions from 1 to 5 are thoroughly solved. Thus, I will raise the score of soundness.

---

> > > ### Author Response · Authors · 2024-11-22
> > >
> > > We are delighted to have addressed all your questions. Thank you for your constructive comments and for increasing the soundness rating for our work. We will be happy to answer any follow-up questions you might have for us.
> > >
> > > Best regards,
> > >
> > > Authors

---

### Official Review · Reviewer_RKHh · 2024-11-03

**Soundness:** 2
**Presentation:** 3
**Contribution:** 2
**Rating:** 6
**Confidence:** 3

**Summary:**

This paper presents a novel framework, Swift Hydra, for training an anomaly detection method that leverages generative AI and reinforcement learning. It introduces Mamba models structured as a Mixture of Experts (MoE), allowing for scalable adaptation of the number of Mamba experts based on data complexity. Experiments conducted on the ADBench benchmark demonstrate that Swift Hydra outperforms other state-of-the-art anomaly detection models while maintaining a relatively short inference time.

**Strengths:**

1.The writing in this paper is clear and well-structured.

2.The incorporation of reinforcement learning, conditional VAE, and Mixture of Experts (MoE) is novel and well-justified.

3.The paper conducts thorough experiments on ADBench and demonstrates strong performance.

**Weaknesses:**

1.Typically, experiments relying on only one dataset are insufficient. It would be beneficial to include additional analysis and comparisons using other popular datasets.

2.It appears that Swift Hydra is not specifically designed for tabular anomaly detection. Have you explored the generalizability of this method to other domains, such as image data or time-series data?

3.The paper introduces several complex components, such as reinforcement learning, which may complicate implementation. Additionally, could you provide an analysis of the computational cost associated with training, including metrics such as the number of parameters and training time per batch?

4.How does the performance of Swift Hydra fare in multi-class or cross-dataset scenarios?

**Questions:**

See weakness above. I will consider raising my score if my concerns are addressed well.

---

> ### Author Response · Authors · 2024-11-19
>
> Question 1. Typically, experiments relying on only one dataset are insufficient. It would be benefi- cial to include additional analysis and comparisons using other popular datasets.
>
> Answer: In our experiment, we tested on ADBench, which is not a single dataset; ADBench actually consists of 57 different datasets, and Table 1 (as shown in our manuscript, page 7) presents the average AUCROC across these 57 datasets. Additionally, the authors of DTE (published at ICLR 2024) also performed comparisons exclusively on ADBench.
> Furthermore, Table 5 (Appendix C.6) provides detailed performance of Swift Hydra on each dataset compared to SOTA methods. Specifically, we compared the F1-score, Recall, Precision, and AUCROC of Swift Hydra with other SOTA methods on each individual dataset in ADBench. Moreover, similar to question 2 from reviewer 2, to enhance the persuasiveness of Swift Hydra's experiment on different datasets,  we have run new experiments and updated the comparison to include more SOTA methods [1,2,3,4,5,6] along with Swift Hydra across the 57 datasets in ADBENCH, as shown in the table below:
>
> | Method                     |  ECOD  |        | DevNet |        | PReNet |         | DeepSAD |        | FEAWAD |         | FTTransformer |        | SwifHydra (MoME) |       |
> |----------------------------|:------:|--------|:------:|--------|:------:|---------|:-------:|--------|:------:|---------|:-------------:|--------|:----------------:|:-----:|
> |                            | AUCROC |   TIF  | AUCROC |   TIF  | AUCROC |   TIF   |  AUCROC |   TIF  | AUCROC |   TIF   |     AUCROC    |   TIF  |      AUCROC      |  TIF  |
> | Train/Test Ratio (40/60\%) |  0.803 | 26.002 |  0.822 | 24.084 |  0.853 | 128.709 |  0.843  | 19.020 |  0.856 | 168.615 |     0.892     | 50.662 |       0.934      | 4.012 |
> | Train/Test Ratio (30/70\%) |  0.782 | 28.318 |  0.810 | 25.544 |  0.839 | 147.743 |  0.831  | 20.532 |  0.822 | 215.262 |     0.837     | 56.232 |       0.913      | 4.793 |
> | Train/Test Ratio (20/80\%) |  0.770 | 30.042 |  0.801 | 26.006 |  0.828 | 184.292 |  0.829  | 22.750 |  0.795 | 224.999 |     0.789     | 60.976 |       0.902      | 5.221 |
> | Train/Test Ratio (10/90\%) |  0.748 | 33.364 |  0.797 | 27.897 |  0.816 | 190.397 |  0.811  | 27.577 |  0.789 | 233.126 |     0.771     | 66.226 |       0.874      | 5.843 |
>
> In the table, semi-supervised methods such as DevNet [4], PReNet [3], DeepSAD [5], and FEAWAD [6] require only a small amount of labeled data to achieve performance comparable to modern unsupervised methods like DTE, Rejex (as shown in Table 1 on page 7 of our manuscript), and ECOD [2]. However, these semi-supervised approaches often face high computational costs, particularly when applied to large datasets. While PReNet and FEAWAD generally exhibit fast prediction times, their performance can significantly slow down on specific datasets in ADBENCH such as Backdoor, CelebA, Census, Donor, and Fraud, resulting in higher overall inference times. This is because certain components in these models rely on computations for each individual datapoint without parallelization, leading to substantial processing delays for datasets with a large number of datapoints.
> The supervised FTTransformer [1], built on robust backbone architectures like ResNet and Transformer, achieves impressive AUC-ROC results. However, its reliance on labeled data limits its applicability in real-world scenarios where labeled data is often scarce. In contrast, our proposed SwiftHydra method provides a distinct advantage by generating unseen data to enhance test set coverage. Combined with its efficient training process leveraging scalable AI techniques such as Mixture of Mamba Experts (MoME), SwiftHydra consistently outperforms other methods in both AUC-ROC and inference time, establishing itself as the most effective and efficient solution for tabular anomaly detection.
>
> [1] Yury Gorishniy, Ivan Rubachev, Valentin Khrulkov, and Artem Babenko. Revisiting deep learning models for tabular data, 2023.
>
> [2] Zheng Li, Yue Zhao, Xiyang Hu, Nicola Botta, Cezar Ionescu, and George Chen. Ecod: Unsuper- vised outlier detection using empirical cumulative distribution functions. IEEE Transactions on Knowledge and Data Engineering, 2022.
>
> [3] Guansong Pang, Chunhua Shen, Huidong Jin, and Anton van den Hengel. Deep weakly-supervised anomaly detection. ArXiv, 1910.13601, 2019a. URL https://arxiv.org/abs/1910. 13601.
>
> [4] Guansong Pang, Chunhua Shen, and Anton van den Hengel. Deep anomaly detection with deviation networks, 2019b. URL https://arxiv.org/abs/1911.08623.
>
> [5] Lukas Ruff, Robert A. Vandermeulen, Nico G¨ornitz, Alexander Binder, Emmanuel M¨uller, Klaus- Robert M¨uller, and Marius Kloft. Deep semi-supervised anomaly detection. In ICLR. OpenRe- view.net, 2020.
>
> [6] Yingjie Zhou, Xucheng Song, Yanru Zhang, Fanxing Liu, Ce Zhu, and Lingqiao Liu. Feature encoding with autoencoders for weakly supervised anomaly detection. TNNLS, 2021.

---

> ### Author Response · Authors · 2024-11-19
>
> Question 2. It appears that Swift Hydra is not specifically designed for tabular anomaly detection. Have you explored the generalizability of this method to other domains, such as image data or time- series data?
>
> Answer: Dear Reviewer, please note that ADBench includes both tabular datasets and image-based formats, so our experiments also demonstrate Swift Hydra’s ability to generalize to image data. For time-series data, we have not conducted tests yet. However, we don’t believe anything precludes our framework from performing well on timeseries data because 1) Swift Hydra is a general self-reinforcing generative framework, making it adaptable to various data types, 2) in terms of dimensionality, time-series data would sit between tabular and images data, for which our framework already performed well. Specifically, for time-series data, our method would generate realistic time-series samples, enabling a suitable detector (designed for time-series data) to improve its performance by learning from the generated samples.

---

> ### Author Response · Authors · 2024-11-19
>
> Question 3. The paper introduces several complex components, such as reinforcement learning, which may complicate implementation. Additionally, could you provide an analysis of the com- putational cost associated with training, including metrics such as the number of parameters and training time per batch?
>
> Answer: RL is a key component that helps the generative model produce more diverse and novel anomalies, and thus it’s essential. For instance, if we relied solely on a generative model (such as diffusion models, GANs, or VAEs), it would only produce high-quality data if the initial data were of high quality. The main limitation is that these models primarily capture features that appear most frequently. However, in anomaly detection, rare features are often critical characteristics of anomalies. By applying RL, we designed reward functions to ensure that the C-VAE generates the most diverse anomalies possible (by incorporating an entropy term in the reward), encouraging the generative model to produce anomaly samples with rare features by chance. These anomalies with rare features are gradually added back into the original dataset, increasing the presence of rare features and enabling the generative model to better capture them, ultimately creating more diverse and challenging data for the detector in subsequent steps.
> Per your request, we have conducted a new set of experiments to demonstrate the number of parameters, training time per batch (in seconds), and total training time for each model (in seconds). The results are presented below.
>
> |               AI Module Type              |                                                  | Number of Parameters | Training time per batch | Total training time |
> |:-----------------------------------------:|--------------------------------------------------|:--------------------:|:-----------------------:|:-------------------:|
> |              C-VAE Generator              |                                                  |        458,907       |          0.0011         |        3.265        |
> |       Mamba-based Detector (Single)       |                                                  |        274,542       |          0.0083         |        8.723        |
> | Mixture of Mamba  Experts Detector (MoME) | Gating Network                                   |         6,164        |          0.0001         |        0.286        |
> |                                           | Expert Network                                   |        33,021        |          0.0026         |        1.287        |
> |                                           | Mixture of Mamba Experts  (20 Experts, topK = 2) |        666,584       |          0.0032         |       489.003       |
>
> As shown in the table above, we have shown the number of parameters for each model, including the C-VAE Generator, Mamba-based Detector (Single model), and Mixture of Mamba Expert Detector (MoME). On average, a single expert in MoME has approximately 33,021 parameters, but due to the presence of 20 experts and a gating network, the total parameter count for MoME is 666,584. In contrast, the Single Mamba-based Detector has around 274,542 parameters, while the C-VAE generator has about 458,907 parameters.
> Besides showing the training time per batch alongside the number of parameters, we have also measured the total training time for each stage of our algorithm. It’s important to note that the total training time for the C-VAE generator and Single Mamba-based Detector is calculated per episode and does not represent the entire RL process. On the other hand, the total training time for the Mixture of Mamba Expert Detector is computed after completing phase 1 using all data generated during that phase.
> All these new results will be included in the Appendix.

---

> ### Author Response · Authors · 2024-11-19
>
> Question 4. How does the performance of Swift Hydra fare in multi-class or cross-dataset scenarios?
>
> Answer: Since our anomaly detection problem only involves two output classes, normal and anomalous, testing on multi-class is not within the scope, we believe.
> Thank you for your interesting question on cross-dataset scenarios! As ADBench provides an excellent benchmark on which the most existing anomaly detection works were evaluated, we also used the same evaluation approaches, testing on all individual 57 datasets. We believe that none of the work has been evaluated in cross-data set scenarios in ADBench due to the challenge of verifying the relevance or correlation between these datasets. However, if these datasets are somehow correlated, it would be possible to leverage our proposed Self-Reinforcing Generative Model to generate more diverse data and capture features from other datasets. This could lead to an increase in the overall performance of the detector in Swift Hydra.

---

> ### Author Response · Authors · 2024-11-22
>
> Dear Reviewer RKHh
>
> Thank you again for the detailed feedback. We hope our responses have addressed your questions sufficiently. We are happy to discuss further if you have follow-up questions for us.
>
> Best regards,
>
> Authors

---

> ### Author Response · Authors · 2024-11-24
>
> Dear Reviewer RKHh,
>
> We'd like to kindly remind you of our recent response. If we have sufficiently addressed your questions, we'd greatly appreciate it if you could adjust your rating as mentioned in your review.
>
> Should you have any follow-up questions or require further clarification, we would be more than happy to address them. Thanks again for your time and constructive comments.
>
> -Authors

---

> > ### Comment · Reviewer_RKHh · 2024-11-26
> >
> > Thank you to the authors for their response and hard work, which effectively addressed most of my concerns. I decide to raise my score.

---

> > > ### Author Response · Authors · 2024-11-26
> > > **Thank you**
> > >
> > > Dear Reviewer RKHh,
> > >
> > > Thank you for increasing the rating and for appreciating our work. We would be happy to assist should you have any follow-up questions or wish to discuss further.
> > >
> > > Again, we appreciate the time and effort you have dedicated to reviewing our paper and engaging in discussions.
> > >
> > > -Authors

---

> > > ### Author Response · Authors · 2024-11-30
> > >
> > > Dear Reviewer RKHh,
> > >
> > > Thank you for dedicating your time to review our paper. Your insightful comments have been invaluable in strengthening our work.
> > >
> > > In response to your feedback, we have addressed the weaknesses by clarifying the benchmark of 57 datasets and conducting many more experiments during this rebuttal period to further improve the manuscript. Given that a score of 6 is considered "marginally above the acceptance threshold," we believe an adjustment to 8, which corresponds to "accept, good paper" on the ICLR scoring metrics, would more accurately reflect your positive evaluation of our paper as *"novel," "well-justified,"* and demonstrating *"strong performance."*
> > >
> > > We sincerely hope that the enhancements made to our manuscript merit a higher rating. Thank you once again for taking the time to read our paper and engage in discussions!
> > >
> > >
> > > Sincerely,
> > >
> > > Authors

---

### Author Response · Authors · 2024-11-29

Dear Reviewers and Area Chair,

We sincerely appreciate your valuable time and effort spent reviewing our manuscript.

In this work, we present a new framework, Swift Hydra, for training an anomaly detection method that leverages Reinforcement Learning to guide Generative AI to generate diverse and challenging data. This approach enables the detector to better generalize to unseen anomalies. Additionally, we introduce Mamba models, structured as a Mixture of Mamba Experts (MoME). This design allows for scalable adaptation of the number of Mamba experts based on the complexity of the data.

As highlighted by the reviewers, our method is both novel and well-justified (RKHh, 9izT, 8B4d), providing interesting insights (uiUH), and our paper is well-written and well-structured (RKHh, 9izT). We are grateful for your constructive feedback, which has been instrumental in improving our manuscript. In response, we have incorporated our discussion into the manuscript to further strengthen it as follows:
________________________________________
**Clarifications**

- **Dataset Diversity (Reviewer RKHh)**. Reviewer RKHh expressed concern that Swift Hydra was only tested on a single dataset, ADBench. We clarified in the main paper that ADBench actually consists of 57 datasets spanning diverse domains, including Image, Healthcare, and Natural Language Processing (lines 340-343). Additionally, we included a detailed table in Appendix C.6 that summarizes the number of data points, the percentage of anomalies, and the domain of each dataset.

- **Model Size and Training Cost (Reviewer RKHh)**. Reviewer RKHh requested a discussion on model size and training cost. In response, we provided a detailed analysis in Appendix C.1.2, outlining the number of parameters and training time for each module, including the CVAE, Gating Network, Experts, Single Detector, and others.

- **Winner-Take-All Behavior (Reviewer 9izT)**. We clarified the occurrence and mechanism of the winner-take-all behavior in the MoE and its gating network (lines 297–302). Furthermore, we detailed how Swift Hydra addresses this phenomenon. Additionally, we emphasized when the gating network is deactivated (lines 303–304) and when it is activated (lines 335–336).

- **Typographical Corrections (Reviewer 9izT)**. We fixed the typo in line 242, as pointed out by reviewer 9izT.

- **Backpropagation on the Reward Function (Reviewer 8B4d)**. To address reviewer 8B4d's concern, we clarified the conditions under which direct backpropagation on the reward function can be used to identify feasible actions (i.e., early training episodes) in lines 251–255.

- **Algorithm Updates (Reviewer 8B4d)**. Following reviewer 8B4d's suggestion, we updated the pseudocode in Algorithm 1 (Appendix A.1) to specify how the balanced dataset is updated in lines 851-854.

- **Trim Function (Reviewer 8B4d)**. We explained the circumstances under which the Trim function (a helper function) might trim out generated data and detailed the strategies for handling such cases (lines 877-884).

- **Normal Data Generation (Reviewer 8B4d)**: We added a point that Swift Hydra can be utilized for generating and synthesizing data in other application contexts where collecting real data is expensive and scarce (lines 534–535), as discussed with Reviewer 8B4d.
________________________________________
**Experiments**

- **Comparisons with SOTA Methods (Reviewer 9izT)**. Following reviewer 9izT's suggestion, we conducted additional comparisons of Swift Hydra with more state-of-the-art (SOTA) detection models on the ADBench benchmark. A detailed analysis of these comparisons is included in Appendix C.2.1.

- **Ablation Studies (Reviewer 9izT)**. Based on reviewer 9izT's feedback, we reorganized the ablation study section and conducted additional experiments (Appendix C.4) to evaluate:
  - The impact of the Self-Reinforcing Module (Appendix C.4.1).
  - The effectiveness of probabilistic cluster assignments (Equation 6) (Appendix C.4.2).
  - The influence of the KL term and reconstruction term in Equation 2 on the AUC-ROC of Swift Hydra (Appendix C.4.3).

- **Toy Data Experiment (Reviewer uiUH)**. As suggested by reviewer uiUH, we added a two-dimensional toy data experiment to illustrate the generalization ability of Swift Hydra in Appendix C.3. This experiment demonstrates how the generated unseen anomalies tighten the decision boundary of MoME around normal data.

In the revised manuscript, these updates are temporarily highlighted in blue for your convenience. Once again, we sincerely appreciate your thoughtful feedback and the opportunity to improve our manuscript.

Best regards,

Authors

---

### Meta-Review · Area_Chair_umac · 2024-12-21

**Metareview:**

**Overview.** The work addresses a weakly-supervised anomaly detection problem where incomplete set of anomaly examples are given for training detectors in a way that can generalize to unseen anomalies. It introduces a method that utilizes reinforcement learning to guide generative models (conditional VAE) to generate diverse synthetic anomaly samples, and then uses the generated samples to train a MoE + Mamba-based anomaly detector. The method is verified on ADBench.

**Pros.**
- Well-written paper [RKHh, 9izT]
- The method effectively leverages RF, conditional VAE, MoE, and Mamba for the studied AD setting, and its effectiveness is empirically and theoretically justified [RKHh, 9izT, 8B4d, uiUH]
- The method shows promising results on tabular data benchmark ADBench [RKHh]
- The visualization results justify the quality of generating diverse anomalies [9izT, uiUH]

**Cons.**
- Only one dataset is used in the experiments [RKHh]
- The method seems data-type-agnostic, but its performance on handling complex structure/dependency in non-tabular data such as image/time series data is unclear [RKHh]
- Lack of computational complexity analysis [RKHh]
- Missing state-of-the-art competing methods on tabular data [9izT]
- No ablation study to verify the effectiveness of RL and MoE [9izT]
- Unrealistic assumption in theorem 2 and theorem 3 [9izT]
- The role and the advantages of the proposed modules against existing similar methods are unclear [uiUH]

**Additional Comments On Reviewer Discussion:**

All four reviewers participate in the discussion. The author rebuttal helps address most of the concerns from [RKHh], resulting in an increased rating to weak accept, and it also clarifies a number of questions from [uiUH, 8B4d]. The concerns on "missing state-of-the-art competing methods on tabular data" and "no ablation study to verify the effectiveness of RL and MoE" from [9izT] have also been addressed in the rebuttal, resulting in an increased rating in the soundness.

The issues of having unrealistic assumption in theorem 2 and theorem 3  are not properly addressed. The authors should clarify these in the final version of the paper.

In the rebuttal, the authors made a claim that the performance on ADBench shows the method's effectiveness on image data.  This is a not properly justified claim. To justify the claim, new comparison should be added to demonstrate the method's superiority over state-of-the-art image AD methods proposed in the computer vision community that show highly impressive performance on the image datasets in ADBench, e.g., AUC of 0.98+ on MVTec. Such comparison results are not found in the paper.

Besides, the detailed results on all image datasets are not given. The authors state that Appendix C.7 presents class-wise detection results for the 57 datasets in ADBench, but it clearly excludes some datasets, such as image datasets. The authors are suggested to provide detailed detection results for all the missing datasets in the final version.

While the work still has room for improvement, the proposed method is novel and shows impressive performance on datasets from ADBench. I therefore recommend acceptance for this paper.

---

### Decision · Program_Chairs · 2025-01-22

Accept (Poster)